# Polar stratospheric nitric acid depletion surveyed from a decadal dataset of IASI total columns

Catherine Wespes[1,a,*], Gaetane Ronsmans[1,a], Lieven Clarisse[1], Susan Solomon[2], Daniel Hurtmans[1], Cathy Clerbaux[1,3], and Pierre-François Coheur[1]

[1]Université libre de Bruxelles (ULB), Spectroscopy, Quantum Chemistry and Atmospheric Remote Sensing (SQUARES), Brussels, Belgium
[2]Department of Earth, Atmospheric and Planetary Sciences, Massachusetts Institute of Technology, Cambridge, Massachusetts, USA
[3]LATMOS/IPSL, Sorbonne Université, UVSQ, CNRS, Paris, France

[a] Co-first authors
[*] Corresponding author: Catherine Wespes (catherine.wespes@ulb.be)

## Abstract

In this paper, we exploit the first 10-year data-record (2008-2017) of nitric acid ($HNO_3$) total columns measured by the IASI-A/Metop infrared sounder, characterized by an exceptional daily sampling and a good vertical sensitivity in the lower-to-mid stratosphere (around 50 hPa), to monitor the relationship between the temperature decrease and the observed $HNO_3$ loss that occurs each year in the Antarctic stratosphere during the polar night. Since the $HNO_3$ depletion results from the formation of polar stratospheric clouds (PSCs) which trigger the development of the ozone ($O_3$) hole, its continuous monitoring is of high importance. We verify here, from the 10-year time evolution of $HNO_3$ together with temperature (taken from reanalysis at 50 hPa), the recurrence of specific regimes in the annual cycle of IASI $HNO_3$ and identify, for each year, the day and the 50 hPa temperature ("drop temperature") corresponding to the onset of strong $HNO_3$ depletion in the Antarctic winter. Although the measured $HNO_3$ total column does not allow the uptake of $HNO_3$ by different types of PSC particles along the vertical profile to be differentiated, an average drop temperature of $194.2 \pm 3.8$ K, close to the nitric acid trihydrate (NAT) existence threshold (~195 K at 50 hPa), is found in the region of potential vorticity lower than $-10 \times 10^{-5}$ K.m$^2$.kg$^{-1}$.s$^{-1}$ (similar to the $70° – 90°$ S equivalent latitude region during winter). The spatial distribution and inter-annual variability of the drop temperature are investigated and discussed. This paper highlights the capability of the IASI sounder to monitor the evolution of polar stratospheric $HNO_3$, a key player in the processes involved in the depletion of stratospheric $O_3$.

## 1 Introduction

The cold and isolated air masses found within the polar vortex during winter are associated with a strong denitrification of the stratosphere due to the formation of PSCs (composed of $HNO_3$, sulphuric acid ($H_2SO_4$) and water ice ($H_2O$)) (e.g. Peter, 1997; Voigt et al., 2000; von König, 2002; Schreiner et al., 2003; Peter and Grooß, 2012). These clouds strongly affect the polar chemistry by (1) acting as surfaces for the heterogeneous activation of chlorine and bromine compounds, in turn leading to enhanced O3 destruction (e.g. Solomon, 1999; Wang and Michelangeli, 2006; Harris et al., 2010; Wegner et al., 2012) and by (2) removing gas-phase $HNO_3$ temporarily or permanently through uptake by PSCs and sedimentation of large PSC particles to lower altitudes. The denitrification of the polar stratosphere during winter delays the reformation of ClONO2, a chlorine reservoir, and, hence, intensifies the O3 hole (e.g. Solomon, 1999; Harris et al., 2010; Tritscher et al., 2021). The heterogeneous reaction rates on PSC surfaces and the uptake of $HNO_3$ strongly depend on the temperature and on the PSC particle type. The

PSCs are classified into three different types based on their composition and optical properties: type Ia
solid nitric acid trihydrate - NAT ($HNO_3 \cdot (H_2O)_3$), type Ib liquid supercooled ternary solution - STS
($HNO_3/H_2SO_4/H_2O$ with variable composition) and type II, crystalline water-ice particles (likely
composed of a combination of different chemical phases) (e.g. Toon et al., 1986; Koop et al., 2000;
Voigt et al., 2000; Lowe and MacKenzie, 2008). In the stratosphere, they mostly consist of mixtures of
liquid/solid STS/NAT particles in varying number densities, with $HNO_3$ being the major constituent of
these particles. The large-size NAT particles of low number density are the principal cause of
sedimentation (Lambert et al., 2012; Pitts et al., 2013; Molleker et al., 2014; Lambert et al., 2016). The
formation temperature of STS ($T_{STS}$) and the thermodynamic equilibrium temperatures of NAT ($T_{NAT}$)
and ice ($T_{ice}$) have been determined, respectively, as: ~192 K (Carslaw et al., 1995), ~195.7 K (Hanson
and Mauersberger, 1988) and ~188 K (Murphy and Koop, 2005) for typical 50 hPa atmospheric
conditions (5 ppmv $H_2O$ and 10 ppbv $HNO_3$). While the NAT nucleation was thought to require pre-
existing ice nuclei, hence, temperatures below $T_{ice}$ (e.g. Zondlo et al., 2000; Voigt et al., 2003), recent
observational and modelling studies have shown that $HNO_3$ starts to condense in early PSC season in
liquid NAT mixtures well above $T_{ice}$ (~4 K below $T_{NAT}$, close to $T_{STS}$) even after a very short temperature
threshold exposure (TTE) to these temperatures but also slightly below $T_{NAT}$ after a long TTE, whereas
the NAT existence persists up to $T_{NAT}$ (Pitts et al., 2013; Hoyle et al., 2013; Lambert et al., 2016; Pitts
et al., 2018). It has been recently proposed that the higher temperature condensation results from
heterogeneous nucleation of NAT on meteoritic dust in liquid aerosol (Voigt et al., 2005; Hoyle et al.,
2013; Großß et al., 2014; James et al., 2018; Tritscher et al., 2021). Further cooling below $T_{STS}$ and $T_{ice}$
leads to nucleation of liquid STS, of solid NAT onto ice and of ice particles mainly from STS (type II
PSCs) (Lowe and MacKenzie, 2008). The formation of NAT and ice has also been shown to be triggered
by stratospheric mountain-waves (Carslaw et al., 1998; Hoffmann et al., 2017). Although the formation
mechanisms and composition of STS droplets in stratospheric conditions are well described (Toon et al.,
1986; Carslaw et al., 1995; Lowe and MacKenzie, 2008), the NAT and ice nucleation processes still
require further investigation (Tritscher et al., 2021). This could be important as the chemistry-climate
models (CCMs) generally oversimplify the heterogeneous nucleation schemes for PSC formation (Zhu
et al., 2015; Spang et al., 2018; Snels et al., 2019), preventing an accurate estimation of $O_3$ levels.
Over the last few decades, several satellite instruments have measured stratospheric $HNO_3$ (e.g.
MLS/UARS (Santee et al., 1999), MLS/Aura (Santee et al., 2007), MIPAS/ENVISAT (Piccolo and
Dudhia, 2007), ACE-FTS/SCISAT (Sheese et al., 2017) and SMR/Odin (Urban et al., 2009)).
Spaceborne instruments such as the CALIOP/CALIPSO lidar and MIPAS/Envisat measuring in the
infrared are capable of detecting and classifying PSC types, allowing their formation mechanisms to be
investigated (Lambert et al., 2016; Pitts et al., 2018; Spang et al., 2018, Tritscher et al., 2021 and
references therein); these satellite data complement in situ measurements (Voigt et al., 2005) and ground-
based lidar (Snels et al., 2019). From these available observational datasets, $HNO_3$ depletion has been
linked to PSC formation and detected below the $T_{NAT}$ threshold (Santee et al., 1999; Urban et al., 2009;
Lambert et al., 2016; Ronsmans et al., 2018), but its relationship to PSCs still needs further investigation
given the complexity of the nucleation mechanisms that depend on several parameters (e.g. atmospheric
temperature, water and $HNO_3$ vapour pressure, time exposure to temperatures, temperature history).
In contrast to the limb satellite instruments mentioned above, the infrared nadir sounder IASI offers a
dense spatial sampling of the entire globe, twice a day (Section 2). While it cannot provide a vertical
profile of $HNO_3$ similar to that from the limb sounders, IASI provides reliable total column
measurements of $HNO_3$ characterized by a maximum sensitivity in the low-middle stratosphere around
50 hPa (20 km) during the dark Antarctic winter (Ronsmans et al., 2016, 2018) where PSCs form (Voigt
et al., 2005; Lambert et al., 2012; Spang et al., 2016, 2018). This study aims to explore the 10-year
continuous $HNO_3$ measurements from IASI to provide a long-term global picture of depletion and of its
dependence on temperatures during polar winter (Section 3). The temperature corresponding to the onset
of the strong depletion in HNO$_3$ records (here referred to as 'drop temperature') is identified in Section
4 for each observed year and discussed in the context of previous studies.
**2 Data**
The HNO$_3$ data used in the present study are obtained from measurements of the Infrared Atmospheric
Sounding Interferometer (IASI) onboard the Metop-A satellite. IASI measures the Earth's and
atmosphere's radiation in the thermal infrared spectral range (645 - 2760 cm$^{-1}$), with a 0.5 cm$^{-1}$ apodized
resolution and a low radiometric noise (Clerbaux et al., 2009; Hilton et al., 2012). Thanks to its polar
sun-synchronous orbit with more than 14 orbits a day and a field of view of four simultaneous footprints
of 12 km at nadir, IASI provides global coverage twice a day (9.30 AM and PM mean local solar time).
That extensive spatial and temporal sampling in the polar regions is key to this study.
The HNO$_3$ vertical profiles are retrieved on a uniform vertical 1 km grid of 41 layers (from the surface
to 40 km with an extra layer above to 60 km) in near-real-time by the Fast Optimal Retrieval on Layers
for IASI (FORLI) software, using the optimal estimation method (Rodgers, 2000). Detailed information
on the FORLI algorithm and retrieval parameters specific to HNO$_3$ can be found in previous papers
(Hurtmans et al., 2012; Ronsmans et al., 2016). For this study, only the total columns (v20151001) are
used, considering (1) the low vertical resolution of IASI with only one independent piece of information
(full width at half maximum - FWHM - of the averaging kernels of ~30 km), (2) the limited sensitivity
of IASI to tropospheric HNO$_3$, (3) the dominant contribution of the stratosphere to the HNO$_3$ total
column and (4) the largest sensitivity of IASI in the region of interest, i.e. in the low and mid-stratosphere
(from ~70 to ~30 hPa), where the HNO$_3$ abundance is the highest (Ronsmans et al., 2016). The IASI
measurements capture the expected depletion of HNO$_3$ within the polar night, as illustrated in Fig. 1 that
shows examples of vertical HNO$_3$ profiles retrieved within the dark Antarctic vortex (above Arrival
Heights) and outside the vortex (above Lauder). The retrieved profiles are shown along with their
associated total retrieval error and averaging kernels (the total column averaging kernel and the so-called
"sensitivity profile" are also represented; see Ronsmans et al., 2016 for more details). The total column
averaging kernel (in black) indicates the sensitivity of the total column measurement to changes in the
vertical distribution of HNO$_3$, hence, the altitude to which the retrieved total column is mainly
sensitive/representative, while the sensitivity profile indicates the extent to which the retrieval at one
specific altitude comes from the spectral measurement rather than the apriori. Above Arrival Heights
during the dark Antarctic winter, we clearly see depleted HNO$_3$ levels in the low and mid-stratosphere
and the altitude of maximum sensitivity at around 30 hPa for this case (values of ~1 along the total
column averaging kernel around that level). In contrast, at Lauder, HNO$_3$ levels larger than the a priori
are observed in the stratosphere with a larger range of maximum sensitivity. The total columns are
associated with a total retrieval error ranging from around 3% at mid- and polar latitudes (except above
Antarctica) to 25% above cold Antarctic surface during winter and with a low absolute bias smaller than
12% when compared to ground-based FTIR measurements, in polar regions over the altitude range
where the IASI sensitivity is the largest (see Hurtmans et al., 2012 and Ronsmans et al., 2016 for more
details). The highest retrieval error measured over the Antarctic arises from a weaker sensitivity above
very cold surface with a degrees of freedom for signal (DOFS) of 0.95, as well as from a poor knowledge
of the seasonally and wavenumber-dependent emissivity above ice surfaces. In order to expand on the
comparisons against FTIR measurements, which cannot be made during the polar night, Fig. 2 (top
panel) presents the time series of daily IASI total HNO$_3$ columns co-located with MLS measurements
within 2.5°x2.5° grid boxes, averaged in the 70°S–90°S equivalent latitude band. In order to account for
the vertical sensitivity of IASI, the averaging kernels associated with each co-located IASI retrieved
profile were applied to the MLS profiles for this cross-comparison. The MLS mixing ratio profiles over
the 215-1.5 hPa pressure range were first interpolated to the FORLI pressure grids and extended down
to the surface by using the FORLI-HNO3 a priori profile, and then converted into partial columns.
Similar variations in the HNO₃ column are captured by the two instruments, with an excellent agreement
in particular for the timing of the strong HNO₃ depletion within the inner vortex core. Note that a similar
good agreement between the two satellite datasets is obtained in other latitude bands (see Fig. 2 bottom
panel for the 50°S–70°S equivalent latitude band; the other bands are not shown).
Quality flags similar to those developed for O₃ in previous IASI studies (Wespes et al., 2017) were
applied a posteriori to exclude data (i) with a corresponding poor spectral fit (e.g. based on quality flags
rejecting biased or sloped residuals, fits with maximum number of iterations exceeded), (ii) with less
reliability (e.g. based on quality flags rejecting suspect averaging kernels, data with less sensitivity
characterized by a DOFS lower than 0.9) or (iii) with tropospheric cloud contamination (defined by a
fractional cloud cover ≥ 25 %). Note that the HNO₃ total column distributions illustrated in sections
below use the median as a statistical average since it is more robust against the outliers than the mean.
Temperature and potential vorticity (PV) fields are taken from the ECMWF ERA Interim Reanalysis
dataset, respectively at 50 hPa and at the potential temperature of 530 K (corresponding to ~20 km
altitude where the IASI sensitivity to HNO₃ is the highest during the Southern Hemisphere (S.H.) winter
(Ronsmans et al., 2016)). Because the HNO₃ uptake by PSCs starts within a few degrees below $T_{NAT}$
(~195.7 K at 50 hPa (Hanson and Mauersberger, 1988)) depending on the meteorological conditions
(Pitts et al., 2013; Hoyle et al., 2013; Lambert et al., 2016; Pitts et al., 2018), a threshold temperature of
195 K is considered in the sections below to identify regions of potential PSC existence. The potential
vorticity is used to delimit dynamically consistent areas in the polar regions. In what follows, we use
either the equivalent latitudes ("eqlat", calculated from PV fields at 530 K) or the PV values to
characterize the relationship between HNO₃ and temperatures in the cold polar regions. Uncertainties in
ERA-Interim temperatures will also be discussed below.
**3 Annual cycle of HNO₃ vs temperatures**
Figure 3a shows the yearly HNO₃ cycle (solid lines, left axis) in the southernmost equivalent latitudes
(70° - 90° S) as measured by IASI over the whole study period (2008–2017). The total HNO₃ variability
in such equivalent latitudes has already been discussed in a previous IASI study (Ronsmans et al., 2018),
where the contribution of the PSCs to the HNO₃ variations was highlighted. The temperature time series,
taken at 50 hPa, is represented as well (dashed lines, right axis). From this figure, different regimes of
HNO₃ total columns *vs* temperature can be observed throughout the year and from one year to another.
In particular, we define here three main regimes (R1, R2 and R3) during the HNO₃/temperature annual
cycle. The full cycle and the main regimes in the 70° - 90° S eqlat region are further represented in Fig.
3b that shows a histogram of the HNO₃ total columns as a function of temperature for the year 2011.
Similar histograms are observed for the other years in the 10-year study period (not shown). The orange
horizontal and vertical lines in Fig. 3a and Fig. 3b, respectively, represent the 195 K threshold
temperature used to identify the onset of HNO₃ uptake by PSCs (see Section 2). The three regimes
identified are:
-   R1 is defined by the maxima in the total HNO₃ abundances covering the months of April and
191       May (~3×10¹⁶ molec.cm⁻²), when the 50 hPa temperature strongly decreases (from ~220 to ~195
192       K). These high HNO₃ levels result from low sunlight, preventing photodissociation, along with
193       the heterogeneous hydrolysis of N₂O₅ to HNO₃ during autumn before the formation of polar
194       stratospheric clouds (Keys et al., 1993; Santee et al., 1999; Urban et al., 2009; de Zafra and
195       Smyshlyaev, 2001). This period also corresponds to the onset of the development of the southern

polar vortex, which is characterized by strong diabatic descent with weak latitudinal mixing across its boundary, isolating polar $HNO_3$-rich air from lower-latitude airmasses. The end of the R1 period marks the start of the strong total $HNO_3$ decrease that intensifies later in R2.

- R2, which extends from June to October, follows the onset of the strong decrease in $HNO_3$ total columns that starts around mid-May in most years when the temperatures fall below 195 K. After a steep initial decline in total $HNO_3$, R2 is characterized by a plateau of total $HNO_3$ minima. For much of this regime, average $HNO_3$ total columns are below $2 \times 10^{16}$ molec.cm$^{-2}$ and the 50 hPa temperatures range mostly between 180 and 190 K.

- R3 starts in October when sunlight returns and the 50 hPa temperatures rise above 195 K. Despite 50 hPa temperatures increasing up to 240 K in summer, the $HNO_3$ total columns stagnate at the R2 plateau levels (around $1.5 \times 10^{16}$ molec.cm$^{-2}$). This regime likely reflects the photolysis of $NO_3$ and $HNO_3$ itself (Ronsmans et al., 2018) as well as the permanent denitrification of the mid-stratosphere, caused by sedimentation of PSCs. The likely renitrification of the lowermost stratosphere (e.g. Braun et al., 2019; Lambert et al., 2012), where the $HNO_3$ concentrations and the IASI sensitivity to $HNO_3$ are lower (Ronsmans et al., 2016), cannot be inferred from the IASI total column measurements. The plateau lasts until approximately February, when $HNO_3$ total column slowly starts increasing, reaching the April-May maximum in R1.

As illustrated in Fig. 3a, the three regimes are observed each year with, however, some interannual variations. For instance, the sudden stratospheric warming (SSW) that occurred in 2010 (see the temperature time series at 20 hPa for the year 2010; green dotted line) yielded higher $HNO_3$ total columns (see green solid line in July - September) (de Laat and van Weele, 2011; Klekociuk et al., 2011; WMO, 2014; Ronsmans et al., 2018).

Figure 3c shows the evolution of the relationship between the daily averaged $HNO_3$ (calculated from a 7-day moving average) with the highest occurrence (in bins of $0.1 \times 10^{16}$ molec.cm$^{-2}$ and of 2K) and the 50 hPa temperature, over the 10–year study period. The orange vertical line represents the 195 K threshold temperature. Figure 3c also highlights the large interannual variability in total $HNO_3$ in R3, while the strong depletion in $HNO_3$ in R2 is consistent every year. Given that PSC formation spans a large range of altitudes (typically between 10 and 30 km) (Höpfner et al., 2006, 2009; Spang et al., 2018; Pitts et al., 2018) and that IASI has maximum sensitivity to $HNO_3$ around 50 hPa (Hurtmans et al., 2012; Ronsmans et al., 2016), the temperatures at two other pressure levels, namely 70 and 30 hPa (i.e. ~15 and ~25 km), have also been tested to investigate the relationship between $HNO_3$ and temperature in the low and mid-stratosphere. The results (not shown here) exhibit a similar $HNO_3$-temperature behavior at the different levels with, as expected, lower and higher temperatures in R2, respectively, at 30 hPa and at 70 hPa (temperatures down to ~180 K at 30 hPa and down to ~185 K at 70 hPa, as compared to temperatures down to ~182 K at 50 hPa, are observed), but still below the NAT formation threshold at these pressure levels ($T_{NAT}$ ~193 K at 30 hPa and ~197 K at 70 hPa) (Lambert et al., 2016). Therefore, the altitude range of maximum IASI sensitivity to $HNO_3$ (see Section 2) is characterized by temperatures that are below the NAT formation threshold at these pressure levels, enabling PSC formation and the denitrification process. Furthermore, the consistency between the 195 K threshold temperature taken at 50 hPa and the onset of the strong total $HNO_3$ depletion seen in IASI data (see Fig. 3a) is in agreement with the largest NAT area that starts to develop in June around 20 km (Spang et al., 2018), which justifies the use of the 195 K temperature at that single representative level in this study.

**4 Onset of $HNO_3$ depletion and drop temperature detection**

To identify the spatial and temporal variability of the onset of the depletion phase, the daily time
evolution of $HNO_3$ during the first 10 years of IASI measurements and the temperatures at 50 hPa are
explored. In particular, the second derivative of $HNO_3$ total column with respect to time is calculated to
detect the strongest rate of decrease seen in the $HNO_3$ time series and to identify its associated day and
50 hPa temperature.

**4.1 HNO₃ vs temperature time series**

Figure 4 shows the time series of the second derivative of $HNO_3$ total column (blue) and of the
temperature (red) with respect to time, averaged in the area of potential vorticity smaller than $-10 \times 10^{-5}$
$K.m^2.kg^{-1}.s^{-1}$ at the potential temperature of 530 K to encompass the region inside the inner polar vortex
where the temperatures are the coldest and the largest depletion of total $HNO_3$ occurs (Ronsmans et al.,
2018). The use of that PV threshold value explains the gaps in the time series during the summer when
the PV does not reach such low levels, while the time series averaged in the 70°- 90° S eqlat band (dashed
blue for the second derivative of $HNO_3$ and grey for the temperature) covers the full year. Note that the
$HNO_3$ time series has been smoothed with a simple spline data interpolation function to avoid gaps in
order to calculate the second derivative of $HNO_3$ total column with respect to time as the daily second-
difference in $HNO_3$ total columns. The horizontal red line shows the 195 K threshold.
As already illustrated in Fig. 3a and Fig. 3c, the strongest rate of $HNO_3$ depletion (i.e. the second
derivative minimum) is found closely around the time that temperatures drop below the 195 K threshold
(except for the year 2009 that shows a longest delay), within a few days to a few weeks (4 to 23 days)
after total $HNO_3$ reaches its maximum, i.e. between the 11th of May (2013) and the 8th of June (2009).
The 50 hPa drop temperatures, i.e. the temperature associated with the strongest rate of $HNO_3$ depletion
detected from IASI, are between 189.2 K and 198.6 K, with the exception of the year 2014, which shows
a drop temperature of 202.8 K. On average over the 10 years of studied IASI measurements, a 50 hPa
drop temperature of 194.2 K ± 3.8 K (1σ standard deviation) is found. Knowing that $T_{NAT}$ can be higher
or lower depending on the atmospheric conditions and that NAT starts to nucleate from ~2–4 K below
$T_{NAT}$ (Pitts et al., 2011; Hoyle et al., 2013; Lambert et al., 2016), the results here tend to demonstrate the
consistency between the 50 hPa drop temperature and the PSC existence temperature in that altitude
region. Note that the range observed in the 50 hPa drop temperature could reflect variations in the
preponderance of one type of PSCs over another from one year to the next. The results further justify
the use of the single 50 hPa level for characterizing and investigating the onset of $HNO_3$ depletion from
IASI. Nevertheless, given the range of maximum IASI sensitivity to $HNO_3$ around 50 hPa, typically
between 70 and 30 hPa (Ronsmans et al., 2016), the drop temperatures are also calculated at these two
other pressure levels (not shown here) in order to estimate the uncertainty of the calculated drop
temperature defined in this study at 50 hPa. The 30 hPa and 70 hPa drop temperatures range respectively
over 185.7 K – 194.9 K and over 194.8 K – 203.7 K, with an average of 192.0 ± 2.9 K and 198.0 ± 3.2
K (1σ standard deviation) over the ten years of IASI. The average values at 30 hPa and 70 hPa fall within
the 1σ standard deviation associated with the average drop temperature at 50 hPa. It is also worth noting
the agreement between the drop temperatures and the NAT formation threshold at these two pressure
levels ($T_{NAT}$ ~ 193 K at 30 hPa and ~197 K at 70 hPa) (Lambert et al., 2016). Finally, it should be noted
that, because the size, shape or location of the vortex vary slightly over the altitude range to which IASI
is sensitive (from ~30 to ~70 hPa during the polar night), the use of a single potential temperature surface
for the calculation of drop temperatures could introduce some uncertainties into the results. However,
several tests suggest that these variations of the vortex are overall minor and, hence, have only limited
influence on the identification of the inner polar vortex (delimited by a PV value of $-10 \times 10^{-5} K.m^2.kg^{-1}.s^{-1}$ at 530 K) and on the determination of the average drop temperature inside that region.

Figures 5a and b show the climatological zonal distribution of $HNO_3$ total columns and of the temperature at 50 hPa, respectively, spanning the 55° S - 90° S geographic latitude band over the first ten years of IASI, with, superimposed, three isocontour levels of potential vorticity (-10, -8 and $-5\times10^{-5}$ $K.m^2.kg^{-1}.s^{-1}$ in blue, cyan and black, respectively) and the isocontours for the 195 K temperature (pink) and for the averaged 194.2 K drop temperature (purple) at 50 hPa. They further illustrate the relationship between the IASI total $HNO_3$ columns and the 50 hPa temperatures. The climatological (2008-2017) PV isocontour of $-10\times10^{-5}$ $K.m^2.kg^{-1}.s^{-1}$ is clearly shown to separate well the region of strong depletion in total $HNO_3$, according to the latitude and the time, until October. The red vertical dashed line indicates the average of the dates on which the 50 hPa drop temperatures are calculated in the area of PV$\leq -10\times10^{-5}$ $K.m^2.kg^{-1}.s^{-1}$ (194.2 ± 3.8 K; see Fig. 4) over the first ten years of IASI. It shows that the strongest rate of $HNO_3$ depletion occurs on average at the end of May (24 May), a few days after the temperature decreases below 195 K. The yearly zonally averaged time series over the 10-year study period can be found in Fig. 6, which shows that IASI measures similar $HNO_3$ total column zonal distributions every year, in particular with respect to the edge of the collar region and of the region of strong depletion (respectively delimited by the PV isocontours of $-5\times10^{-5}$ $K.m^2.kg^{-1}.s^{-1}$ and of $-10\times10^{-5}$ $K.m^2.kg^{-1}.s^{-1}$ at 530 K). Like for Fig.4, an exact timing or a few days between the time that temperatures drop below the 195 K threshold and the start of the $HNO_3$ depletion is visible every year in Fig. 6. A longest delay is also observed for the year 2009. Note that the mismatch between the 10-year average of the dates on which the 195 K threshold temperature is reached and that of the dates for the drop temperatures (see Fig. 5 a and b) is driven by the year 2013, which is characterized by the lowest temperatures during the Antarctic winter over the 10-year study period and, hence, the earliest date for the drop temperature (11th of May; see Fig. 4 and Fig. 6).

**4.2 Spatial distribution of drop temperatures**

To explore the capability of IASI to monitor the onset of $HNO_3$ depletion at a large scale, figure 7 shows, for each year of the study period, the spatial distribution of the 50 hPa drop temperatures based on the second derivative minima of total $HNO_3$ averaged in 1°×1° grid cells. The region of interest here is delimited by a PV value of $-8\times10^{-5}$ $K.m^{-2}.kg^{-1}.s^{-1}$ at 530 K, in order to investigate an area a bit larger than the inner vortex core that was the focus of the preceding discussion (delineated in green in figure 7 by the PV isocontour of $-10\times10^{-5}$ $K.m2.kg^{-1}.s^{-1}$ averaged over the interval 10 May to 15 July). The isocontour of $-10\times10^{-5}$ $K.m^2.kg^{-1}.s^{-1}$ for the minimum PV (in cyan) encountered at 530 K over the 10 May to 15 July period for each year, as well as the isocontours of 195 K for the average temperatures and the minimum temperatures, are also represented. The calculated drop temperatures corresponding to the onset of $HNO_3$ depletion inside the averaged PV isocontour are found to vary between ~180 and ~210 K and the corresponding dates range between ~mid-May and mid-July (not shown here). Although the range of drop temperatures and dates for 1°×1° bins is broader than that found for the inner vortex averages discussed above, the results are qualitatively consistent. For example, the year 2014 that shows the highest inner vortex average drop temperature in Figure 4 is characterized by the highest drop temperatures above the eastern Antarctic. Note, however, that the high extremes in the drop temperature, mainly found above the eastern Antarctic, should be considered with caution: they correspond to specific regions above ice surfaces with emissivity features that are known to yield errors in the IASI retrievals (Hurtmans et al., 2012; Ronsmans et al., 2016). Indeed, bright land surfaces such as ice might in some cases lead to poor $HNO_3$ retrievals. Although wavenumber-dependent surface emissivity atlases are used in FORLI (Hurtmans et al., 2012), this parameter remains critical and causes poorer retrievals that, in some instances, pass through the series of quality filters and could affect the drop temperature calculation.

The averaged isocontour of 195 K encircles fairly well the area of $HNO_3$ drop temperatures lower than
195 K (typically from ~187 K to ~195 K), which means that the bins inside that area include airmasses
that experience the NAT threshold temperature during a long time over the 10 May – 15 July period.
That area encompasses the inner vortex core (delimited by the isocontour of $-10\times10^{-5}$ $K.m^2.kg^{-1}.s^{-1}$ for
the PV averaged over the 10 May – 15 July period) and shows pronounced minima (lower than $-0.5\times10^{14}$
$molec.cm^{-2}.d^{-2}$) in the second derivative of the $HNO_3$ total column with respect to time (not shown here),
which indicate a strong and rapid $HNO_3$ depletion. The area enclosed between the two isocontours of
195 K for the temperatures, the averaged one and the one for the minimum temperatures, shows generally
higher drop temperatures and weakest minima (larger than $-0.5\times10^{14}$ $molec.cm^{-2}.d^{-2}$) in the second
derivative of the $HNO_3$ total column (not shown). That area is also typically enclosed by the isocontour
of $-10\times10^{-5}$ $K.m^2.kg^{-1}.s^{-1}$ for the minimum PV, meaning that the bins inside correspond, at least for one
day over the 10 May – 15 July period, to airmasses located at the inner edge of the vortex and
characterized by temperature lower than the NAT threshold temperature. The fact that the weakest
minima in the second derivative of total HNO3 are observed in that area (not shown) indicates a weak
and slow $HNO_3$ depletion that might be explained by air masses at the inner edge of the vortex
experiencing only a short period with temperatures below the NAT threshold temperature. It could also
reflect mixing with strongly $HNO_3$-depleted and colder airmasses from the inner vortex core. Mixing
with these already depleted airmasses could also explain the higher drop temperatures detected in those
bins. These sometimes unrealistic high drop temperatures are generally detected later (after the strong
$HNO_3$ depletion occurs in the inner vortex core, i.e. after the 10 May – 15 July period considered here –
not shown), which supports the transport, in those bins, of previously $HNO_3$-depleted airmasses and the
likely mixing at the edge of the vortex. Note, however, that previous studies have shown a generally
weak mixing in the Antarctic between the edge region and the vortex core (e.g. Roscoe et al., 2012).
Finally, these spatial variations might also partly reflect some uncertainty in the drop temperature
calculation, introduced by the use of temperature at a single pressure level (50 hPa) and of PV on a single
potential temperature surface (530 K) while the sensitivity of IASI to changes in the $HNO_3$ profiles
extends over a range from ~30 to ~70 hPa during the polar night. It should be noted that biases in the
ECMWF ERA Interim temperatures used in this work are too small to explain the large range of drop
temperatures calculated here. Indeed, Lambert and Santee (2018) found only a small warm bias, with
median differences around 0.5 K, reaching 0–0.25 K in the southernmost regions of the globe at ~68–21
hPa where PSCs form, through comparisons with the Constellation Observing System for Meteorology,
Ionosphere and Climate (COSMIC) data.
Except above some parts of Antarctica which are prone to larger retrieval errors and where unrealistic
high drop temperatures are found, the overall range in the 50 hPa drop temperature for total $HNO_3$ inside
the isocontour for the averaged temperature of 195 K typically extends from ~187 K to ~195 K, which
falls within the range of PSC nucleation temperature at 50 hPa: from slightly below $T_{NAT}$ to around 3-4
K below the ice frost point - $T_{ice}$ - depending on atmospheric conditions, on TTE and on the specific
formation mechanism (i.e., the type of PSC developing) (Pitts et al., 2011; Peter and Grooß, 2012; Hoyle
et al., 2013). This underlines well the benefit of the excellent spatial and temporal coverage of IASI,
which allows the rapid and critical depletion phase to be captured in detail over a large scale.
**5 Conclusions**
In this paper, we have explored the added value of the dense $HNO_3$ total column dataset provided by the
IASI/Metop-A satellite over a full decade (2008–2017) for monitoring the stratospheric depletion phase
that occurs each year in the S.H. and for investigating its relationship to the NAT formation temperature.
To that end, we focused on and delimited the coldest polar region of the S.H. using a specific PV value
at 530 K (~50 hPa, PV of $-10\times10^{-5}$ $K.m^2.kg^{-1}.s^{-1}$) and stratospheric temperatures at 50 hPa, taken from
the ECMWF ERA Interim reanalysis. That single representative pressure level has been considered in
this study given the maximum sensitivity of IASI to $HNO_3$ around that level, which lies in the range
where the PSCs formation/denitrification processes occur.
The annual cycle of total $HNO_3$, as observed from IASI, has first been characterized according to the
temperature evolution. Three regimes (R1 to R3) in the total $HNO_3$ - 50 hPa temperature relationship
were highlighted from the time series over the S.H. polar region: R1 is defined during April and May
and characterized by a rapid decrease in 50 hPa temperatures while $HNO_3$ accumulates over the pole;
R2, from June to October, follows the onset of the depletion that starts around mid-May in most years
when the 50 hPa temperatures fall below 195 K (considered here as the onset of PSC nucleation phase
at that level), with a strong consistency from year to year; R3, defined from October through March
when total $HNO_3$ remains at low R2 plateau levels, despite the return of sunlight and heat, characterizes
the strong denitrification of the stratosphere, likely due to PSC sedimentation to lower levels where the
IASI sensitivity is low. For each year over the 10-year study period, the use of the second derivative of
the $HNO_3$ column versus time was then found to be particularly valuable to detect the onset of the $HNO_3$
condensation into PSCs. It is captured, on average from IASI, a few days before June with a delay of 4–
23 days after the maximum in total $HNO_3$. The corresponding temperatures ('drop temperatures') were
detected between 189.2 K and 202.8 K (194.2 ± 3.8 K on average over the 10 years), which tends to
demonstrate the good consistency between the 50 hPa drop temperature and the PSC formation
temperatures in that altitude region. Finally, the annual and spatial variability (within $1° × 1°$) in the drop
temperature was further explored from IASI total $HNO_3$. Inside the isocontours of 195 K for the average
temperatures and of $-10 \times 10^{-5}$ $K.m^2.kg^{-1}.s^{-1}$ for the averaged PV at 530 K, the drop temperatures are
detected between ~mid-May and mid-July, typically range between ~187 K to ~195 K and are associated
with the lowest minima (lower than $-0.5 \times 10^{14}$ $molec.cm^{-2}.d^{-2}$) in the second derivative of the $HNO_3$ total
column with respect to time, indicating a strong and rapid $HNO_3$ depletion. Except for unrealistic drop
temperatures (~210 K) that were found in some years above eastern Antarctica and suspected to result
from unfiltered poor quality retrievals arising from emissivity issues above ice, the range of drop
temperatures is interestingly found to be in line with the PSC nucleation temperature that is known, from
previous studies, to strongly depend on several factors (e.g. meteorological conditions, $HNO_3$ vapour
pressure, temperature threshold exposure, presence of meteoritic dust). At the edge of the vortex,
considering the isocontours of 195 K for the minimum temperatures or of $-10 \times 10^{-5}$ $K.m^2.kg^{-1}.s^{-1}$ for the
minimum PV, higher and later drop temperatures along with weakest minima in the second derivative
of the $HNO_3$ total column with respect to time, indicating a slow $HNO_3$ depletion, are found. These
likely result from a short temperature threshold exposure or mixing with already depleted airmasses from
the inner vortex core. The results of this study highlight the ability of IASI to measure the variations in
total $HNO_3$ and, in particular, to capture and monitor the rapid depletion phase over the whole Antarctic
region.
We show in this study that the IASI dataset allows the variability of stratospheric $HNO_3$ throughout the
year (including the polar night) in the Antarctic to be captured. In that respect, it offers observational
means to monitor the relation of $HNO_3$ to temperature and the related formation of PSCs. Despite the
limited vertical resolution of IASI which does not allow investigation of the $HNO_3$ uptake by the
different types of PSCs during their formation and growth along the vertical profile, the $HNO_3$ total
column measurements from IASI constitute an important new dataset for exploring the strong polar
depletion over the whole stratosphere. This is particularly relevant considering the mission continuity,
which will span several decades with the planned follow-on missions. Indeed, thanks to the three
successive instruments (IASI-A launched in 2006 and still operating, IASI-B in 2012, and IASI-C in
2018) that demonstrate an excellent stability of the Level-1 radiances, the measurements will soon
provide an unprecedented long-term dataset of $HNO_3$ total columns. Further work could also make use
of this unique data set to investigate the relation between $HNO_3$, $O_3$, and meteorology in the changing
climate.

**Data availability**
The IASI $HNO_3$ data processed with FORLI-$HNO_3$ v0151001 are available upon request to the
corresponding author.

**Author contributions**
G.R. and C.W. performed the analysis, wrote the manuscript and prepared the figures. L.C. contributed
to the analysis. S.S., P.-F. C. and L.C. contributed to the interpretation of the results. D.H. was
responsible for the retrieval algorithm development and the processing of the IASI $HNO_3$ dataset. All
authors contributed to the writing of the text and reviewed the manuscript.

**Competing interests**
The authors declare no competing interests.

**Acknowledgements**
IASI has been developed and built under the responsibility of the Centre National d'Etudes Spatiales
(CNES, France). It is flown on board the Metop satellites as part of the EUMETSAT Polar System. The
IASI L1 data are received through the EUMETCast near-real-time data distribution service. The research
was funded by the F.R.S.-FNRS, the Belgian State Federal Office for Scientific, Technical and Cultural
Affairs (Prodex arrangement 4000111403 IASI.FLOW) and EUMETSAT through the Satellite
Application Facility on Atmospheric Composition Monitoring (ACSAF). G. Ronsmans is grateful to the
Fonds pour la Formation à la Recherche dans l'Industrie et dans l'Agriculture of Belgium for a PhD
grant (Boursier FRIA). L. Clarisse is a research associate supported by the F.R.S.-FNRS. C. Clerbaux is
grateful to CNES for financial support. S. Solomon is supported by the National Science Foundation
(NSF-1539972). We also would like to thank the three reviewers for their helpful comments and
corrections and, in particular, M. Santee for her in-depth reviews, which have substantially improved
the paper quality.

**Figure captions**

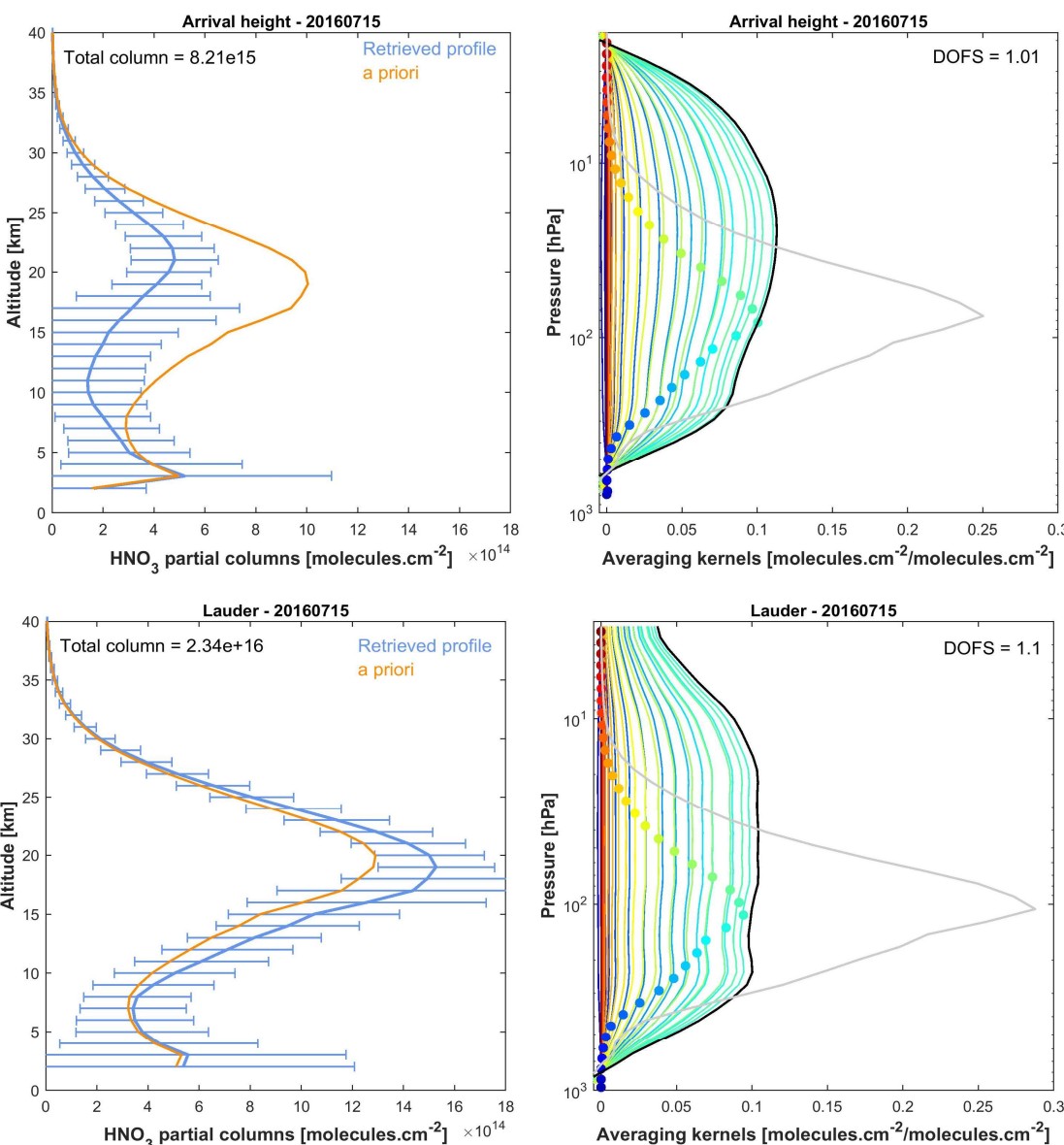

**Figure 1**. Examples of IASI HNO$_3$ vertical profiles (in molec.cm$^{-2}$) with corresponding averaging kernels (in
molec.cm$^{-2}$/molec.cm$^{-2}$; colored lines, with the altitude of each kernel represented by the colored dots)
along with the total column averaging kernels (black) and the sensitivity profiles (grey) (both divided by 10) above
Arrival Heights (77.49°S, 166.39°E, top panels) and Lauder (45.03°S, 169.40°E; bottom panels). The error bars
associated with the HNO$_3$ vertical profile represent the total retrieval error. The a priori profile is also represented.
The total column and the DOFS values are indicated.

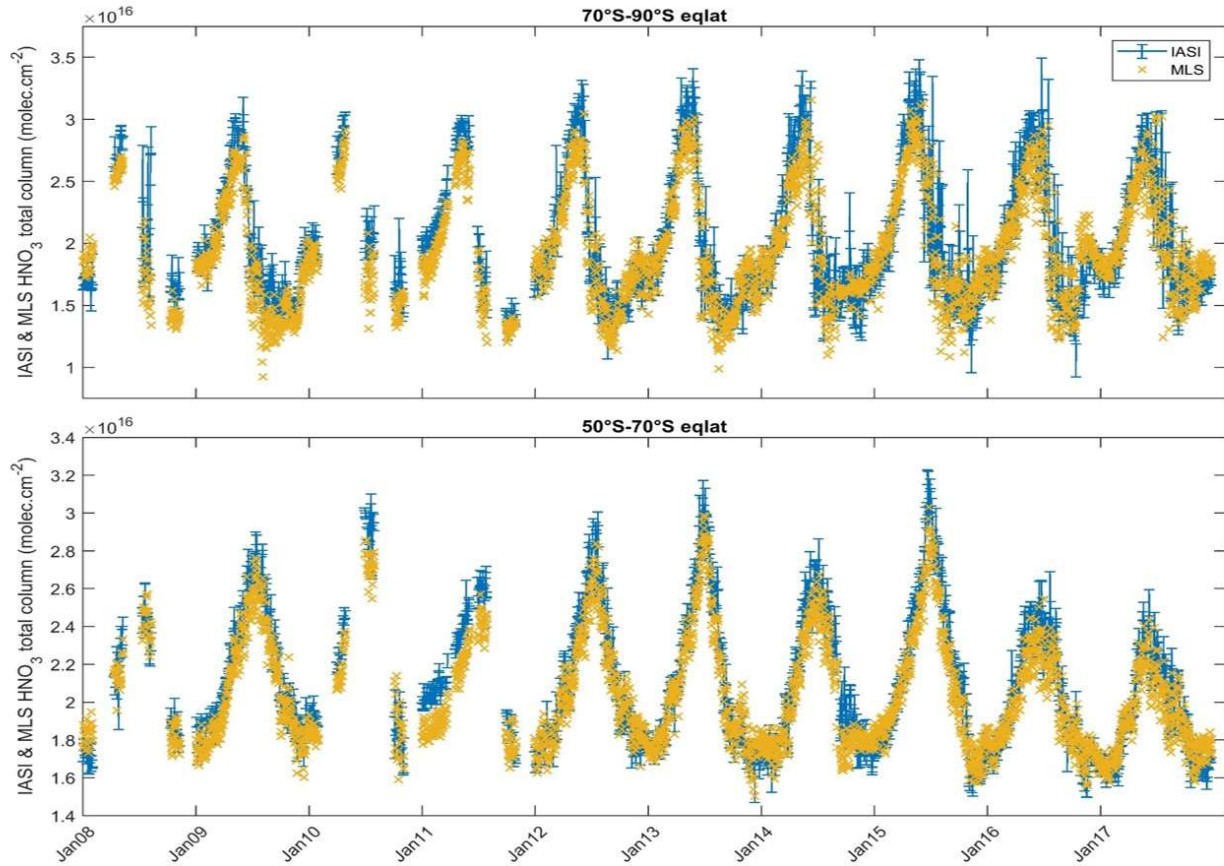

**Figure 2.** Time series of daily IASI total $HNO_3$ column (blue) co-located with MLS and of MLS total $HNO_3$ columns (orange) within 2.5°x2.5° grid boxes, averaged in the 70°S–90°S (top panel) and the 50°S–70°S (bottom panel) equivalent latitude bands. Note that the MLS total column estimates were obtained by extending the MLS partial stratospheric column values using the FORLI-$HNO_3$ a priori information (see text for details). The error bars (blue) represent $3\sigma$, where $\sigma$ is the standard deviation around the IASI $HNO_3$ daily average.

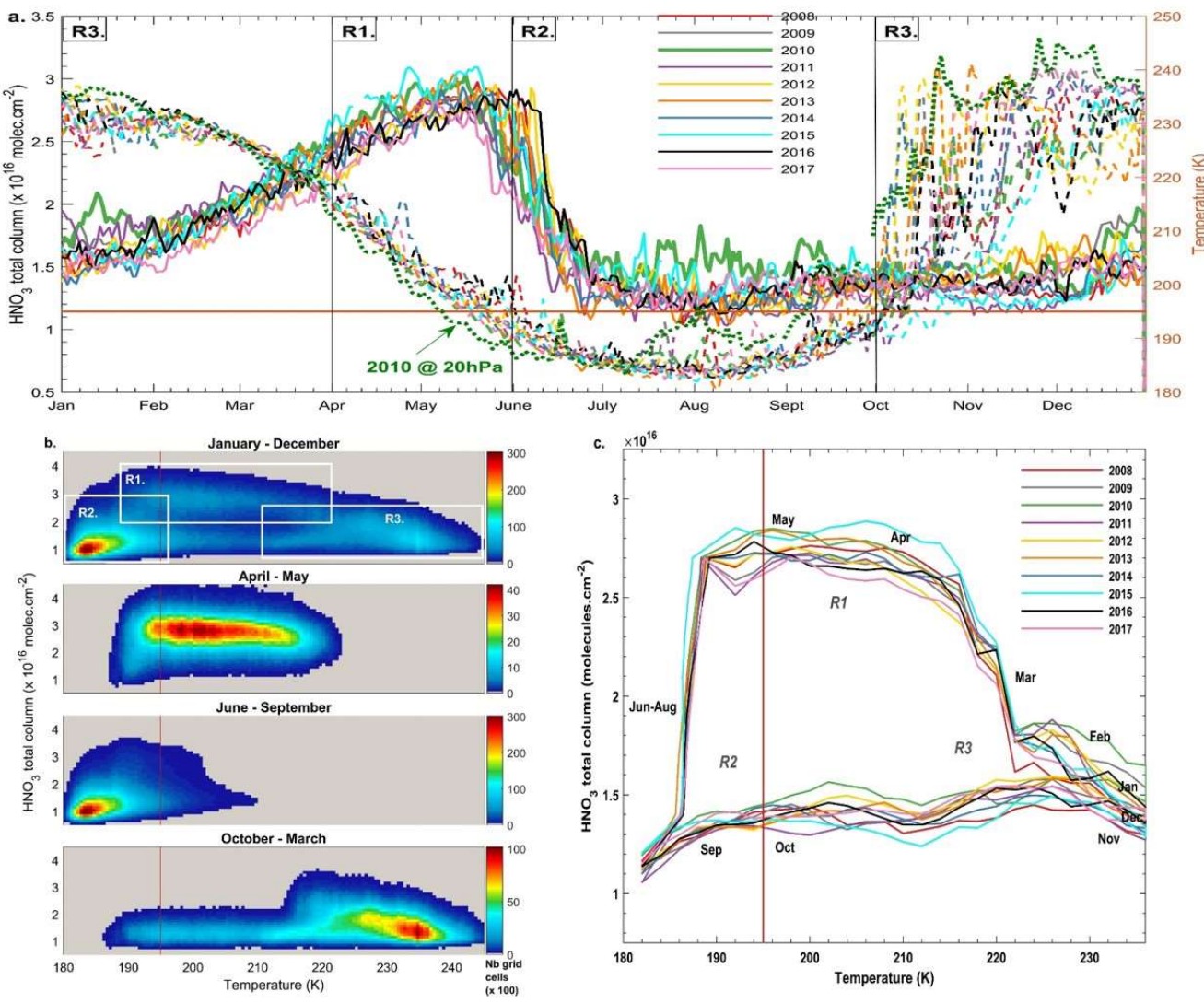

**Figure 3.** (a) Time series of daily averaged $HNO_3$ total columns (solid lines) and temperatures taken at 50 hPa
(dashed lines) in the 70° - 90° S equivalent latitude band, for the years 2008 – 2017. The green dotted line
represents the temperatures at 20 hPa for the year 2010. (b) $HNO_3$ total columns versus temperatures (at 50 hPa)
histogram during the year 2011, over the whole year (top) and for the 3 defined regimes (R1 - R3) separated in
(a). The colors refer to the number of gridded measurements in each cell. (c) Evolution of daily averaged $HNO_3$
total columns with the highest occurrence (in bins of $0.1 \times 10^{16}$ molec.cm$^{-2}$ and 2 K) as a function of the 50 hPa
temperature for the years 2008 – 2017. The orange horizontal or vertical lines represent the 195 K threshold
temperature.

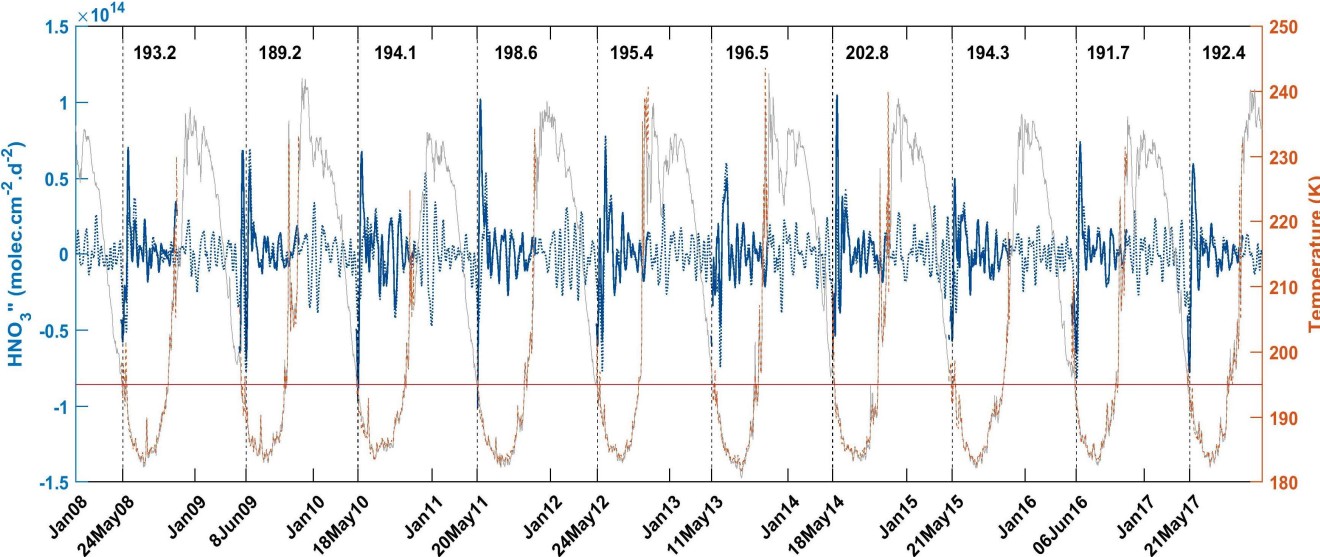

**Figure 4.** Time series of total $HNO_3$ second derivative (blue, left y-axis) and of the 50 hPa temperature (red, right y-axis), in the region of potential vorticity at 530 K lower than $-10\times10^{-5}$ $K.m^2.kg^{-1}.s^{-1}$. The red horizontal line corresponds to the 195 K temperature. The vertical dashed lines indicate the second derivative minimum in $HNO_3$ for each year. The corresponding dates (in bold, on the x-axis) and temperatures are also indicated. The time series of total $HNO_3$ second derivative (dashed blue) and of temperature (grey) in the 70° – 90° S eqlat band are also represented.

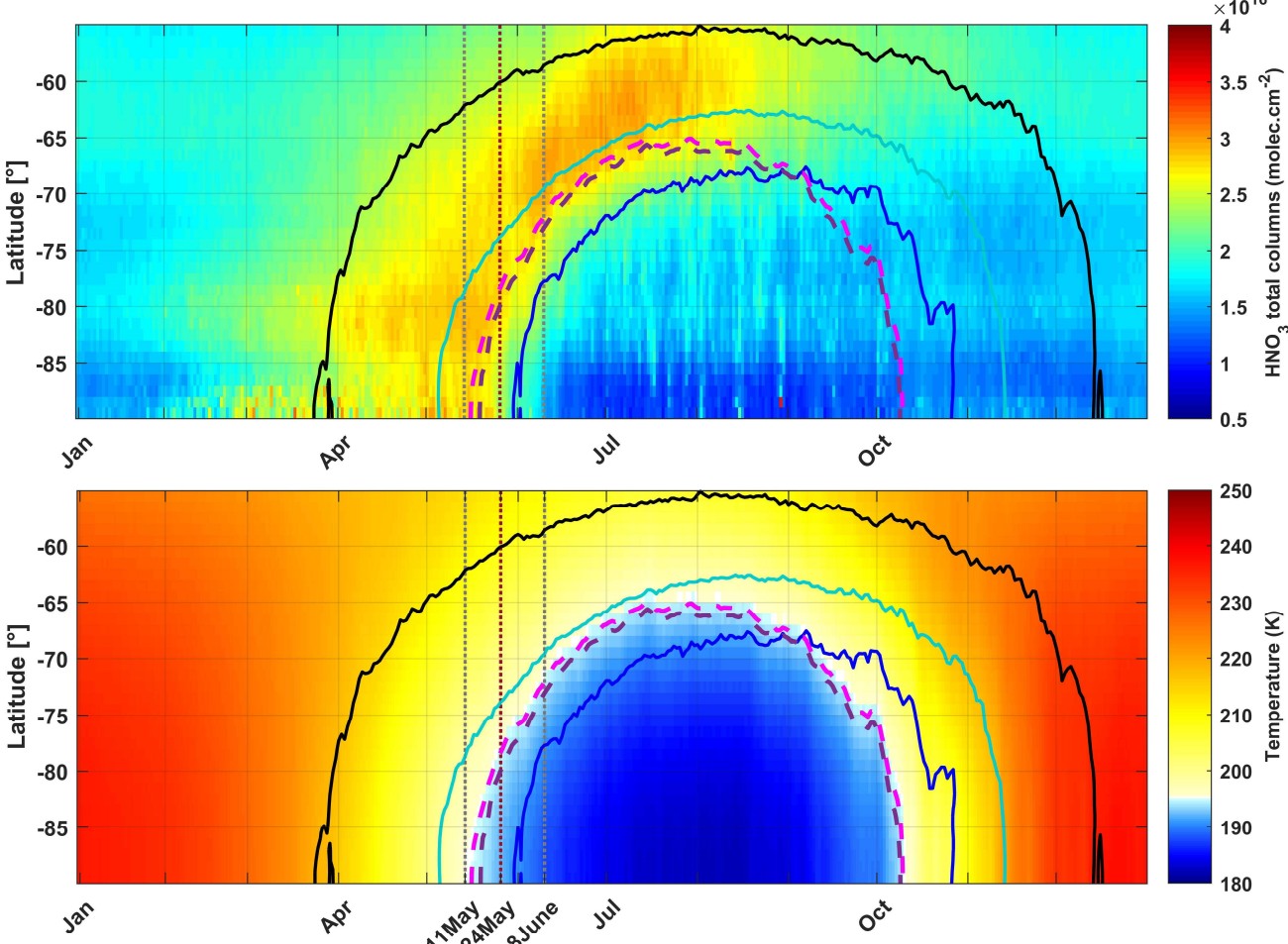

**Figure 5.** Zonal distributions of (a) HNO₃ total columns (in molec.cm⁻²) from IASI and (b) temperatures at 50
hPa from ERA Interim (in K) in the 55° S to 90° S geographical latitude band and averaged over the years 2008
– 2017. Three isocontours for the climatological (2008-2017) and zonally averaged PV of -5 (black), -8 (cyan)
and -10 (blue) (×10⁻⁵ K.m².kg⁻¹.s⁻¹) at 530 K, as well as the isocontours for the 195 K climatological (2008-2017)
zonally averaged temperature (pink) and for the averaged 194.2 K drop temperature (purple) at 50 hPa are
superimposed. The vertical grey dashed lines mark the earliest and latest dates for the averaged drop temperature
in the 10-year IASI record and the red one indicates the average date for the drop temperatures calculated in the
area delimited by the -10×10⁻⁵ K.m².kg⁻¹.s⁻¹ PV contour.

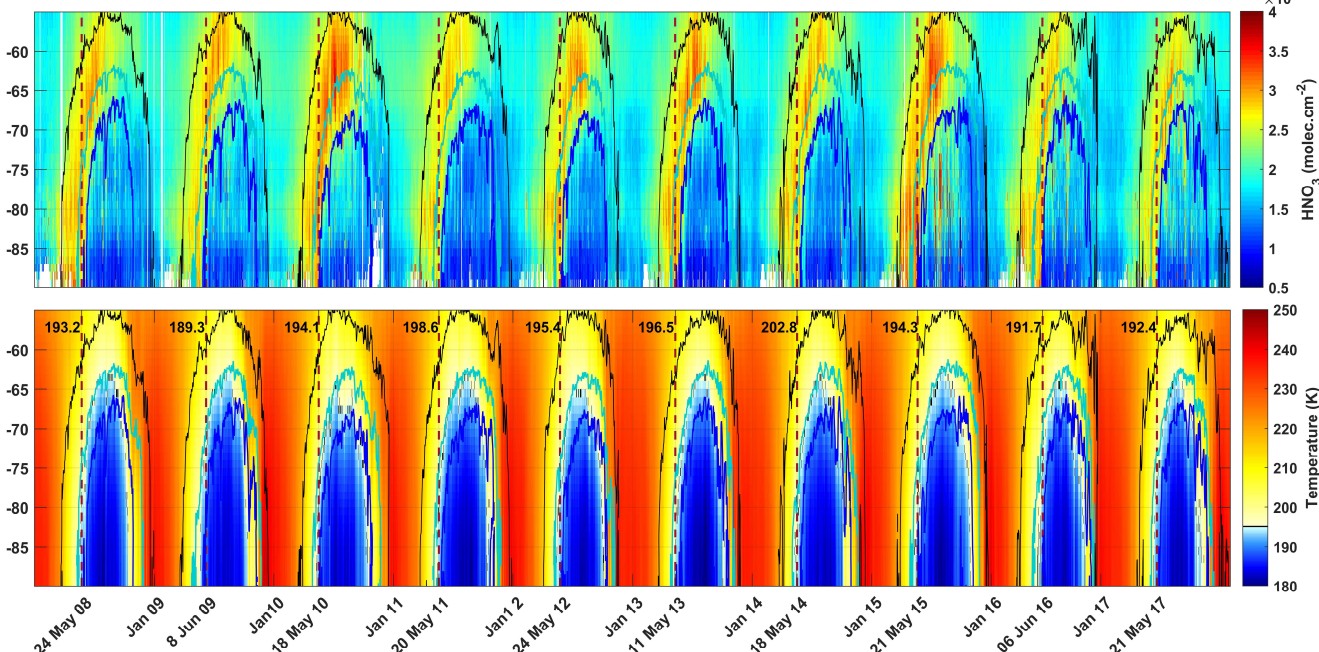

**Figure 6.** Zonally averaged distributions of (top) HNO₃ total columns (in molec.cm⁻²) from IASI and (bottom) temperatures at 50 hPa from ERA Interim (in K). The geographical latitude range is from 55° to 90° south and the isocontours are PVs of -5 (black), -8 (cyan) and -10 (blue) (× 10⁻⁵ K.m².kg⁻¹.s⁻¹ at 530 K). The vertical red dashed lines correspond to the second derivative minima each year in the area delimited by a -10×10⁻⁵ K.m².kg⁻¹.s⁻¹ PV contour.

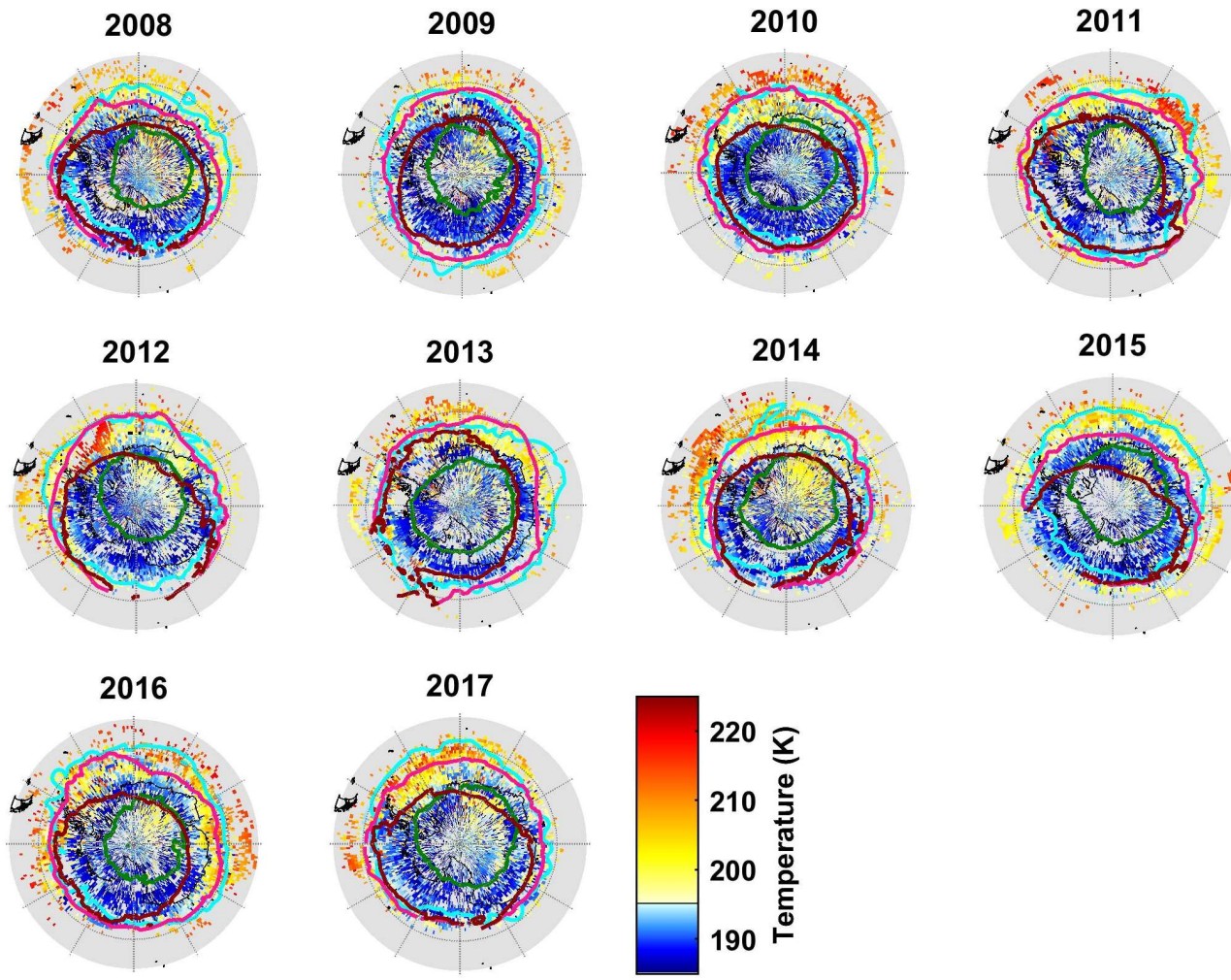

**Figure 7.** Spatial distribution (1°×1°) of the drop temperature at 50 hPa (K) (calculated from the total $HNO_3$
second derivative minima) for each year of IASI (2008–2017), in a region defined by a PV of $-8×10^{-5}$ $K.m^2.kg^{-1}.s^{-1}$. The isocontours of $-10×10^{-5}$ $K.m^2.kg^{-1}.s^{-1}$ at 530 K for the averaged PV (in green) and the minimum PV (in
cyan) encountered over the period 10 May –15 July for each year and the isocontours of 195 K at 50 hPa for the
averaged (in red) and the minimum (in pink) temperatures over the same period are represented.

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
