# Peer review of "Polar stratospheric nitric acid depletion surveyed from a decadal dataset of IASI total columns"

_Atmospheric Chemistry and Physics, 2020_

## Referee Comment (RC1) · Anonymous Referee #1 · 31 Aug 2020

In this manuscript, Ronsmans et al. present vertical column amounts of nitric acid (HNO3) as derived from IASI observations over the Southern Hemisphere between 2008 and 2017. These are analysed in comparison with temperatures at 50 hPa to characterize the depletion of gas-phase HNO3 in the wintertime Antarctic polar vortex by uptake in polar stratospheric clouds (PSCs). As a measure for the onset of HNO3 depletion, the so-called 'drop-temperature', as defined by the minimum curvature of the HNO3 column amounts against time, is introduced.

After the foreseeable end of limb-observations in the microwave and thermal infrared spectral region, measurements from operational weather satellites by nadir sounding spectrometers will be the only possibility to inform about trace gases, like HNO3, which are important to describe the state of the stratosphere in the midst of intensifying cli-

mate change. Though not being able to derive vertical profiles, it is a least possible to derive total column amounts of HNO3 from the IASI instruments on the operational Metop weather satellites.

The major part of the data (2008-2016) reported in this manuscript was already published in Ronsmans et al. (2018), also together with temperatures at 50 hPa. For example, Fig. 4 (top) of the actual manuscript is a zoom of Fig. 3 of Ronsmans et al. (2018) to the southern latitudes with one Antarctic winter added. To derive reliable conclusions from these measurements, an in-depth characterization of these datasets on HNO3 is indispensable. While Ronsmans et al. (2016) provide a first validation of the observations by comparison with FTIR solar absorption measurements, a characterization given the extreme conditions within the dark Antarctic polar vortex is missing. This is one of the majors concerns why I think the paper should not be published in ACP in its present form. However, it should be quite straightforward to provide at least a first comparison with HNO3 observations by the Microwave Limb Sounder (MLS) which has a large temporal and spatial overlap with the IASI dataset.

Specific comments:

L3, 'good vertical sensitivity':

This has not been shown in this paper. It is necessary to demonstrate this for the dataset discussed here given the cold Antarctic stratosphere.

L8, 'denitrification':

Are you certain, that 'denitrification' is also used for the uptake of HNO3 in particles? Perhaps 'removal from the gasphase'.

L59, 'a maximum sensitivity in the mid-stratosphere around 50 hPa':

This must be shown here for the extreme conditions in the Antarctic vortex - also since all later analyses in the paper use temperatures at 50 hPa. What is the vertical variability of this level of maximum sensitivity within the development inside the vortex,

especially later in the winter when, due to sedimentation of PSC particles, HNO3 concentrations at those levels are very low?

L79, 'The total columns yield a total retrieval error of 10% and a low bias (10.5%) compared to ground-based FTIR measurements (Hurtmans et al., 2012; Ronsmans et al., 2016).':

As these numbers are used also later in the manuscript, their validity has to be confirmed for the condition in the dark vortex, which cannot be achieved with comparisons to sun-dependent FTIR observations. As mentioned above, I strongly suggest to perform comparisons with the MLS dataset.

L105, 'These high HNO3 levels result from low sunlight,...':

This is not the only, and probably not the central explanation for the increasing column amounts. Dynamical effects on total columns of stratospheric gases (downwelling within the vortex) have to be considered.

Figure 2:

I think the vertical dashed line '10Jun09' does not fit to the minimum of the solid blue curve (?)

L154, 'in the areas of potential vorticity smaller than $-10$...'

PV at which potential temperature level is used here?

L159, 'Note that the HNO3 time series has been smoothed':

As the drop temperatures (and dates) are introduced as the central new method presented in the manuscript, it is necessary to explore their behaviour in more detail. Can you give an estimate of the error of this measure by considering e.g. the effect of the numerical smoothing. Please show also the 1st derivative to be able to judge on the uncertainties of the 2nd derivative. How do the drop temperatures vary when using different pressure levels (e.g. 70 hPa)?

L184, 'The calculated drop temperatures vary significantly between ∼180 and ∼210 K. These high extremes are only found in very few cases and should be considered with caution as they correspond to specific regions above ice shelves with emissivity features that are known to yield errors in the IASI retrievals':

I find the discussion around the deviations of the drop temperatures very confusing. At the beginning of the manuscript it is stated, that the error of the measured total column amounts is in the order of 10%. Here it is argued that 'above ice shelves' it might be higher. Also, in Fig. 5 one can see that there are large regions over eastern Antarctica where drop temperatures are often clearly above 195K even inside the red circles. This is not explained satisfactorily in the manuscript. Here, again, it would be important to investigate on the reliability, consistency and homogeneity of the IASI HNO3 values. As mentioned above, this could be accomplished with a comparison to MLS observations.

L195, 'Overall, despite these limitations, the spatial variability in the drop 50 hPa temperatures for IASI total HNO3 is well in agreement with the natural variation in PSCs nucleation temperatures':

Given the extended areas where the drop temperatures are larger than 195K, this statement is not convincing.

L204, 'denitrification phase':

See statement about 'denitrification' above.

L230, 'To the best of our knowledge, it is the first time that such a large satellite observational data set of stratospheric HNO3 concentrations is exploited to monitor the evolution HNO3 versus temperatures.'

This sounds somehow exaggerated given all the previous work on HNO3/temperature/PSCs, e.g. by use of the MLS dataset and also since the correlation with temperature has already been shown in Ronsmans et al., 2018.

Technical comments:

[Figure]

L27, '(e.g. (Toon...))':

I think the inner bracket level is not necessary.

L30, 'sedimentation(Lambert...':

Space missing

L34, 'temperature':

'temperatures'

L51:

Bracket levels?

L102, 'The red vertical line in Fig. 1a and Fig. 1b':

There is no vertical red line in Fig. 1a. You mean horizontal?

L106, references:

Brackets seem wrong.

Figure 2, caption, 'in the70—':

Space missing.

L155, 'and the total HNO3 depletion are the coldest':

Makes no sense.

L164, 'temperature are'

'temperatures are'

References

Ronsmans, G., Wespes, C., Hurtmans, D., Clerbaux, C., and Coheur, P.-F.: Spatio-temporal variations of nitric acid total columns from 9 years of IASI measurements –

a driver study, Atmos. Chem. Phys., 18, 4403–4423, https://doi.org/10.5194/acp-18-4403-2018, 2018.

Ronsmans, G., Langerock, B., Wespes, C., Hannigan, J. W., Hase, F., Kerzenmacher, T., Mahieu, E., Schneider, M., Smale, D., Hurtmans, D., Mazière, M. de, Clerbaux, C., and Coheur, P.-F.: First characterization and validation of FORLI-HNO 3 vertical profiles retrieved from IASI/Metop, Atmos. Meas. Tech., 9, 4783–4801, https://doi.org/10.5194/amt-9-4783-2016, 2016.

---

## Referee Comment (RC2) · Anonymous Referee #3 · 31 Aug 2020

The paper presents an analysis of a 10-year IASI column $HNO_3$ record (2008-2017) for the Southern Hemisphere in conjunction with ERA-I reanalysis temperatures. The sequestration of $HNO_3$ into PSCs is extremely temperature sensitive and it's unlikely that a single temperature at 50hPa is sufficent to capture anything but the most basic features. The description of the polar $HNO_3$ variation presented in the paper is already well known from numerous other studies. The lack of vertical resolution in the IASI $HNO_3$ measurements severely limits the interpretation of the results and precludes differentiation between denitrification and renitrification e.g. consider the effect of the vertical integration through depleted higher layers overlaying lower enhanced layers. Although the IASI $HNO_3$ data has much better 2D horizontal resolution than any other measurement this has not been developed as a tool to provide information beyond that

of satellite instruments that measure only along the orbit track. CALIOP PSC information is available for the same time frame, why was this not used? Certainly, PSC volumes vs time would be helpful in providing the underlying interannual variability of PSC types (NAT, STS, ice) to compare with the resulting drop temperatures derived from IASI. Similarly, at least some comparisions of the IASI HNO3 column with integrated column calculated from Aura MLS are necesssary to establish the validity of the measurements in the most severely depleted inner vortex core.

Regarding the sensitivity of the IASI column HNO3 measurements, I suggest presenting a few examples of vertical HNO3 profiles (from a model or data), ranging from non-depleted to extreme depletion with calculations of the corresponding calculated integrated IASI column. This would help to indicate the sensitivity of the column measurement to changes in the vertical distribution of HNO3 ... i.e. generate profiles of the change in the IASI column HNO3 wrt the actual change in HNO3 at a level, j, ... d(column)/d(HNO3)j.

I do not recommend publishing this paper in ACP without attention to the points raised in this review.

Specific comments:

L2: "good vertical sensitivity" ... only column HNO3 measurements are discussed here - there is no vertical resolution in the measurements.

L8: in [the] Antarctic

L10: 191K is also consistent with STS temperatures (192 K) and is actually closer than TNAT (195 K)

L18: add more recent references e.g. Peter and Gross (2012)

L28: Much more has been done in the past decade with MIPAS and CALIOP that should be referenced

L53: Studies of HNO3 depletion and PSC formation predate the sensors named in the paragraph e.g. the Santee et al (1999) reference used UARS/MLS launched in 1991, measurement using balloons should have been be referenced here.

L59: This section should explain what is meant by "maximum sensitivity" etc.

L79: Information on the data quality for IASI HNO3 is poor. Is the value of bias and uncertainty the same for depleted and non-depleted conditions?

L82: Yet, problems with the retrievals because of cloud contamination seem to remain even after the <25% cloud fraction filter is applied.

L83: Cloud contamination? Tropospheric cloud only or also thick ice PSCs?

L102: Why was 2011 chosen?

L106: Heterogeneous hydrolysis of N2O5 requires aerosol particles. So this process starts with cold binary aerosols (i.e. sulfates) before the formation of STS?

L108: extends

Figure 1 caption: Each figure title in 1(b) needs to state the year e.g. "January - December 2011 or put a label "2011" above the whole figure.

Figure 1 caption: 50 hpa => 50 hPa

Figure 1 caption: it is not clear to what 0.1E16 molec. cm-2. This low value is not even on the y-axis of the figures.

Figures 1(a) and 1(c): Are the HNO3 and temperature structures (localized peaks and valleys) visible in the time series in 1(a) quite well correlated when plotted as a scatter diagram as in 1(c), but without the 7-day averaging?

L123: 7-day

L124 and Figure 1 caption: "in the range of" : only one value is given and not a range of values

L129: The onset of depletion seems to start when the temperatures fall substantially below 190K from inspection of Fig 1(c) and quite far below the red line marked at 195K.

L130: Supplementary material - this does not appear to be available from the ACP website.

L136-137: Why are two temperatures (180 and 185 K) quoted for 30hPa? Why is the actual value from Fig1(c) (I estimate this as about 188K) for the 50hPa temperature not given in L129?

L138: "characterized by" seems the wrong description for the chance occurrence that the maximum sensitivity of IASI HNO3 falls in the same altitude range as the PSCs.

L139-146: This section rather seems to belong in the conclusions.

L148:Clearly this does not "go beyond the vertically integrated view" since the column HNO3 is all that is available. It could be reworded as "To identify the spatial and temporal variability of the column HNO3 ..."

L164: drop temperatures

L165-169: Denitrification is the term used to describe the permanent removal of some HNO3 from the gas phase by sedimentation of PSCs. Sequestration is the term used to describe the uptake of HNO3 from the gas phase into PSCs. Denitrification by STS is a lengthy process compared to NAT since the smaller STS particles sediment slowly. STS can (and frequently does) form without the prior nucleation of NAT. IASI alone cannot discriminate between these processes and it should not be assumed that what is observed is the "onset of HNO3 denitrification".

Figure 3 caption: sumperimposed => superimposed

L170: Figures 3a and b

L171: three isocontour levels

L174: lines indicate

L185-187: 210K is much too high for PSC formation, but could possibly be NAT that is in process of melting? If these are observed over ocean then they warrant further investigation. However, why are specific regions with emissivity features not flagged as such? They should be discarded rather than "used with caution".

L189: Modern reanalysis temperatures (e.g. ERA-I) do not "feature large uncertainties" large enough to account for a 195K to 210K shift.

L195-L201: The limitations of the reanalysis temperatures seems to be an accuracy of better than 1K and clearly this in no way limits the derivation of the "50hPa drop temperature" which simply necessitates finding the 50hPa reanalysis temperature that corresponds to the second derivative wrt time minimum in column HNO3.

What is meant by "spatial variability"? The plots in Fig 5 show the spatial distribution of the drop temperature over a number of years but what variability is being considered? Interannual? Why have these spatial maps of drop temperatures not been compared with published maps of PSC types made by CALIOP or MIPAS. Wouldn't some correlation be expected according to the arguments made here? i.e. NAT PSCs at the higher temperature e.g. the highest temperatures (orange) appear downstream of the Palmar Peninsula in the "NAT ring" structure described by Hopfner et al (2006).

L200: It underlines ... What does "it" refer to? The subject of the previous sentence is "the spatial variability" but that has not been defined.

L201:critical denitrification phase

L205: Nothing has been presented that demonstrates PSC occurrence. For that you would need to compare to actual data on PSCs from CALIOP and/or MIPAS.

L205: to PSCs occurrence to PSCs ??

L224: Again, the suspect data should be discarded because of the detrimental impact

on the scientific analysis. Also, if you cannot manage to work out and apply adequate quality control to your own data then you have no reason to expect anyone else to do so.

L230: "To the best of our knowledge, it is the first time that such a large satellite observational data set of stratospheric HNO3 concentrations is exploited to monitor the evolution HNO3 versus temperatures"

In fact you cite several papers that have done exactly this, but let's take the one published over two decades ago by Santee et al (1999) titled "Six years of UARS Microwave Limb Sounder HNO3 observations : Seasonal, interhemispheric, and interannual variations in the lower stratosphere". https://doi.org/10.1029/1998JD100089

Not only does this paper compare HNO3 with UKMO temperatures we are referred to a more complete paper on this topic on p8241 ... "The correlation of the HNO3 behavior with temperature during this time period, and its implications for PSC phase and composition, is explored in detail by Santee et al (1998).

I noticed that the outside edge of the "HNO3 collar region" at 465K was defined by these authors as inside the 0.25 x E-4 K m2 kg-1 s-1 PV contour. This seems at odds with the 1E-4 value that is used for the second derivative minimum calculation in this paper and seemingly places the boundary quite far equatorward.

Santee et al (1998) also includes a description of the heterogeneous hydration of N2O5 that would be helpful in response to the question above on L106.

L231: "It could constitute a new accurate climatological parameter that could be inserted in the PSCs classification schemes."

The analysis presented does not support this statement. Specifically, how could the HNO3 column amount be used in a classification scheme?

L240: "All authors contributed to the writting of the text and reviewed the manuscript."

[Figure]

writting => writing

---

## Short Comment (SC1) · 28 Sep 2020

**Comment on "A nitric acid dataset from IASI for polar stratospheric denitrification studies"**
**by Ronsmans et al.**

This manuscript analyzes 10 years of IASI $HNO_3$ total column measurements in conjunction with ERA-Interim temperatures to characterize the onset of PSC formation in the Antarctic lower stratospheric vortex. The high-density horizontal sampling afforded by IASI is valuable, and the approach of using the minimum in the second derivative of the $HNO_3$ total column with respect to time to identify the onset of $HNO_3$ uptake into PSCs is interesting and potentially useful. However, several aspects of the analysis and/or its description in the manuscript are flawed. Although referee comments raising serious issues with this manuscript have been posted, there are a number of points that we would like to add and/or further elaborate.

Respectfully,
Gloria Manney and Michelle Santee

General comment: Throughout this manuscript, starting with its title, the term "denitrification" is taken to be synonymous with the uptake of gas-phase $HNO_3$ through the formation of PSCs. Although not without precedent, this approach is contrary to common practice and may lead to confusion. Condensation of $HNO_3$ in PSCs is usually referred to as "sequestration", while the term "denitrification" is usually reserved for the permanent removal of $HNO_3$ from the lower stratosphere through the sedimentation of PSCs. In the absence of analysis of direct PSC measurements (e.g., from an instrument such as CALIOP), the occurrence of true denitrification can only be inferred from space-borne measurements of gaseous $HNO_3$ when abundances do not rebound as PSCs dissipate at the end of winter, suggesting permanent removal. Thus the "drop temperature" derived in this study is indicative only of the onset of PSC formation, not the onset of denitrification, as is stated in numerous places in the paper.

Abstract
- L2: It is misleading (particularly for those who read only the abstract of the paper) to characterize the IASI $HNO_3$ total columns as having "good vertical sensitivity". Indeed, this optimistic assessment is directly contradicted in Section 2, where IASI is stated to have "low vertical sensitivity ... with only one independent piece of information" (L76).

Introduction
- L48-49: It should be made more clear that this is by no means an exhaustive list of spaceborne instruments that have measured stratospheric $HNO_3$.

Section 2
- The information provided about the IASI $HNO_3$ retrieval, data quality, and data screening is insufficient. This information is critical to assessing the robustness of the reported results, and readers should not be forced to refer to previous papers to find it.
- In later sections (e.g., L186, L225), errors in IASI retrievals arising from issues with emissivity above ice shelves are invoked to account for some dubious results, but no

mention of these poor-quality retrievals is made in the "Data" section, nor is it explained why quality-control measures fail to properly filter out these suspect data points.

- L78: 10 km can hardly be characterized as the "mid-stratosphere".
- L84: "normal" has a specific statistical meaning and is not the appropriate word here.
- L85-86: The validity of the analysis approach depends on the 50 hPa pressure surface and the 530 K isentropic surface being in very close proximity during Antarctic winter. This implicit assumption should be explicitly justified in the paper.
- L89-91: It is highly problematic to use a single theta level to distinguish inside from outside vortex regions for column measurements. This approach implicitly (and erroneously) assumes that the vortex does not tilt, shrink, or expand with height over the altitude range considered. A better approach would have been to check PV over a range of levels and discard measurements classified as outside the vortex at any one of those levels. A similar comment can be made concerning the use of a single pressure level for temperature. Again, it might have been better to use a range of T over the ~10–30 km layer where IASI has most sensitivity. Some attempt is made to justify the latter choice (using 195 K at 50 hPa) in Section 3 (L141-142) and Section 4 (L168-169), but the arguments are not convincing, as the authors themselves appear to recognize when they state (L188-189) "hence, the use of temperature at a single pressure level might be restrictive to some extent".

Section 3

- The definition of the three "regimes" in the $T/HNO_3$ relationship seems arbitrary and not well justified. For example, R1 is defined to begin in April, but Fig. 1a shows that $HNO_3$ values start to increase rapidly and temperatures start to decrease rapidly in March (or even February, as noted in L117), not April. Only R2 encompasses a steep change in $HNO_3$, but that regime also includes a lengthy period during which $HNO_3$ remains nearly constant. It might have been better to break R2 into an "onset of PSC formation" phase and a "denitrification plateau" phase. Moreover, as defined in the paper, R2 extends through, not to, September as stated in L108. These problems are evident in the discussion in this section, as in some cases the behavior ascribed to one regime actually occurs in another.
- L102 and Fig. 1 caption: The red line in Fig. 1a is horizontal, not vertical, and Fig. 1b contains no such line – it is on Fig. 1c. Neither red line is defined in the caption.
- L102 and Fig. 1: 2011 was a particularly cold and long-lasting Antarctic winter, and thus it is arguably not representative. Some explanation for why that year was selected for highlighting in Fig. 1b is needed.
- L105-106: The contribution of confined descent inside the developing vortex bringing air rich in $HNO_3$ from above into the domain where IASI is most sensitive has been ignored here – isn't descent also a factor leading to the observed high $HNO_3$ total column values in early austral autumn?
- L115-116: In addition to a lack of citations of earlier papers on renitrification of the lowermost stratosphere (LMS), this sentence is not a very clear expression of the fact that IASI is not sensitive to the LMS and hence renitrification has little impact on the observed evolution of total column $HNO_3$.

- L119-121: Why is 2010 highlighted in Fig. 1a (green line)?  Other recent Antarctic winters were also disturbed with some minor SSW activity, e.g., 2012 and 2013.  Did those episodes not affect the $HNO_3$ distribution?  Also, why does the green line show T at 20 hPa, when the other curves show T at 50 hPa?  More explanation for why the authors chose to show this particular level for this particular year is needed.
- Fig. 1c: In general this plot is not well explained or well motivated.  By showing the position in temperature / $HNO_3$ space of the bin with the maximum number of observations, important information about the range of those values on a given day is omitted.  The ranges in Fig. 1b suggest that the values at a given time may span most of the $HNO_3$ axis in Fig. 1c, rendering the curves shown less meaningful.  In addition, it is stated (L127) that this figure highlights the interannual variability in total $HNO_3$, but interannual variability is also clearly seen in panel (a), which is much easier to interpret.  The discussion relates the picture in Fig. 1c to the three regimes, but since they are not marked on this panel, it cannot easily be examined without reference to Fig. 1a.  It is therefore not obvious what additional value this figure brings to the paper.
- L125: $HNO_3$ columns are said to slowly increase as the T decreases over "February to May, i.e., R3 to R1".  However, R3 is defined to start in October, and actually the slow increase in total $HNO_3$ starts before February, arguably even as early as December.
- L126: In the discussion of strong and rapid HNO3 depletion, "June (R1-R2)" should be "June-August (R2)".

Section 4
- Fig. 2 and its caption: More should be said about the agreement (or lack thereof) between the dashed and solid $HNO_3$ and the grey and red T lines when they both exist.  Some readers may question why the PV approach is used, given the gaps in those curves.  Also, perhaps this is just an optical illusion, but the solid blue line appears to be thicker in some years (2011, 2014, 2016, 2017) than in the others.  If that is the case, then that also needs to be explained.  In the caption, the level to which the stated PV value pertains (presumably 530 K) should be specified.
- L155: It is not appropriate to characterize the total $HNO_3$ depletion in the inner vortex as being the "coldest".
- L160: The wording in this sentence is garbled.
- L162-163: 23 is more than "a few" days.
- L174-179 and Fig. 3 caption: The description of the figure is confusing.  It is stated in both in L174-175 and the caption that the vertical red dashed line indicates, at 90S, the 10-year average of the drop temperatures (191.1 K) calculated from the $HNO_3$ second derivative time series in the area delimited by the $-10 \times 10^{-6}$ $K.m2.kg^{-1}.s^{-1}$ PV contour.  It's not clear how a vertical line on a time series plot can represent a temperature value.  Perhaps the authors meant to say the average date on which T dropped below the 195 K threshold at 90S?  Moreover, the discussion above indicated that the value of 191.1 K was the average for the inner vortex (defined by either PV or EqL), not specifically at the South Pole (90S).  In addition, the scale for the PV contour should be $10^{-5}$, not $10^{-6}$.  Then in L176-177, it is stated that the "delay of 4-23 days between the maximum in total

HNO$_3$ and the start of the depletion is also visible" – but how is a range of values (which arises from different years) visible in a climatological plot?

- Fig. 4: Very little discussion is devoted to this figure; it is merely noted (L177-178) that it shows the reproducibility of the IASI measurements of HNO$_3$ depletion from year to year.  Since Fig. 1 already makes this point, the added value of Fig. 4 is not clear.
- Fig. 5: How relevant is the PV contour averaged over the May to October period, when the dates of the onset of HNO$_3$ depletion are May to June (or possibly July)?  Why include August, September, and October in this average?
- L181: "the drop 50 hPa temperatures" should be "the 50 hPa drop temperatures".
- L183: Technically, the isocontour represents –10, not ≤ –10.
- L184-185: First, how does the range of dates corresponding to the onset of HNO$_3$ depletion reported here – mid-May to early July – relate to that reported (L163) in connection with Fig. 2, which was 17 May to 10 June?  Does the difference in these estimates arise because the former is based on averages in 1°×1° bins, whereas the latter is based on a vortex average within the PV contour?  July seems rather late for the onset of PSC formation.  Similarly, the range in 50 hPa drop T is quoted as 188.2 K to 196.6 K in L164, whereas here drop Ts vary over a wider range, from 180 to 210 K.  The values at both extremes of this range are unrealistic.  Indeed, the date and T ranges found in connection with Fig. 5 call into question the analysis method.
- L189-196: The questionable results derived from this analysis cannot be pinned on biases in the ERA-Interim data.  The statement is made that "Reanalysis data sets are, indeed, known to feature large uncertainties", but the uncertainty in modern reanalysis temperatures (typically less than ~1 K) is by no means large enough to account for drop Ts as extreme as 180 and 210 K.  The reliability of reanalysis temperatures in the polar lower stratosphere (including those from ERA-Interim) has been conclusively demonstrated in several recent papers, notably by Lawrence et al. [2018] and Lambert and Santee [2018].  Although both papers are cited here, their implications have apparently been overlooked.
- L197-199: This sentence is confusing and its intended meaning is unclear.  It appears to be comparing apples (the spatial variability in drop T seen in the maps in Fig. 5) to oranges ("natural" variations in PSC nucleation T, TTE, and PSC formation mechanism).  Perhaps the authors meant the spatial variability in those parameters (and not the values themselves), but that is not how the sentence is constructed.  In any case, further discussion of comparisons of Fig. 5 with previously published results is warranted.
- L199-200: A number of other satellite data sets have captured gas-phase HNO$_3$ depletion (from both sequestration and denitrification) on similarly large scales.

Conclusions
- L225-226: It is stated that "the range of drop temperatures is interestingly found in line with the PSCs nucleation temperature that is known, from previous studies, to strongly depend on a series a factors".  In fact, the derived range (180–210 K) is so large that it is arguably not in line with previous work, and it is therefore difficult to see how the IASI total column HNO$_3$ measurements provide added value (as stated in L203) to studies of

Antarctic PSC formation and the interannual variability therein beyond that obtained from other satellite $HNO_3$ datasets.

- L230-231: The statement that this paper represents "the first time that such a large satellite observational data set of stratospheric $HNO_3$ concentrations is exploited to monitor the evolution $HNO_3$ versus temperatures" is wholly unsupportable. In fact, there is a substantial body of literature on the relationship between $HNO_3$ and temperature, including studies of long-term vertically resolved datasets. In particular, Lambert et al. [2016] (which is cited in a number of places in this manuscript, but only in passing) examined 10 years of Aura MLS $HNO_3$ in the Antarctic winter vortex and its relationship to T – including temperature history (a factor that has been largely ignored here) and T with respect to $T_{ICE}$ – as well as PSC composition as determined by CALIOP. In general, discussion of how the current results fit into the context of the findings from Lambert et al. [2016] and other relevant prior studies is inadequate.
- L233-234: More explanation of how $HNO_3$ total column amounts could be used to inform PSC classification schemes is needed to justify this statement, especially given how spatially heterogeneous and layered PSCs have been shown to be.

Finally, in addition to the serious substantive issues enumerated above and in the formal reviews of the official referees, the manuscript suffers from the poor quality of the writing. If this paper were to be eventually accepted for publication, it would require extensive copy-editing to improve the English.

---

## Author Comment (AC1) · 15 Dec 2020

**Response to reviewer #1**

We thank the reviewer for her/his in depth review comments that have help us to improve the clarity of the manuscript. Kindly find below our responses to each of the comments (quoted between []). We hope that our responses will address the main issues and that the changes made will convince that the IASI $HNO_3$ dataset has the potential to contribute to stratospheric studies and, more particularly, to the time evolution of the polar processes.

**Major comments**

[The major part of the data (2008-2016) reported in this manuscript was already published in Ronsmans et al. (2018), also together with temperatures at 50 hPa. For example, Fig. 4 (top) of the actual manuscript is a zoom of Fig. 3 of Ronsmans et al. (2018) to the southern latitudes with one Antarctic winter added.] This paper indeed builds on the study of Ronsmans et al. (2018) but it goes a step further in showing the potential of the IASI-$HNO_3$ dataset for polar stratospheric studies, which was not detailed in Ronsmans et al. (2018). If MLS allows resolving the $HNO_3$ profile between 11 km and 30 km, the potential of IASI lies in its exceptional spatial and temporal sampling. We demonstrate here that despite its limited vertical sensitivity forcing us to consider one total column, the information content that lies in the low-middle stratosphere is good enough to expand on polar stratospheric denitrification studies, usually performed using limb sounder measurements, and to continue their long-term record given the end of limb-observations in the microwave and thermal infrared spectral region.

[While Ronsmans et al. (2016) provide a first validation of the observations by comparison with FTIR solar absorption measurements, a characterization given the extreme conditions within the dark Antarctic polar vortex is missing. This is one of the majors concerns why I think the paper should not be published in ACP in its present form. However, it should be quite straightforward to provide at least a first comparison with HNO3 observations by the Microwave Limb Sounder (MLS) which has a large temporal and spatial overlap with the IASI dataset.] The referee is kindly invited to refer to the figure 3 (top and bottom panels) of Ronsmans et al. (2016) that presents the global distributions of the degrees of freedom for signal (DOFS, top panels) and of the altitude of maximum sensitivity of IASI to the $HNO_3$ profile, separately for January (left) and July (right) 2011, when the strong $HNO_3$ depletion occurs within the cold Antarctic winter. Figure 3 of Ronsmans et al. (2016) clearly shows DOFS of around 0.95-1.05 inside the Antarctic polar vortex, demonstrating the ability of IASI to measure a total column of $HNO_3$ even above these coldest regions. It is also worth to note here that the measurements characterized by a low vertical resolution (DOFS < 0.9) or a poor spectral fit have been filtered out of this analysis. This is now better mentioned in Section 2 of the revised manuscript:

"Quality flags similar to those developed for $O_3$ in previous IASI studies (Wespes et al., 2017) were applied a posteriori to exclude data (i) with a corresponding poor spectral fit (e.g. based on quality flags rejecting biased or sloped residuals, fits with maximum number of iteration exceeded), (ii) with less reliability (e.g. based on quality flags rejecting suspect averaging kernels, data with less sensitivity characterized by a DOFS lower than 0.9) or (iii) with cloud contamination (defined by a fractional cloud cover larger than 25 %)."

Despite the fact that a validation of the IASI measurements within the Antarctic polar vortex against ground-based FTIR measurements could not be provided (these observations requiring sunlight), we agree that an evaluation of the IASI measurements in the Antarctic night was missing. Hence, as

suggested by the referee, we have preformed cross-comparison with observations by MLS in three equivalent latitude bands (see Figure 1 here below). We would like to point out that we here compare total columns measured by IASI with VMR measured by MLS at several pressure levels that cover the highest sensitivity of IASI (at ~50 hPa, ~70 hPa and ~30 hPa for the sake of the comparison). Hence, the comparison of IASI columns with MLS measurements is mostly qualitative at this stage and differences are expected for this reason. Note also that we have preferred comparing IASI $HNO_3$ columns with VMR measured by MLS at specific levels instead of integrated columns calculated from MLS, given the difference in the sensitivity profile between IASI and MLS, the non-negligible IASI sensitivity to $HNO_3$ in the troposphere where MLS does not measure $HNO_3$ etc, which makes the integrated columns from IASI vs MLS not directly comparable. It should be pointed out finally that part of the differences between IASI and MLS are likely due to the different number of co-located data within the 2.5°x2.5° grid cells considered here for the comparison, with a much larger number of observations for IASI (through the quality filtering) than for MLS.

Despite this, the comparison shows similar spatial and seasonal variations between IASI total $HNO_3$ columns and MLS VMR between ~70 and 30 hPa in the different latitude bands, in particular, in the southernmost equivalent latitudes (see top panel). The strong $HNO_3$ depletion is well captured by both IASI and MLS measurements with a perfect match for the onset of the depletion. It further supports the good sensitivity of IASI to $HNO_3$ in the range of these pressure levels, justifying the methodology used in this study.

The cross-comparison with MLS is indeed insightful and gives further credit on the IASI observations during the polar night. That comparison figure between IASI and MLS has therefore been included in Section 2 of the revised manuscript and the text was changed to:
 "In order to expand on the comparisons against FTIR measurements which is impossible during the polar night, Figure 1 (top panel) presents the time series of daily IASI total $HNO_3$ columns co-located with MLS VMR measurements within 2.5x2.5 grid boxes at three pressure levels (at 30, 50 and 70 hPa), averaged in the 70°S–90°S equivalent latitude band. Similar variations in $HNO_3$ are captured by the two instruments with an excellent agreement for the timing of the strong $HNO_3$ depletion within the inner vortex core. IASI $HNO_3$ variations generally match well those of MLS $HNO_3$ in each latitude band (see Figure 1 bottom panel for the 50°S–70°S equivalent latitude band; the other bands are not shown here)."

**Specific comments**

[L3, 'good vertical sensitivity': This has not been shown in this paper. It is necessary to demonstrate this for the dataset discussed here given the cold Antarctic stratosphere.]
As stated in the text, we here refer to "a good vertical sensitivity in the low and middle stratosphere", not to a good vertical resolution of the measurement.

As mentioned in the manuscript, this paper builds on the previous studies of Ronsmans et al. (2016) and (2018). Despite a poor vertical resolution between the retrieved layers forcing us to consider a total column, the sensitivity of IASI to $HNO_3$ was shown to vary with altitude and to be highest in the low-middle stratosphere, even within the cold Antarctic polar vortex (Ronsmans et al. (2016)). This means that the variability in the measured total column is mainly representative of that layer. As said above, we recall here that similarly to the earlier studies, $HNO_3$ measurements characterized by a poor spectral fit or by a low vertical sensitivity (DOFS < 0.9) have been filtered out of this analysis. This is now clearly mentioned in Section 2 of the revised manuscript:

"Quality flags similar to those developed for $O_3$ in previous IASI studies (Wespes et al., 2017) were applied a posteriori to exclude data (i) with a corresponding poor spectral fit (e.g. based on quality flags rejecting biased or sloped residuals, fits with maximum number of iteration exceeded), (ii) with less reliability (e.g. based on quality flags rejecting suspect averaging kernels, data with less sensitivity characterized by a DOFS lower than 0.9) or (iii) with cloud contamination (defined by a fractional cloud cover larger than 25 %)."

[L8, 'denitrification': Are you certain, that 'denitrification' is also used for the uptake of HNO3 in particles? Perhaps 'removal from the gas phase'.]
We thank the referee for this remark. We are of course aware that the so-called "denitrification" defines the permanent removal of NOy from an airmass due to the gravitational sedimentation of NOy-containing particles. We agree that, from IASI, we can only measure a "removal from the gas phase", caused by sequestration into particles with or without sedimentation. Careful attention has now been made in the manuscript to avoid abusive use of the term "denitrification". Hence, "onset of $HNO_3$ denitrification" has been changed to "the onset of $HNO_3$ depletion" where appropriated in the revised manuscript. The title has also been changed accordingly to:
"Polar stratospheric $HNO_3$ depletion surveyed from a decadal dataset of IASI total columns".

[L59, 'a maximum sensitivity in the mid-stratosphere around 50 hPa': This must be shown here for the extreme conditions in the Antarctic vortex - also since all later analyses in the paper use temperatures at 50 hPa. What is the vertical variability of this level of maximum sensitivity within the development inside the vortex, especially later in the winter when, due to sedimentation of PSC particles, HNO3 concentrations at those levels are very low?]
See our responses to the general comments. Here again, we refer to the figure 3 (top and bottom panels) of Ronsmans et al. (2016) that presents the global distributions of the degrees of freedom for signal (DOFS, top panels) and of the altitude of maximum sensitivity of IASI to the $HNO_3$ profile (bottom panel), separately for January (left) and July (right) 2011, when the strong $HNO_3$ depletion occurs within the cold Antarctic winter. It shows clearly that the altitude of maximum sensitivity of the total columns is invariant at equatorial and tropical latitudes, whereas it varies with seasons at middle and polar latitudes. Above the Antarctic, the altitude of maximum sensitivity varies between ~9 km in summer and ~22 km in winter. The variations of the altitude of maximum sensitivity follow the altitude variations of maximum $HNO_3$ concentrations.

This is now more explicit at several places in the revised manuscript; e.g. in Section 1: "IASI provides reliable total column measurements of $HNO_3$ characterized by a maximum sensitivity in the low-middle stratosphere around 50 hPa (20 km) during the dark Antarctic winter (Ronsmans et al., 2016; 2018) …" and in Section 2: "… the largest sensitivity of IASI in the region of interest, i.e. in the low and mid-stratosphere (from 70 to 30 hPa), where the $HNO_3$ abundance is the highest (Ronsmans et al., 2016).

In order to convince the referee that IASI measurements capture the expected variations of $HNO_3$ within the polar night, we provide in Figure 2 below examples of vertical $HNO_3$ profiles retrieved within the dark Antarctic vortex (above Arrival Height) and outside the vortex (above Lauder). The retrieved profiles are shown along with their associated total retrieval error and averaging kernels (the total column AvK and the so-called "sensitivity profile" are also represented). The sum of the averaging kernels indicates how the true state at a specific altitude changes the retrieved total column, i.e. the altitude to which the retrieved total column is mainly sensitive/representative. Above Arrival Height during the dark Antarctic winter, we clearly see depleted $HNO_3$ levels in the low and mid-stratosphere and the altitude of maximum sensitivity at around 30 hPa. At Lauder on the contrary, $HNO_3$ levels larger than the a priori are observed in the stratosphere with a larger range of maximum sensitivity.

[L79, 'The total columns yield a total retrieval error of 10% and a low bias (10.5%) compared to ground-based FTIR measurements (Hurtmans et al., 2012; Ronsmans et al., 2016).': As these numbers are used also later in the manuscript, their validity has to be confirmed for the condition in the dark vortex, which cannot be achieved with comparisons to sun-dependent FTIR observations. As mentioned above, I strongly suggest to perform comparisons with the MLS dataset.]

Figure 4 of Ronsmans et al. (2016) illustrates the global distribution of the total retrieval error for $HNO_3$ (integrated over 5 to 35 km) separately for January (left) and July (right) over the period of the IASI measurements. The mid- and polar latitudes are characterized by low total retrieval errors of around ~3-5% - which corresponds to a reduction by a factor of 18-30 compared to the prior uncertainty (90%) and indicates a real gain of information – except above Antarctica during wintertime where the errors reach 25%. They are explained by (1) a weaker sensitivity (i.e. a larger smoothing error which represents in all cases the larger source of the retrieval error) above such cold surface (DOFS of ~0.95 within the dark Antarctic vortex – see figure 3 of Ronsmans et al., 2016) and by (2) a misrepresentation of the wavenumber-dependent surface emissivity above ice surface (Hurtmans et al., 2012). This is made more explicit in Section 2 of the revised manuscript:

"The total columns are associated with a total retrieval error ranging from around 3% at mid- and polar latitudes to 25% above cold Antarctic surface during winter (due to a weaker sensitivity above very cold surface with a DOFS of ~0.95 and to an poor knowledge of the seasonally and wavenumber-dependent emissivity above ice surfaces which induces larger forward model errors), and a low bias (lower than 12%) in polar regions over the altitude range where the IASI sensitivity is largest, when compared to ground-based FTIR measurements (see Hurtmans et al., 2012; Ronsmans et al., 2016 for more details)."

A validation against ground-based sun-dependent FTIR measurements could not be provided during the dark Antarctic winter, we refer on the good agreement with MLS (suggested by the referee) to underline the potentiality of IASI to detect the $HNO_3$ variations as well within the Antarctic winter (see general comment).

[L105, 'These high $HNO_3$ levels result from low sunlight…': This is not the only, and probably not the central explanation for the increasing column amounts. Dynamical effects on total columns of stratospheric gases (downwelling within the vortex) have to be considered.]

We thank the referee for this correction. The sentence has been rewritten as follows:

"These high $HNO_3$ levels result from low sunlight, preventing photodissociation, along with the heterogeneous hydrolysis of $N_2O_5$ to $HNO_3$ during autumn before the formation of polar stratospheric clouds (Keys et al., 1993; Santee et al., 1999; Urban et al., 2009; DeZafra et al., 2001). This period also corresponds to the onset of the deployment of the southern polar vortex which is characterized by strong diabatic descent with weak latitudinal mixing across its boundary, isolating polar $HNO_3$-rich air from lower latitudinal airmasses."

[Figure 2: I think the vertical dashed line '10Jun09' does not fit to the minimum of the solid blue curve (?)]

The referee is right; there was a bug for the automatically detection of the drop temperature, as well as for the detection of the corresponding dates in this figure. The figure has been corrected. The position of the drop temperatures does now perfectly match the yearly minima of the total $HNO_3$ second derivative. An average drop temperature over the ten years of IASI of 194.2 +/- 3.8 K is now calculated, which is even closer to $T_{NAT}$.

[L154, 'in the areas of potential vorticity smaller than -10 …': PV at which potential temperature level is used here?]
As mentioned in Section 2 of the submitted manuscript, "the potential vorticity (PV) fields are taken from the ECMWF ERA Interim Reanalysis dataset at the potential temperature of 530 K (corresponding to ~20 km altitude where the IASI sensitivity to $HNO_3$ is the highest during the Southern Hemisphere (S.H.) winter (Ronsmans et al., 2016)".

[L159, 'Note that the $HNO_3$ time series has been smoothed': As the drop temperatures (and dates) are introduced as the central new method presented in the manuscript, it is necessary to explore their behaviour in more detail. Can you give an estimate of the error of this measure by considering e.g. the effect of the numerical smoothing. Please show also the 1st derivative to be able to judge on the uncertainties of the 2nd derivative. How do the drop temperatures vary when using different pressure levels (e.g. 70 hPa)?]
As explained in the text, we actually only used a simple robust spline smoothing function to fill gaps in the time series, hence it has no particular impact on the detection of the drop temperature and its corresponding date.

Figure 3 here below represents the figure 2 of the manuscript along with the 1st derivative of $HNO_3$ total column with respect to time superimposed, as asked by the referee. We can clearly see that the minima of the 2nd derivative match or just precede those of the 1st derivative of total $HNO_3$ with respect to time.

Figure 4 below represents the figure 2 of the manuscript but for the temperature at 30 hPa (top panel) and 70 hPa (bottom panel) for the sake of comparison. As expected, the drop temperatures are the lowest when using the temperatures at 30 hPa. They vary from 185-195 K (~192K on average) at 30 hPa to 195-204 K (~198 K on average) at 70 hPa with values of ~189-202 K (~194 K on average) at 50 hPa.

As explained in the manuscript, the use of the 195 K at 50 hPa as single level for the analysis is justified by the fact that it corresponds best to the maximum of IASI vertical sensitivity during the polar night (see Figure 3 of Ronsmans et al. 2016 and responses to related comments above); another justification is found a posteriori by the consistency between the 195 K threshold temperature taken at 50 hPa and the onset of the strong total $HNO_3$ depletion seen by IASI, which matches the NAT development that occurs in June around that level. However, we fully agree that the $HNO_3$ abundances over a large part of the stratosphere (between 70 and 30 hPa) contribute to the total $HNO_3$ variations detected by IASI and that this inevitably affects the drop temperature calculation at 50 hPa. In order to address this issue, we have added in the manuscript the range of drop temperatures when calculated at these two other pressure levels (from 185 K to 204 K); this indeed allows the reader to better judge on the uncertainty of the drop temperature at 50 hPa (189-202 K). We thank the referee for his suggestion. The text in the revised manuscript is changed to:

"… Nevertheless, given the range of maximum IASI sensitivity to $HNO_3$ around 50 hPa, typically between 70 and 30 hPa (Ronsmans et al., 2016), the drop temperatures are also calculated at these two other pressure levels (not shown here) to estimate the uncertainty of the calculated drop temperature defined in this study at 50 hPa. The 30 hPa and 70 hPa drop temperatures range respectively over 185.7 K – 194.9 K and over 194.8 K – 203.7 K, with an average of 192.0 +/- 2.9 K and 198.0 +/- 3.2 K (1σ standard deviation) over the ten years of IASI. The average values at 30 hPa and 70 hPa fall within the 1σ standard deviation associated with the average drop temperature at 50 hPa. It is also worth noting the agreement between the drop temperatures and the NAT formation threshold at these two pressure levels ($T_{NAT}$ ~193 K at 30 hPa and ~197 K at 70 hPa) (Lambert et al., 2016)."

[L184, 'The calculated drop temperatures vary significantly between 180 and 210 K. These high extremes are only found in very few cases and should be considered with caution as they correspond to specific regions above ice shelves with emissivity features that are known to yield errors in the IASI retrievals': I find the discussion around the deviations of the drop temperatures very confusing. At the beginning of the manuscript it is stated, that the error of the measured total column amounts is in the order of 10%. Here it is argued that 'above ice shelves' it might be higher. Also, in Fig. 5 one can see that there are large regions over eastern Antarctica where drop temperatures are often clearly above 195K even inside the red circles. This is not explained satisfactorily in the manuscript. Here, again, it would be important to investigate on the reliability, consistency and homogeneity of the IASI HNO3 values. As mentioned above, this could be accomplished with a comparison to MLS observations.]

See our response above about the characterization of the $HNO_3$ retrievals in terms of total retrieval error and of its spatial/temporal distribution: The largest errors (25%) are found above Antarctica during wintertime and are due to (1) a weaker sensitivity (i.e. a larger smoothing error which represents in all cases the larger source of the retrieval error) above such cold land surface (DOFS of ~0.95 within the dark Antarctic vortex – see figure 3 of Ronsmans et al., 2016) and to (2) a poor knowledge of the (seasonally and wavenumber-dependent) emissivity of ice surfaces (Hurtmans et al., 2012). This is now clearly mentioned in Section 2 of the revised manuscript.

Bright land surface such as desert or ice might in some cases lead to poor $HNO_3$ retrievals due to a poor knowledge of the wavenumber-dependent emissivity above such surfaces, which can alter the retrieval by compensation effects (Wespes et al., 2009). FORLI relies on the monthly climatology of surface emissivity built by Zhou et al. (2011) from several years of IASI measurements on a 0.5x0.5 grid and for each 8461 IASI spectral channels when available, or on the MODIS climatology that is unfortunately restricted to only 12 channels in the IASI spectral range; see Hurtmans et al. (2012) for more details. Although wavenumber-dependent surface emissivity atlases are used in FORLI, it is clear that this parameter remains critical and causes poorer retrievals that, in some instances, pass the posterior filtering. The total $HNO_3$ columns over eastern Antarctica which show drop temperatures much above 195K might precisely be related to this. We have made this clear in Section 4.2 of the revised version:

"…emissivity features that are known to yield errors in the IASI retrievals. Indeed, bright land surface such as ice might in some cases lead to poor $HNO_3$ retrievals. Although wavenumber-dependent surface emissivity atlases are used in FORLI (Hurtmans et al., 2012), this parameter remains critical and causes poorer retrievals that, in some instances, pass through the series of quality filters and affect the drop temperature calculation."

[L195, 'Overall, despite these limitations, the spatial variability in the drop 50 hPa temperatures for IASI total $HNO_3$ is well in agreement with the natural variation in PSCs nucleation temperatures': Given the extended areas where the drop temperatures are larger than 195K, this statement is not convincing.]
The sentence has been rewritten for clarity:

"Except above some parts of Antarctica which are prone to larger errors, the overall range in the drop 50 hPa temperature for total $HNO_3$, inside the isocontour for the 195 K temperature, typically extends from ~187 K to 195 K, which fall within the range of PSCs nucleation temperature at 50 hPa …".

Furthermore, in response to G. Manney and M. Santee, the contour of $-10\times10^{-5}K.m^2.kg^{-1}.s^{-1}$ based on the minimum PV encountered at 50 hPa over the 10 May to 15 July period as well as the isocontours of 195 K at 50 hPa for the averaged temperatures and the minima over the same period are also now represented in the revised Fig.5 and the distribution of the drop temperatures is much better described and explained in the revised version:

"The averaged isocontour of 195 K encircles well the area of $HNO_3$ drop temperatures lower than 195 K, which means that the bins inside that area characterize airmasses that experience the NAT threshold temperature during a long time over the 10 May – 15 July period. That area encompasses the inner vortex core (delimited by the isocontour of $-10 \times 10^{-5} K.m^2.kg^{-1}.s^{-1}$ for the averaged PV), but is larger, and show pronounced minima (lower than $-0.5 \times 10^{14}$ molec.cm$^{-2}$.d$^{-2}$) in the second derivative of the $HNO_3$ total column with respect to time (not shown here), which indicate a strong and rapid $HNO_3$ depletion.

The area enclosed between the two isocontours of 195 K for the temperatures, the averaged one and the one for the minimum temperatures, show higher drop temperatures and weakest minima (larger than $-0.5 \times 10^{14}$ molec.cm$^{-2}$.d$^{-2}$) in the second derivative of the $HNO_3$ total column (not shown). That area is also enclosed by the isocontour of $-10 \times 10^{-5} K.m^2.kg^{-1}.s^{-1}$ for the minimum PV, meaning that the bins inside correspond, at least for one day over the 10 May – 15 July period, to airmasses located at the inner edge of the vortex and characterized by temperature lower than the NAT threshold temperature. The weakest minima in the second derivative of total $HNO_3$ (not shown) observed in that area indicate a weak and slow $HNO_3$ depletion and might be explained by a short period of the NAT threshold temperature experienced at the inner edge of the vortex. It could also reflect a mixing with strong $HNO_3$-depleted and colder airmasses from the inner vortex core. The mixing with these "already" depleted airmasses could also explained the higher drop temperatures detected in those bins. Finally, note also that these high drop temperatures are generally detected later (after the $HNO_3$ depletion occurs, i.e. after the 10 May – 15 July period considered here – not shown), which supports the transport, in those bins, of earlier $HNO_3$-depleted airmasses and the likely mixing at the edge of the vortex."

[L204, 'denitrification phase': See statement about 'denitrification' above.]
See our response above.

[L230, 'To the best of our knowledge, it is the first time that such a large satellite observational data set of stratospheric HNO3 concentrations is exploited to monitor the evolution HNO3 versus temperatures.': This sounds somehow exaggerated given all the previous work on HNO3/temperature/PSCs, e.g. by use of the MLS dataset and also since the correlation with temperature has already been shown in Ronsmans et al., 2018.]
We here simply refer to the unprecedented potential of IASI in terms of its exceptional spatial and temporal sampling. Ronsmans et al. (2018) also referred to the IASI dataset and correlations with temperature were done but in a lesser extent. In order to avoid overselling, the sentence has been rewritten:
"We show in this study that the IASI dataset allows capturing the variability of stratospheric $HNO_3$ throughout the year (including the polar night) in the Antarctic. In that respect, it offers a new observational means to monitor the relation of $HNO_3$ to temperature and the related formation of PSCs."

**Technical comments:**

L27, '(e.g. (Toon…))': I think the inner bracket level is not necessary.
L30, 'sedimentation(Lambert …): Space missing
L34, 'temperature': 'temperatures'
L51: Bracket levels?
L102, 'The red vertical line in Fig. 1a and Fig. 1b': There is no vertical red line in Fig. 1a. You mean horizontal?
L106, references: Brackets seem wrong.
Figure 2, caption, 'in the70—': Space missing.
L155, 'and the total HNO3 depletion are the coldest': Makes no sense.

L164, 'temperature are': 'temperatures are'

All the technical comments have been corrected.

[Figure]

Figure 1. Time series of daily IASI total $HNO_3$ column (blue, left y-axis) co-located with MLS and of MLS VMR $HNO_3$ within 2.5x2.5 grid boxes at three pressure levels (at 30, 50 and 70 hPa; right y-axis), averaged in the 70°S–90°S (top panel), the 50°S–70°S (middle panel) and in the 30°S–50°S (bottom panel) equivalent latitude bands. The error bars (light blue) represents 3σ, where σ is the standard deviation around the IASI $HNO_3$ daily average.

[Figure]

Figure 2. Examples of IASI HNO$_3$ vertical profiles (in molec.cm$^{-2}$) with corresponding averaging kernels (in molec.cm$^{-2}$/molec.cm$^{-2}$; with the total column averaging kernels (black) and the sensitivity profiles (grey)) above Arrival Height (77.49°S, 166.39°E, top panels) and Lauder (45.03°S, 169,40°E; bottom panels). The error bars associated with the HNO$_3$ vertical profile represent the total retrieval error. The a priori profile is also represented. The total column and the DOFS values are indicated.

[Figure]

Figure 3. Time series of total HNO₃ first derivative (green, left y-axis), of total HNO₃ second derivative (blue, left y-axis) and of the temperature at 50 hPa (red, right y-axis), in the region of potential vorticity lower than -10 x10⁻⁵ K.m².kg⁻¹.s⁻¹. The red horizontal line corresponds to the 195 K temperature. The vertical dashed lines indicate the second derivative minimum in HNO₃ for each year. The corresponding dates (in bold, on the x-axis) and temperatures are also indicated. The time series of total HNO₃ second derivative (dashed blue) and of temperature at 50 hPa (grey) in the 70–90°S Eqlat band are also represented.

[Figure]

Figure 4. Same as Figure 3 but for the temperature at 30 hPa (top panel) and 70 hPa (bottom panel).

---

## Author Comment (AC2) · 15 Dec 2020

**Response to reviewer #3**

We thank the reviewer for her/his in depth review comments that have help us to improve the clarity of the manuscript. Kindly find below our responses to each of the comments (quoted between []). We hope that our responses will address the main issues and that the changes made will convince that the IASI $HNO_3$ dataset has the potential to contribute to stratospheric studies and, more particularly, to the time evolution of the polar processes.

**Major comments**

[The description of the polar HNO3 variation presented in the paper is already well known from numerous other studies.]

The purpose of this paper is to demonstrate the interest of IASI for $HNO_3$ stratospheric studies (Ronsmans et al., 2018) after having undergone a rigorous validation exercise (Ronsmans et al., 2016). If limb measurements allows resolving the $HNO_3$ profile in the stratosphere, the potential of IASI lies in its exceptional spatial and temporal sampling. We demonstrate here that despite its limited vertical resolution forcing us to consider one total column, the information content that actually lies in the low and middle stratosphere offers potential to expand on previous polar stratospheric denitrification studies, usually performed using limb sounder measurements, and to continue the long-term records of $HNO_3$ started with the latter. We have tried in this paper not to repeat too much of our earlier work but some duplication was unavoidable; in particular, with respect to vertical sensitivity and errors (these are two aspects that referee1 finds in fact insufficiently described here).

[The lack of vertical resolution in the IASI HNO3 measurements severely limits the interpretation of the results and precludes differentiation between denitrification and renitrification e.g. consider the effect of the vertical integration through depleted higher layers overlaying lower enhanced layers.]

We understand that the referee sees this as a limitation. However, despite the lack of vertical resolution, which is recognized in the paper and which forces us to consider total $HNO_3$ columns, IASI is characterized by a good sensitivity to $HNO_3$ at specific levels, in particular, in the range between ~70 hPa to ~30 hPa in the southernmost latitude in winter and as such it provides an adequate means to investigate the stratospheric processes in the polar nights.

In order to justify this further, we would like to refer to the figure 3 (top and bottom panels) of Ronsmans et al. (2016) that presents global distributions of the degrees of freedom for signal (DOFS, top panels) and of the altitude of maximum sensitivity of IASI to the $HNO_3$ profile (bottom panel), separately for January (left) and July (right) 2011, when the strong $HNO_3$ depletion occurs within the cold Antarctic winter. It shows clearly that the altitude of maximum sensitivity of the total columns is invariant at equatorial and tropical latitudes, whereas it varies with seasons at middle and polar latitudes. Above the Antarctic, the altitude of maximum sensitivity varies between ~9 km in summer and ~22 km in winter. The variations of the altitude of maximum sensitivity follow the altitude variations of maximum $HNO_3$ concentrations.

We agree that the IASI sensitivity was insufficiently put forward in the text. We made it more explicit at several places in the revised manuscript; e.g. in Section 1: "IASI provides reliable total column measurements of $HNO_3$ characterized by a maximum sensitivity in the low-middle stratosphere around 50 hPa (20 km) during the dark Antarctic winter (Ronsmans et al., 2016; 2018) …" and in Section 2: "… the largest sensitivity of IASI in the region of interest, i.e. in the low and mid-stratosphere (from 70 to 30 hPa), where the $HNO_3$ abundance is the highest (Ronsmans et al., 2016)."

[Although the IASI HNO3 data has much better 2D horizontal resolution than any other measurement this has not been developed as a tool to provide information beyond that of satellite instruments that measure only along the orbit track.]

We do not fully agree. The determination of the drop temperature using the second derivative exploits the large dataset of daily IASI measurements. Furthermore, the spatial distributions of the drop temperature calculated at 50hPa, which are presented in the figure 5 of the manuscript, do actually take advantage of the excellent spatial/temporal resolution of IASI to provide information throughout the entire vortex and outside. This would probably not be feasible with other types of measurements.

[CALIOP PSC information is available for the same time frame, why was this not used? Certainly, PSC volumes vs time would be helpful in providing the underlying interannual variability of PSC types (NAT, STS, ice) to compare with the resulting drop temperatures derived from IASI. Similarly, at least some comparisons of the IASI HNO3 column with integrated column calculated from Aura MLS are necesssary to establish the validity of the measurements in the most severely depleted inner vortex core.]

Thank you for this comment. It is certainly a good idea to use the CALIOP measurements in support but this goes beyond the goal of this paper, which is to demonstrate the capability of IASI to measure $HNO_3$ columns that are relevant for stratospheric studies. Using CALIOP PSC information and, in particular, comparing the spatial distributions of IASI derived drop temperatures (Figure 5 of the revised paper) with maps of CALIOP PSC would be very interesting in order to go a step further in the analyses of the underlying $HNO_3$ condensation processes, but it will be challenging and add significant complexity given the high variability in the distribution of PSC types.

Regarding the second point on a comparison with MLS, we fully agree that this is highly relevant; it was also a request of referee#1. We provide here below a comparison with observations by MLS in three equivalent latitude bands (see Figure 1). We would like to point out that we here compare total columns measured by IASI with VMR measured by MLS at several pressure levels that cover the highest sensitivity of IASI (at ~50 hPa, ~70 hPa and ~30 hPa for the sake of the comparison). Hence, the comparison of IASI columns with MLS measurements is mostly qualitative at this stage and differences are expected for this reason. Note also that we have preferred comparing IASI $HNO_3$ columns with VMR measured by MLS at specific levels instead of integrated columns calculated from MLS, given the difference in the sensitivity profile between IASI and MLS, the non-negligible IASI sensitivity to $HNO_3$ in the troposphere where MLS does not measure $HNO_3$ etc, which makes the integrated columns from IASI vs MLS not directly comparable. It should be pointed out finally that part of the differences between IASI and MLS are likely due to the different number of co-located data within the 2.5°x2.5° grid cells considered here for the comparison, with a much larger number of observations for IASI (through the quality filtering) than for MLS.

Despite this, the comparison shows similar spatial and seasonal variations between IASI total $HNO_3$ columns and MLS VMR between ~70 and 30 hPa in the different latitude bands, in particular, in the southernmost equivalent latitudes (see top panel). The strong $HNO_3$ depletion is well captured by both IASI and MLS measurements with a perfect match for the onset of the depletion. It further supports the good sensitivity of IASI to $HNO_3$ in the range of these pressure levels, justifying the methodology used in this study.

The cross-comparison with MLS is indeed insightful and gives further credit on the IASI observations during the polar night. That comparison figure between IASI and MLS has therefore been included in Section 2 of the revised manuscript and the text was changed to:

"In order to expand on the comparisons against FTIR measurements which is impossible during the polar night, Figure 1 (top panel) presents the time series of daily IASI total $HNO_3$ columns co-located with MLS VMR measurements within 2.5x2.5 grid boxes at three pressure levels (at 30, 50 and 70 hPa), averaged in the 70°S–90°S equivalent latitude band. Similar variations in $HNO_3$ are captured by the two instruments with an excellent agreement for the timing of the strong $HNO_3$ depletion within the inner vortex core. IASI $HNO_3$ variations generally match well those of MLS $HNO_3$ in each latitude band (see Figure 1 bottom panel for the 50°S–70°S equivalent latitude band; the other bands are not shown here)."

[Regarding the sensitivity of the IASI column HNO3 measurements, I suggest presenting a few examples of vertical HNO3 profiles (from a model or data), ranging from non-depleted to extreme depletion with calculations of the corresponding calculated integrated IASI column. This would help to indicate the sensitivity of the column measurement to changes in the vertical distribution of HNO3 ... i.e. generate profiles of the change in the IASI column HNO3 wrt the actual change in HNO3 at a level, j, d(column)/d(HNO3)j.]

This is an example of information reported in earlier work and that we have tried not to repeat extensively here. To summarize, the validation study of Ronsmans et al. (2016) provides a complete characterization of the IASI $HNO_3$ retrievals: it shows example of vertical $HNO_3$ profiles along with the total retrieval error, the a apriori profiles and associated averaging kernels profiles ($d(HNO3_{ret})i/d(HNO3_{true})j$), along with the total column averaging kernel ($d(column_{ret})/d(HNO3_{true})j$) and the sensitivity profile ($d(HNO3_{ret})i/d(column_{true})$), were already given in Figures 1 and 2 of that study. Note that the averaging kernel profile describes how the true state changes the estimate at a specific altitude, i.e. how the retrieval smooths the true profile. The sum of the elements of an averaging kernel characterizing the retrieval at a specific altitude returns the sensitivity of the retrieval at that altitude, i.e. to which extent the retrieval at that specific altitude comes from the spectral measurement rather than the apriori, while the sum of the averaging kernels indicates how the true state at a specific altitude changes the retrieved total column, i.e. the altitude to which the retrieved total column is mainly sensitive/representative.

Figure 3 (top and bottom panels) of Ronsmans et al. (2016) further presents the global distributions of the degrees of freedom for signal (DOFS, top panels) and of the altitude of maximum sensitivity of the retrieval to the $HNO_3$ profile (bottom panel), separately for January (left) and July (right) 2011, when the strong $HNO_3$ depletion occurs within the cold Antarctic winter. It clearly shows that above the Antarctic, the altitude of maximum sensitivity varies between ~9 km in summer and ~22 km in winter (~ 50 hPa) on average.

To address the comment of the referee without repeating too much of the earlier results, we have carefully verified the manuscript with regard to unclear or incomplete statements about vertical sensitivity. The following has been added in Section 1: "IASI provides reliable total column measurements of HNO3 with a maximum sensitivity in the low-middle stratosphere around 50 hPa (20 km) during the dark Antarctic winter (Ronsmans et al., 2016; 2018) …" and in Section 2: "… the largest sensitivity of IASI in the region of interest, i.e. in the low and mid-stratosphere (from  70 to 30 hPa), where the $HNO_3$ abundance is the highest (Ronsmans et al., 2016).

In order to convince the referee that IASI measurements capture the expected variations of $HNO_3$ within the polar night, we provide in Figure 1 below examples of vertical $HNO_3$ profiles retrieved within the dark Antarctic vortex (above Arrival Height) and outside the vortex (above Lauder). The retrieved profiles are shown along with their associated total retrieval error and averaging kernels (the total column AvK and the so-called "sensitivity profile" are also represented). Above Arrival Height during the dark Antarctic winter, we clearly see depleted $HNO_3$ levels in the low and mid-stratosphere and the altitude

of maximum sensitivity at around 30 hPa. At Lauder on the contrary, $HNO_3$ levels larger than the a priori are observed in the stratosphere with a larger range of maximum sensitivity.

**Specific comments**

[L2: "good vertical sensitivity" ... only column HNO3 measurements are discussed here - there is no vertical resolution in the measurements.]
See our response to the second general comment above.

As stated in the text, we here refer to "a good vertical sensitivity in the low and middle stratosphere", not to a good vertical resolution of the measurement. Note that $HNO_3$ vertical profile are retrieved from IASI measurements, not simply total columns; Hence, even if the sensitivity covers the entire altitude range from the troposphere to the stratosphere with no clear decorrelation (poor resolution) between the retrieved layers forcing us to consider a total column, it is shown to variate with the altitude and to be highest in the low-middle stratosphere, which means that the variability in the measured total column is mainly representative of that layer.

As mentioned in the manuscript, this paper builds on the previous studies of Ronsmans et al. (2016) and (2018), where the vertical sensitivity of IASI to $HNO_3$ measurements is shown to be highest in the low and mid-stratosphere, even within the cold Antarctic polar vortex, with the degrees of freedom for signal (DOFS) that ranges from 0.9 to 1.2 at all latitudes. Note also that similarly to these two previous studies, $HNO_3$ measurements characterized by a poor spectral fit or by a low vertical sensitivity (DOFS < 0.9) have been filtered out of this analysis. This is now clearly mentioned in Section 2 of the revised manuscript:

"Quality flags similar to those developed for $O_3$ in previous IASI studies (Wespes et al., 2017) were applied a posteriori to exclude data (i) with a corresponding poor spectral fit (e.g. based on quality flags rejecting biased or sloped residuals, fits with maximum number of iteration exceeded), (ii) with less reliability (e.g. based on quality flags rejecting suspect averaging kernels, data with less sensitivity characterized by a DOFS lower than 0.9) or (iii) with cloud contamination (defined by a fractional cloud cover larger than 25 %)."

[L10: 191K is also consistent with STS temperatures (192 K) and is actually closer than TNAT (195 K)]
Indeed but as stated in the manuscript: "… recent observational and modelling studies have shown that $HNO_3$ starts to condense in early PSC season in liquid NAT mixtures well above Tice (~4 K below $T_{NAT}$, close to $T_{STS}$)…". The NAT nucleation temperature at 50 hPa range from slightly below $T_{NAT}$ to around 3-4 K below Tice, depending on atmospheric conditions, on TTE and on the type of formation mechanisms (Pitts et al., 2011; Peter and GrooS, 2012; Hoyle et al., 2013).

Note that in replying to referee#1 we have identified a bug for the automatically detection of the drop temperature, as well as for the detection of the corresponding dates in the figure 2 of the manuscript. It has been corrected. The position of the drop temperatures does now perfectly match the yearly minima of the total $HNO_3$ second derivative. An average drop temperature over the ten years of IASI of 194.2 +/- 3.8 K is now calculated, which is even closer to $T_{NAT}$.

Finally, as requested by referee #1, we also now clearly mention in Section 4.1 of the manuscript the range of drop temperatures when calculated at two other pressure levels to better judge on the uncertainty of the drop temperature at 50 hPa (see Figure 3 here below):

"… Nevertheless, given the range of maximum IASI sensitivity to $HNO_3$ around 50 hPa, typically between 70 and 30 hPa (Ronsmans et al., 2016), the drop temperatures are also calculated at these two other pressure levels (not shown here) to estimate the uncertainty of the calculated drop temperature defined in this study at 50 hPa. The 30 hPa and 70 hPa drop temperatures range respectively over 185.7 K – 194.9 K and over 194.8 K – 203.7 K, with an average of 192.0 +/- 2.9 K and 198.0 +/- 3.2 K ($1\sigma$ standard deviation) over the ten years of IASI. The average values at 30 hPa and 70 hPa fall within the $1\sigma$ standard deviation associated with the average drop temperature at 50 hPa. It is also worth noting the agreement between the drop temperatures and the NAT formation threshold at these two pressure levels ($T_{NAT}$ ~193 K at 30 hPa and ~197 K at 70 hPa) (Lambert et al., 2016)."

[L18: add more recent references e.g. Peter and Gross (2012). L28: Much more has been done in the past decade with MIPAS and CALIOP that should be referenced]
Thank you for this suggestion. Peter and GrooS (2012) was cited elsewhere in the manuscript but has been added here as well. Note that the goal of the introduction is not to provide an exhaustive list of all studies related to the PSC thermodynamics. Several general reference papers are cited and we have decided to put more focus here on $HNO_3$.

[L59: This section should explain what is meant by "maximum sensitivity" etc.]
See our responses to the second major comment and specific comments above.

[L79: Information on the data quality for IASI HNO3 is poor. Is the value of bias and uncertainty the same for depleted and non-depleted conditions?]
The reader is here invited to refer to the figure 4 of Ronsmans et al. (2016) which illustrates the global distribution of the total retrieval error for $HNO_3$ (integrated over 5 to 35 km) separately for January (left) and July (right) over the period of the IASI measurements. The mid- and polar latitudes are characterized by low total retrieval errors of around ~3-5% - which corresponds to a reduction by a factor of 18-30 compared to the prior uncertainty (90%) and indicates a real gain of information – except above Antarctica during wintertime where the errors reach 25%. They are explained by (1) a weaker sensitivity (i.e. a larger smoothing error which represents in all cases the largest source of the retrieval error) above such cold surface (DOFS of ~0.95 within the dark Antarctic vortex – see figure 3 of Ronsmans et al., 2016) and by (2) a poor knowledge of the wavenumber-dependent surface emissivity above ice surface, which also varies in time (Hurtmans et al., 2012). ). This is made more explicit in Section 2 of the revised manuscript:

"The total columns are associated with a total retrieval error ranging from around 3% at mid- and polar latitudes to 25% above cold Antarctic surface during winter (due to a weaker sensitivity above very cold surface with a DOFS of ~0.95 and to an poor knowledge of the seasonally and wavenumber-dependent emissivity above ice surfaces which induces larger forward model errors), and a low bias (lower than 12%) in polar regions over the altitude range where the IASI sensitivity is largest, when compared to ground-based FTIR measurements (see Hurtmans et al., 2012; Ronsmans et al., 2016 for more details)."

[L82: Yet, problems with the retrievals because of cloud contamination seem to remain even after the <25% cloud fraction filter is applied.]
We do not understand the referee comment here. In this section of the manuscript, we only describe the quality flags used in our analysis.

[L83: Cloud contamination? Tropospheric cloud only or also thick ice PSCs?]

The clouds that have most impact are clearly tropospheric water clouds. Cirrus clouds or PSCs are mostly transparent in the IR; thick cirrus however show up in the longwave part of the IASI spectrum, below 900 cm$^{-1}$. We have added "tropospheric cloud contamination" in the text.

Note that the threshold of 25 % cloud cover was carefully chosen after a series of tests, which have shown that these scenes could be treated as cloud-free without significant impact on the retrievals (Hurtmans et al., 2012).

[L102: Why was 2011 chosen?]
As expected from figure 1c, any other year could have been chosen instead of the year 2011 to illustrate the HNO$_3$ total columns versus temperatures (at 50 hPa) histogram in figure 1b. It is now clearly mentioned in the revised manuscript:

"Similar histograms are observed for the ten years of IASI measurements (not shown)."

[L106: Heterogeneous hydrolysis of N2O5 requires aerosol particles. So this process starts with cold binary aerosols (i.e. sulfates) before the formation of STS?]
Indeed, previous studies have shown enhanced HNO$_3$ columns during autumn in Antarctica and have attributed them to decreasing sunlight and conversion of N$_2$0$_5$ to HNO$_3$ by the reaction of N$_2$0$_5$ with background aerosols, before the formation of polar stratospheric clouds (e.g. Keys et al., Nature, 1993). At these temperatures, the conversion may occur on binary sulfuric aerosols.

The sentence has been rewritten as follows:

"These high HNO$_3$ levels result from low sunlight, preventing photodissociation, along with the heterogeneous hydrolysis of N$_2$O$_5$ to HNO$_3$ during autumn before the formation of polar stratospheric clouds (Keys et al., 1993; Santee et al., 1999; Urban et al., 2009; DeZafra et al., 2001). This period also corresponds to the onset of the deployment of the southern polar vortex which is characterized by strong diabatic descent with weak latitudinal mixing across its boundary, isolating polar HNO$_3$-rich air from lower latitudinal airmasses."

[L129: The onset of depletion seems to start when the temperatures fall substantially below 190K from inspection of Fig 1(c) and quite far below the red line marked at 195K.]
The onset of HNO$_3$ depletion starts in June at around 190K, which is in agreement with figure 1a.

[L136-137: Why are two temperatures (180 and 185 K) quoted for 30hPa? Why is the actual value from Fig1(c) (I estimate this as about 188K) for the 50hPa temperature not given in L129?]
The sentence has been rewritten for clarity:
"The results (not shown here) exhibit a similar HNO$_3$-temperature behaviour at the different levels with, as expected, lower and larger temperatures in R2, respectively, at 30 hPa (down to 180 K) and at 70 hPa 145 (down to 185K), but still below the NAT formation threshold at these pressure levels ($T_{NAT}$ =193 K at 30 hPa and 197 K at 70 hPa) (Lambert et al., 2016)."

[L138: "characterized by" seems the wrong description for the chance occurrence that the maximum sensitivity of IASI HNO3 falls in the same altitude range as the PSCs.]
Changed to: "… the altitude range of maximum IASI sensitivity to HNO$_3$ (see Section 2) is characterized by temperatures that are below the NAT formation threshold at these pressure levels, enabling the PSCs formation and the denitrification process."

[L139-146: This section rather seems to belong in the conclusions.]
L150-154 of the revised manuscript has been moved to the conclusions.

[L148: Clearly this does not "go beyond the vertically integrated view" since the column HNO3 is all that is available. It could be reworded as "To identify the spatial and temporal variability of the column HNO3 ..."]
Corrected as suggested.

[L165-169: Denitrification is the term used to describe the permanent removal of some HNO3 from the gas phase by sedimentation of PSCs. Sequestration is the term used to describe the uptake of HNO3 from the gas phase into PSCs. Denitrification by STS is a lengthy process compared to NAT since the smaller STS particles sediment slowly. STS can (and frequently does) form without the prior nucleation of NAT. IASI alone cannot discriminate between these processes and it should not be assumed that what is observed is the "onset of HNO3 denitrification".]
We thank the referee for this remark. We are of course aware of the definition of the so-called "denitrification". We agree that, from IASI, we can only measure a "removal from the gas phase", caused by sequestration into particles with or without sedimentation. Careful attention has now been made in the manuscript to avoid abusive use of the term "denitrification". Hence, "onset of HNO$_3$ denitrification" has been changed to "the onset of HNO$_3$ depletion" in L.169 and where appropriated in the revised manuscript and he title has also been changed accordingly to:
"Polar stratospheric HNO$_3$ depletion surveyed from a decadal dataset of IASI total columns".

[L185-187: 210K is much too high for PSC formation, but could possibly be NAT that is in process of melting? If these are observed over ocean then they warrant further investigation. However, why are specific regions with emissivity features not flagged as such? They should be discarded rather than "used with caution".]
Bright land surface such as desert or ice might in some cases lead to poor HNO$_3$ retrievals due to a poor knowledge of the wavenumber-dependent emissivity above such surfaces, which can alter the retrieval by compensation effects (Wespes et al., 2009). FORLI relies on the monthly climatology of surface emissivity built by Zhou et al. (2011) from several years of IASI measurements on a 0.5x0.5 grid and for each 8461 IASI spectral channels when available, or on the MODIS climatology that is unfortunately restricted to only 12 channels in the IASI spectral range; see Hurtmans et al. (2012) for more details. Although wavenumber-dependent surface emissivity atlases are used in FORLI, it is clear that this parameter remains critical and causes poorer retrievals that, in some instances, pass the posterior filtering. The total HNO$_3$ columns over eastern Antarctica which show drop temperatures much above 195K might precisely be related to this. We have made this clear in Section 4.2 of the revised version:

"…emissivity features that are known to yield errors in the IASI retrievals. Indeed, bright land surface such as ice might in some cases lead to poor HNO$_3$ retrievals. Although wavenumber-dependent surface emissivity atlases are used in FORLI (Hurtmans et al., 2012), this parameter remains critical and causes poorer retrievals that, in some instances, pass through the series of quality filters and affect the drop temperature calculation."

We refer on the good agreement with MLS (suggested by the referee) to underline the potentiality of IASI to detect the HNO$_3$ variations as well within the Antarctic winter (see general comment and Figure 1 here below).

[L189: Modern reanalysis temperatures (e.g. ERA-I) do not "feature large uncertainties" large enough to account for a 195K to 210K shift. L195-L201: The limitations of the reanalysis temperatures seems

to be an accuracy of better than 1K and clearly this in no way limits the derivation of the "50hPa drop temperature" which simply necessitates finding the 50hPa reanalysis temperature that corresponds to the second derivative wrt time minimum in column HNO3.]

We agree with the referee's comment; the discussion about the potential role of the uncertainty of the ECMWF reanalysis temperature on the drop temperature has been removed from the section, hence, this paragraph has been strongly revised accordingly:

"Biases in ECMWF reanalysis are too small for explaining the spatial variation in drop temperatures. Thanks to the assimilation of an advanced Tiros Operational Vertical Sounder (ATOVS) around 1998–2000 in reanalyses, to the better coverage of satellite instruments and to the use of global navigation satellite system (GNSS) radio occultation (RO) (Schreiner et al., 2007; Wang et al., 2007; Lambert and Santee, 2018; Lawrence et al., 2018), the uncertainties have been vastly reduced. Comparisons of the ECMWF ERA Interim dataset used in this work with the COSMIC data (Lambert and Santee, 2018) found a small warm bias, with median differences around 0.5 K, reaching 0–0.25 K in the southernmost regions of the globe at ~68–21 hPa where PSCs form."

[What is meant by "spatial variability"? The plots in Fig 5 show the spatial distribution of the drop temperature over a number of years but what variability is being considered? Interannual? Why have these spatial maps of drop temperatures not been compared with published maps of PSC types made by CALIOP or MIPAS. Wouldn't some correlation be expected according to the arguments made here? i.e. NAT PSCs at the higher temperature e.g. the highest temperatures (orange) appear downstream of the Palmar Peninsula in the "NAT ring" structure described by Hopfner et al (2006).]

Corrected: "Figure 5 shows the spatial variability" → "Figure 5 shows the spatial distribution".

We do not understand the referee's comment here. Figure 5 of the manuscript shows the spatial distribution of the drop temperature calculated inside a region enclosed by an isocontour PV of -8x10$^{-5}$ K.m$^2$.kg$^{-1}$.s$^{-1}$, which, hence, encircles a region larger than the inner vortex core (see Figures 3 and 4 of the manuscript). The drop temperatures much above the NAT formation temperature, which are mostly found outside the averaged isocontour PV of -10x10$^{-5}$ K.m$^2$.kg$^{-1}$.s$^{-1}$, do not correspond to high minima (>-0.5 x10$^{14}$ molec.cm$^{-2}$.d$^{-2}$) in the second derivative of HNO$_3$ total column with respect to time. We cannot argue that it corresponds to the NAT belt of Höpfner et al. (2006) downstream of the Antarctic Peninsula, which was enclosed inside the region of the NAT threshold temperature; the highest drop temperatures from IASI are found on the contrary outside the isocontour of the NAT threshold temperature (see figure 5 of the revised manuscript). In addition, comparing the distributions of drop temperatures from IASI with PSC information from CALIPSO/MIPAS remain difficult given the difference in spatial coverage and, most importantly, the highly variable distribution of PSC types and of the NAT belt, temporally (daily) and spatially (Höpfner et al., 2006; Lambert et al., 2012).

Finally, in response to G. Manney and M. Santee, the contour of −10×10$^{-5}$K.m$^2$.kg$^{-1}$.s$^{-1}$ based on the minimum PV encountered at 50 hPa over the 10 May to 15 July period as well as the isocontours of 195 K at 50 hPa for the averaged temperatures and the minima over the same period are also now represented in the revised Fig.5 and the distribution of the drop temperatures is much better described and explained in the revised version:

"The averaged isocontour of 195 K encircles well the area of HNO$_3$ drop temperatures lower than 195 K, which means that the bins inside that area characterize airmasses that experience the NAT threshold temperature during a long time over the 10 May – 15 July period. That area encompasses the inner vortex core (delimited by the isocontour of −10×10$^{-5}$K.m$^2$.kg$^{-1}$.s$^{-1}$ for the averaged PV), but is larger, and show pronounced minima (lower than -0.5 x10$^{14}$ molec.cm$^{-2}$.d$^{-2}$) in the second derivative of the HNO$_3$ total column with respect to time (not shown here), which indicate a strong and rapid HNO$_3$ depletion.

The area enclosed between the two isocontours of 195 K for the temperatures, the averaged one and the one for the minimum temperatures, show higher drop temperatures and weakest minima (larger than -0.5 x$10^{14}$ molec.cm$^{-2}$.d$^{-2}$) in the second derivative of the HNO$_3$ total column (not shown). That area is also enclosed by the isocontour of $-10\times10^{-5}$K.m$^2$.kg$^{-1}$.s$^{-1}$ for the minimum PV, meaning that the bins inside correspond, at least for one day over the 10 May – 15 July period, to airmasses located at the inner edge of the vortex and characterized by temperature lower than the NAT threshold temperature. The weakest minima in the second derivative of total HNO$_3$ (not shown) observed in that area indicate a weak and slow HNO$_3$ depletion and might be explained by a short period of the NAT threshold temperature experienced at the inner edge of the vortex. It could also reflect a mixing with strong HNO$_3$-depleted and colder airmasses from the inner vortex core. The mixing with these "already" depleted airmasses could also explained the higher drop temperatures detected in those bins. Finally, note also that these high drop temperatures are generally detected later (after the HNO$_3$ depletion occurs, i.e. after the 10 May – 15 July period considered here – not shown), which supports the transport, in those bins, of earlier HNO$_3$-depleted airmasses and the likely mixing at the edge of the vortex."

[L205: Nothing has been presented that demonstrates PSC occurrence. For that you would need to compare to actual data on PSCs from CALIOP and/or MIPAS.]
Corrected: "PSCs occurrence" → "NAT formation temperature"

[L224: Again, the suspect data should be discarded because of the detrimental impact on the scientific analysis. Also, if you cannot manage to work out and apply adequate quality control to your own data then you have no reason to expect anyone else to do so.]
See our response to comment [L185-187] above.

[L230: "To the best of our knowledge, it is the first time that such a large satellite observational data set of stratospheric HNO3 concentrations is exploited to monitor the evolution HNO3 versus temperatures" In fact you cite several papers that have done exactly this, but let's take the one published over two decades ago by Santee et al (1999) titled "Six years of UARS Microwave Limb Sounder HNO3 observations : Seasonal, interhemispheric, and interannual variations in the lower stratosphere". https://doi.org/10.1029/1998JD100089. Not only does this paper compare HNO3 with UKMO temperatures we are referred to a more complete paper on this topic on p8241 ... "The correlation of the HNO3 behavior with temperature during this time period, and its implications for PSC phase and composition, is explored in detail by Santee et al (1998). I noticed that the outside edge of the "HNO3 collar region" at 465K was defined by these authors as inside the 0.25 x E-4 K m2 kg-1 s-1 PV contour. This seems at odds with the 1E-4 value that is used for the second derivative minimum calculation in this paper and seemingly places the boundary quite far equatorward. Santee et al (1998) also includes a description of the heterogeneous hydration of N2O5 that would be helpful in response to the question above on L106.]
We here simply refer to the unprecedented potential of IASI in terms of its exceptional spatial and temporal sampling. Ronsmans et al. (2018) also referred to the IASI dataset and correlations with temperature were done but in a lesser extent. In order to avoid overselling, the sentence has been rewritten:
"We show in this study that the IASI dataset allows capturing the variability of stratospheric HNO$_3$ throughout the year (including the polar night) in the Antarctic. In that respect, it offers a new observational means to monitor the relation of HNO$_3$ to temperature and the related formation of PSCs."

In this study, we use the PV fields taken from the ECMWF ERA Interim Reanalysis dataset at the potential temperature of 530 K (corresponding to ~20 km where the IASI sensitivity to HNO$_3$ is the highest), while Santee et al. (1998) considered 465K. We clearly see from Figures 3a and 4 of the

manuscript that PV contours at -0.5e-4 K m2 kg-1 s-1 and at -0.8e-4 K m2 kg-1 s-1 encompass the so-called $HNO_3$ collar region. The PV value of -1e-4 K m2 kg-1 s-1 is used in this study to calculate the drop temperature based on the second derivative minimum as it clearly encompass the regions inside the inner polar vortex (see Figure 3a and 4 of the manuscript).

[L231: "It could constitute a new accurate climatological parameter that could be inserted in the PSCs classification schemes." The analysis presented does not support this statement. Specifically, how could the HNO3 column amount be used in a classification scheme?]
This sentence has been removed.

**Technical comments:**

L8: in [the] Antarctic
L53: Studies of HNO3 depletion and PSC formation predate the sensors named in the paragraph e.g. the Santee et al (1999) reference used UARS/MLS launched in 1991, measurement using balloons should have been be referenced here.
L108: extends
Figure 1 caption: Each figure title in 1(b) needs to state the year e.g. "January - December 2011 or put a label "2011" above the whole figure.
Figure 1 caption: 50 hpa => 50 hPa
Figure 1 caption: it is not clear to what 0.1E16 molec. cm-2. This low value is not even on the y-axis of the figures.
Figures 1(a) and 1(c): Are the HNO3 and temperature structures (localized peaks and valleys) visible in the time series in 1(a) quite well correlated when plotted as a scatter diagram as in 1(c), but without the 7-day averaging?
L123: 7-day
L124 and Figure 1 caption: "in the range of" : only one value is given and not a range of values
L130: Supplementary material - this does not appear to be available from the ACP website.
L164: drop temperatures
Figure 3 caption: sumperimposed => superimposed
L170: Figures 3a and b
L171: three isocontour levels
L174: lines indicate
L200: It underlines ... What does "it" refer to? The subject of the previous sentence is "the spatial variability" but that has not been defined.
L201:critical denitrification phase
L205: to PSCs occurrence to PSCs ??
L240: "All authors contributed to the writting of the text and reviewed the manuscript."
writting => writing

All the technical comments have been corrected.

[Figure]

Figure 1. Time series of daily IASI total HNO$_3$ column (blue, left y-axis) co-located with MLS and of MLS VMR HNO$_3$ within 2.5x2.5 grid boxes at three pressure levels (at 30, 50 and 70 hPa; right y-axis), averaged in the 70°S–90°S (top panel), the 50°S–70°S (middle panel) and in the 30°S–50°S (bottom panel) equivalent latitude bands. The error bars (light blue) represents 3σ, where σ is the standard deviation around the IASI HNO$_3$ daily average.

[Figure]

Figure 2. Examples of IASI HNO$_3$ vertical profiles (in molec.cm$^{-2}$) with corresponding averaging kernels (in molec.cm$^{-2}$/molec.cm$^{-2}$; with the total column averaging kernels (black) and the sensitivity profiles (grey)) above Arrival Height (77.49°S, 166.39°E, top panels) and Lauder (45.03°S, 169,40°E; bottom panels). The error bars associated with the HNO$_3$ vertical profile represent the total retrieval error. The a priori profile is also represented. The total column and the DOFS values are indicated.

[Figure]

Figure 3. Time series of total $HNO_3$ second derivative (blue, left y-axis) and of the temperature (red, right y-axis) at 30 hPa (top panel) and 70 hPa (bottom panel), in the region of potential vorticity lower than -10 x$10^{-5}$ K.m$^2$.kg$^{-1}$.s$^{-1}$. The red horizontal line corresponds to the 195 K temperature. The vertical dashed lines indicate the second derivative minimum in $HNO_3$ for each year. The corresponding dates (in bold, on the x-axis) and temperatures are also indicated. The time series of total $HNO_3$ second derivative (dashed blue) and of temperature at 50 hPa (grey) in the 70–90°S Eqlat band are also represented.

---

## Author Comment (AC3) · 15 Dec 2020

**Response to Gloria Manney and Michelle Santee**

We thank Gloria Manney and Michelle Santee for their extensive comments. Kindly find below our responses to each (quoted between []). We hope that our responses will clarify the main issues they have addressed. In particular, we hope that with the changes made, also in reply to the two anonymous reviewers, we have made more convincing that the IASI HNO$_3$ dataset has the potential to contribute to stratospheric studies in general, and to the time evolution of the polar processes in particular.

**General comment**
Throughout this manuscript, starting with its title, the term "denitrification" is taken to be synonymous with the uptake of gas-phase HNO3 through the formation of PSCs. Although not without precedent, this approach is contrary to common practice and may lead to confusion. Condensation of HNO3 in PSCs is usually referred to as "sequestration", while the term "denitrification" is usually reserved for the permanent removal of HNO3 from the lower stratosphere through the sedimentation of PSCs. In the absence of analysis of direct PSC measurements (e.g., from an instrument such as CALIOP), the occurrence of true denitrification can only be inferred from space-borne measurements of gaseous HNO3 when abundances do not rebound as PSCs dissipate at the end of winter, suggesting permanent removal. Thus the "drop temperature" derived in this study is indicative only of the onset of PSC formation, not the onset of denitrification, as is stated in numerous places in the paper.

We agree that, from IASI, we can only detect a "removal from the gas phase", caused by sequestration into particles with or without sedimentation. This misuse of the term "denitrification" was also highlighted by the two anonymous referees. Careful attention has been given in the manuscript to avoid abusive use of the term "denitrification". Hence, "onset of HNO$_3$ denitrification" has been changed to "the onset of HNO$_3$ depletion" in L.169 and where appropriate in the revised manuscript. The title has also been changed accordingly to: "Polar stratospheric HNO$_3$ depletion surveyed from a decadal dataset of IASI total columns".

**Specific comments**

[Abstract: L2: It is misleading (particularly for those who read only the abstract of the paper) to characterize the IASI HNO3 total columns as having "good vertical sensitivity". Indeed, this optimistic assessment is directly contradicted in Section 2, where IASI is stated to have "low vertical sensitivity ... with only one independent piece of information" (L76).]

As stated in the text, we here refer to "a good vertical sensitivity in the low and middle stratosphere", not to a good vertical resolution of the measurement. Note that HNO$_3$ vertical profiles are retrieved from IASI measurements, not simply total columns. Hence, even if the sensitivity covers the entire altitude range from the troposphere to the stratosphere with no clear decorrelation (DOFS~1) between the retrieved layers, it is shown in Ronsmans et al. (2016) that the highest sensitivity lies in the low-middle stratosphere, depending on latitude and season (from ~70 to 30 hPa within the cold Antarctic winter). This means that the variability in the measured total column is mainly representative of that layer. "low vertical sensitivity" in L76 has been changed to "low vertical resolution" to be more in line with the above.

We agree that the IASI sensitivity was insufficiently put forward in the text. We made it more explicit at several places in the revised manuscript; e.g. in Section 1: "IASI provides reliable total column measurements of HNO$_3$ characterized by a maximum sensitivity in the low-middle stratosphere around 50 hPa (20 km) during the dark Antarctic winter (Ronsmans et al., 2016;

2018) …" and in Section 2: "… the largest sensitivity of IASI in the region of interest, i.e. in the low and mid-stratosphere (from 70 to 30 hPa), where the $HNO_3$ abundance is the highest (Ronsmans et al., 2016).

[Introduction: L48-49: It should be made more clear that this is by no means an exhaustive list of spaceborne instruments that have measured stratospheric HNO3.]
The study of Santee et al. (1999) on MLS/UARS measurements has been added:

"Several satellite instruments measure stratospheric $HNO_3$ (e.g. MLS/Aura (e.g. Santee et al., 2007), MIPAS/ENVISAT (Piccolo 50 and Dudhia, 2007), ACE-FTS/SCISAT (Sheese et al., 2017) and SMR/Odin (Urban et al., 2009))."

[Section 2: The information provided about the IASI HNO3 retrieval, data quality, and data screening is insufficient. This information is critical to assessing the robustness of the reported results, and readers should not be forced to refer to previous papers to find it.]
The reader is here invited to refer to the figure 4 of Ronsmans et al. (2016) which illustrates the global distribution of the total retrieval error for $HNO_3$ (integrated over 5 to 35 km) separately for January (left) and July (right) over the period of the IASI measurements. The mid- and polar latitudes are characterized by low total retrieval errors of around ~3-5% - which corresponds to a reduction by a factor of 18-30 compared to the prior uncertainty (90%) and indicates a real gain of information – except above Antarctica during wintertime where the errors reach 25%. They are explained by (1) a weaker sensitivity (i.e. a larger smoothing error which represents in all cases the largest source of the retrieval error) above such cold surface (DOFS of ~0.95 within the dark Antarctic vortex – see figure 3 of Ronsmans et al., 2016) and by (2) a misrepresentation of the wavenumber-dependent surface emissivity above ice surface (Hurtmans et al., 2012). As also required by the two anonymous referees, this is now made more explicit in Section 2 of the revised manuscript:

 "The total columns are associated with a total retrieval error ranging from around 3% at mid- and polar latitudes to 25% above cold Antarctic surface during winter (due to a weaker sensitivity above very cold surface with a DOFS of ~0.95 and to an poor knowledge of the seasonally and wavenumber-dependent emissivity above ice surfaces which induces larger forward model errors), and a low bias (lower than 12%) in polar regions over the altitude range where the IASI sensitivity is largest, when compared to ground-based FTIR measurements (see Hurtmans et al., 2012; Ronsmans et al., 2016 for more details)."

Note also that similarly to these two previous studies, $HNO_3$ measurements characterized by a poor spectral fit or by a low information content (DOFS < 0.9) have been filtered out of this analysis. This is now clearly mentioned in Section 2 of the revised manuscript:

"Quality flags similar to those developed for $O_3$ in previous IASI studies (Wespes et al., 2017) were applied a posteriori to exclude data (i) with a corresponding poor spectral fit (e.g. based on quality flags rejecting biased or sloped residuals, fits with maximum number of iteration exceeded), (ii) with less reliability (e.g. based on quality flags rejecting suspect averaging kernels, data with less sensitivity characterized by a DOFS lower than 0.9) or (iii) with cloud contamination (defined by a fractional cloud cover larger than 25 %)."

[In later sections (e.g., L186, L225), errors in IASI retrievals arising from issues with emissivity above ice shelves are invoked to account for some dubious results, but no mention of these

poor-quality retrievals is made in the "Data" section, nor is it explained why quality-control measures fail to properly filter out these suspect data points.]
See our response to the above comment.

Bright land surface such as desert or ice might in some cases lead to poor $HNO_3$ retrievals due to a poor knowledge of the wavenumber-dependent emissivity above such surfaces, which can alter the retrieval by compensation effects (Wespes et al., 2009). FORLI relies on the monthly climatology of surface emissivity built by Zhou et al. (2011) from several years of IASI measurements on a 0.5x0.5 grid and for each 8461 IASI spectral channels when available, or on the MODIS climatology that is unfortunately restricted to only 12 channels in the IASI spectral range; see Hurtmans et al. (2012) for more details. Although wavenumber-dependent surface emissivity atlases are used in FORLI, it is clear that this parameter remains critical and causes poorer retrievals that, in some instances, pass the posterior filtering. The total $HNO_3$ columns over eastern Antarctica which show drop temperatures much above 195K might precisely be related to this. We have made this clear in Section 4.2 of the revised version:
"…emissivity features that are known to yield errors in the IASI retrievals. Indeed, bright land surface such as ice might in some cases lead to poor $HNO_3$ retrievals. Although wavenumber-dependent surface emissivity atlases are used in FORLI (Hurtmans et al., 2012), this parameter remains critical and causes poorer retrievals that, in some instances, pass through the series of quality filters and affect the drop temperature calculation."

[L78: 10 km can hardly be characterized as the "mid-stratosphere".]
It has been corrected:
"… in the low and mid-stratosphere (from ~70 to ~30 hPa),…"

[L84: "normal" has a specific statistical meaning and is not the appropriate word here.]
The reviewers are right; "normal" has been removed.

[L85-86: The validity of the analysis approach depends on the 50 hPa pressure surface and the 530 K isentropic surface being in very close proximity during Antarctic winter. This implicit assumption should be explicitly justified in the paper.]
Figure 1 below represents the figure 2 of the manuscript but for the temperature at 30 hPa (top panel) and 70 hPa (bottom panel) for the sake of comparison. As expected, the drop temperatures are the lowest when using the temperatures at 30 hPa. They vary from 185-195 K (~192K on average) at 30 hPa to 195-204 K (~198 K on average) at 70 hPa with values of ~189-202 K (~194 K on average) at 50 hPa.

As explained in the manuscript, the use of the 195 K at 50 hPa as single level for the analysis is justified by the fact that it corresponds best to the maximum of IASI vertical sensitivity during the polar night (see Figure 3 of Ronsmans et al. 2016 and responses to related comments above); another justification is found a posteriori by the consistency between the 195 K threshold temperature taken at 50 hPa and the onset of the strong total $HNO_3$ depletion seen by IASI, which matches the NAT development that occurs in June around that level. However, we fully agree that the $HNO_3$ abundances over a large part of the stratosphere (between 70 and 30 hPa) contribute to the total $HNO_3$ variations detected by IASI and that this inevitably affects the drop temperature calculation at 50 hPa. In order to address this issue and as also requested by referee #1, we have added in the manuscript the range of drop temperatures when calculated at these

two other pressure levels (from 185 K to 204 K); this indeed allows the reader to better judge on the uncertainty of the drop temperature at 50 hPa (189-202 K). The text in the revised manuscript is changed to:

"… Nevertheless, given the range of maximum IASI sensitivity to $HNO_3$ around 50 hPa, typically between 70 and 30 hPa (Ronsmans et al., 2016), the drop temperatures are also calculated at these two other pressure levels (not shown here) to estimate the uncertainty of the calculated drop temperature defined in this study at 50 hPa. The 30 hPa and 70 hPa drop temperatures range respectively over 185.7 K – 194.9 K and over 194.8 K – 203.7 K, with an average of 192.0 +/- 2.9 K and 198.0 +/- 3.2 K (1σ standard deviation) over the ten years of IASI. The average values at 30 hPa and 70 hPa fall within the 1σ standard deviation associated with the average drop temperature at 50 hPa. It is also worth noting the agreement between the drop temperatures and the NAT formation threshold at these two pressure levels ($T_{NAT}$ ~193 K at 30 hPa and ~197 K at 70 hPa) (Lambert et al., 2016)."

See comment here below for the justification of a single theta level (530 K) for the PV.

[L89-91: It is highly problematic to use a single theta level to distinguish inside from outside vortex regions for column measurements. This approach implicitly (and erroneously) assumes that the vortex does not tilt, shrink, or expand with height over the altitude range considered. A better approach would have been to check PV over a range of levels and discard measurements classified as outside the vortex at any one of those levels. A similar comment can be made concerning the use of a single pressure level for temperature. Again, it might have been better to use a range of T over the ~10–30 km layer where IASI has most sensitivity. Some attempt is made to justify the latter choice (using 195 K at 50 hPa) in Section 3 (L141-142) and Section 4 (L168-169), but the arguments are not convincing, as the authors themselves appear to recognize when they state (L188-189) "hence, the use of temperature at a single pressure level might be restrictive to some extent".]

Here again, the approach that we have followed was to select the levels that correspond best to the altitude of IASI maximum vertical sensitivity during the polar night (see Figure 3 of Ronsmans et al. 2016 and responses to related comments above). We agree, however, that considering PV over the range of the largest IASI sensitivity (from ~30 to ~70 hPa during the polar night) would allow the reader to better judge on the uncertainty of our approach. To that end, the figure 2 below compares the maps of PV at 475 K (~65 hPa), 530 K (~50 hPa) and 600 K (~30 hPa) over the southern latitudes averaged over the period 15 May – 15 July (period of drop temperatures detection inside the inner vortex core) for the year 2008. They show quite similar shape of the vortex over the altitude of maximum IASI sensitivity which, hence, has only small influence on our delimitation of the inner polar vortex (delimited by a PV value of $-10\times10^{-5}$K.m$^2$.kg$^{-1}$.s$^{-1}$ at 530 K) and, thus, on the detection of the drop temperature averaged inside that region (see Figure 2 of the manuscript). Note, furthermore, that our approach has no influence on the spatial distribution of the drop temperature illustrated in Fig.5 of the manuscript, which is independent of the PV.

See comment here above for the justification of the use of a single pressure level (50 hPa) for the temperature.

[Section 3: The definition of the three "regimes" in the T/HNO3 relationship seems arbitrary and not well justified. For example, R1 is defined to begin in April, but Fig. 1a shows that

HNO3 values start to increase rapidly and temperatures start to decrease rapidly in March (or even February, as noted in L117), not April. Only R2 encompasses a steep change in HNO3, but that regime also includes a lengthy period during which HNO3 remains nearly constant. It might have been better to break R2 into an "onset of PSC formation" phase and a "denitrification plateau" phase. Moreover, as defined in the paper, R2 extends through, not to, September as stated in L108. These problems are evident in the discussion in this section, as in some cases the behavior ascribed to one regime actually occurs in another.]

The definition of the three "regimes" in the T/HNO$_3$ relationship made here is actually based on changed in both HNO$_3$ and T, not only in HNO$_3$.

We did not stated in our manuscript that "HNO$_3$ values start to increase rapidly and temperatures start to decrease rapidly in March (or even February, as noted in L117), not April". In the manuscript, it is clearly stated in L117: "The plateau lasts until approximately February, where HNO$_3$ total column slowly starts increasing, reaching the April-May maximum in R1". Our statement specifically justified the start of R1 in April.

We changed "R2 extends from June to September" to "R2 extends from June to October" in L108.

[L102 and Fig. 1 caption: The red line in Fig. 1a is horizontal, not vertical, and Fig. 1b contains no such line – it is on Fig. 1c. Neither red line is defined in the caption.]
For Fig.1a: "horizontal" has been changed to "vertical".
Fig. 1b and 1c do contain a red vertical line.
The red horizontal or vertical lines are now mentioned in the caption of the revised manuscript.

[L102 and Fig. 1: 2011 was a particularly cold and long-lasting Antarctic winter, and thus it is arguably not representative. Some explanation for why that year was selected for highlighting in Fig. 1b is needed.]
As expected from figure 1c, any other year could have been chosen instead of the year 2011 to illustrate the HNO$_3$ total columns versus temperatures (at 50 hPa) histogram in figure 1b. It is now clearly mentioned in the revised manuscript:

"Similar histograms are observed for the ten years of IASI measurements (not shown)."

[L105-106: The contribution of confined descent inside the developing vortex bringing air rich in HNO3 from above into the domain where IASI is most sensitive has been ignored here – isn't descent also a factor leading to the observed high HNO3 total column values in early austral autumn?]
The domain where IASI is the most sensitive does actually cover the maximum HNO$_3$ concentrations, hence, the high HNO$_3$ total column values cannot be explained by the descent of HNO$_3$ rich air. However, in response to the two anonymous referees, the sentence has been rewritten as follows:
"These high HNO$_3$ levels result from low sunlight, preventing photodissociation, along with the heterogeneous hydrolysis of N$_2$O$_5$ to HNO$_3$ during autumn before the formation of polar stratospheric clouds (Keys et al., 1993; Santee et al., 1999; Urban et al., 2009; DeZafra et al., 2001). This period also corresponds to the onset of the deployment of the southern polar vortex which is characterized by strong diabatic descent with weak latitudinal mixing across its boundary, isolating polar HNO$_3$-rich air from lower latitudinal airmasses."

[L115-116: In addition to a lack of citations of earlier papers on renitrification of the lowermost stratosphere (LMS), this sentence is not a very clear expression of the fact that IASI is not sensitive to the LMS and hence renitrification has little impact on the observed evolution of total column HNO3.]

The renitrification at lower stratospheric layers was merely mentioned here and it was not meant to be extensively reviewed. To address the comment, Lambert et al. (2012) , which was already cited at several places of the manuscript has been added here. It is clearly stated in the revised version that a likely renitrification of the LMS could hardly be detected given the maximum sensitivity of IASI to $HNO_3$ at higher levels than those at which it occurs:

"The likely renitrification of the lowermost stratosphere (Braun et al., 2019; Lambert et al., 2012), where the $HNO_3$ concentrations and the IASI sensitivity to $HNO_3$ are lower (Ronsmans et al., 2016), cannot be inferred from the IASI measurements."

[L119-121: Why is 2010 highlighted in Fig. 1a (green line)? Other recent Antarctic winters were also disturbed with some minor SSW activity, e.g., 2012 and 2013. Did those episodes not affect the HNO3 distribution? Also, why does the green line show T at 20 hPa, when the other curves show T at 50 hPa? More explanation for why the authors chose to show this particular level for this particular year is needed.]

As explained in the text, 2010 is chosen because of its highest $HNO_3$ levels and highest temperatures within the Antarctic winter. No strong warming and related enhanced $HNO_3$ levels are observed from IASI for the years 2012 and 2013 (see Fig. 1a and Fig.4 of the manuscript). We have chosen to illustrate the temperature at 20 hPa for 2010 (dotted green line) in addition to the ones at 50 hPa (dashed lines) for each year simply because that level shows a distinct increase in temperature (cfr de Laat and van Weele, 2011) reflecting the presence of a SSW during the winter of 2010, while at 50 hPa, the increase in temperatures is smaller (dashed green line).

[Fig. 1c: In general this plot is not well explained or well motivated. By showing the position in temperature / HNO3 space of the bin with the maximum number of observations, important information about the range of those values on a given day is omitted. The ranges in Fig. 1b suggest that the values at a given time may span most of the HNO3 axis in Fig. 1c, rendering the curves shown less meaningful. In addition, it is stated (L127) that this figure highlights the interannual variability in total HNO3, but interannual variability is also clearly seen in panel (a), which is much easier to interpret. The discussion relates the picture in Fig. 1c to the three regimes, but since they are not marked on this panel, it cannot easily be examined without reference to Fig. 1a. It is therefore not obvious what additional value this figure brings to the paper.]

We agree that figure 1c does not bring additional information in comparison with the figures 1a and 1b; however, it is an original way to give insight into the $HNO_3$/temperature cycle and, in that respect, it nicely complements figure 1a. We would not be in favour of removing it.

Regarding the other comment, it is true that the daily range of $HNO_3$ values around those of highest occurrence is not represented in Fig. 1c but note that it does not correspond to the range of $HNO_3$ values in Fig.1b which cover 3 months of IASI measurements. Hence, we do not agree with the comments that "The ranges in Fig. 1b suggest that the values at a given time may span most of the $HNO_3$ axis in Fig. 1c, rendering the curves shown less meaningful". The daily

variability associated with the $HNO_3$ time series in the equivalent latitude bands can be found in Ronsmans et al. (2018).

In order to respond to the comment, the three regimes that were identified in Fig. 1a and Fig. 1b are now also indicated in Fig. 1c of the revised manuscript.

[L125: HNO3 columns are said to slowly increase as the T decreases over "February to May, i.e., R3 to R1". However, R3 is defined to start in October, and actually the slow increase in total HNO3 starts before February, arguably even as early as December.]
Here again we would like to stress that we did not only consider the change in $HNO_3$, but well the changes in both $HNO_3$ and temperature; $HNO_3$ columns do indeed increase as the temperature decrease over February to May but before February the $HNO_3$ levels increase as temperature also increase.

[L126: In the discussion of strong and rapid HNO3 depletion, "June (R1-R2)" should be "June-August (R2)".]
We indicate in the revised version: "… the strong and rapid $HNO_3$ depletion occurring in June (R2)"

[Section 4: Fig. 2 and its caption: More should be said about the agreement (or lack thereof) between the dashed and solid HNO3 and the grey and red T lines when they both exist. Some readers may question why the PV approach is used, given the gaps in those curves. Also, perhaps this is just an optical illusion, but the solid blue line appears to be thicker in some years (2011, 2014, 2016, 2017) than in the others. If that is the case, then that also needs to be explained. In the caption, the level to which the stated PV value pertains (presumably 530 K) should be specified.]
The PV approach is indeed preferred for the calculation of the drop temperatures and the corresponding dates because it better delimits the inner vortex core. The time series in the 70–90°S Eqlat band are only represented for consistency with Fig.1a. Even if the time series in the PV isocontour of $-10\times10^{-5}K.m^2.kg^{-1}.s^{-1}$ or in the 70–90°S Eqlat band are very close during the Antarctic winter, differences in the drop temperature calculation might be found.

Only one blue solid line is plotted, hence, its width is the same over the IASI period.

The potential temperature at which the PV is taken (530 K) is now mentioned in the caption of the revised manuscript.

[L155: It is not appropriate to characterize the total HNO3 depletion in the inner vortex as being the "coldest".]
Indeed a word was missing here. It has been corrected: "… the regions inside the inner polar vortex where the temperatures are the coldest and the total $HNO_3$ depletion occurs."

[L160: The wording in this sentence is garbled.]
It has been rewritten for clarity: "Note that the $HNO_3$ time series has been smoothed with a simple spline data interpolation function to avoid gaps in order to calculate the second derivative of $HNO_3$ total column with respect to time as the daily second-difference $HNO_3$ total column".

[L162-163: 23 is more than "a few" days.]
It has been changed to: "…within some days…"

[L174-179 and Fig. 3 caption: The description of the figure is confusing. It is stated in both in L174-175 and the caption that the vertical red dashed line indicates, at 90S, the 10-year average of the drop temperatures (191.1 K) calculated from the HNO3 second derivative time series in the area delimited by the −10×10-6 K.m2.kg-1.s-1 PV contour. It's not clear how a vertical line on a time series plot can represent a temperature value. Perhaps the authors meant to say the average date on which T dropped below the 195 K threshold at 90S? Moreover, the discussion above indicated that the value of 191.1 K was the average for the inner vortex (defined by either PV or EqL), not specifically at the South Pole (90S). In addition, the scale for the PV contour should be 10-5, not 10-6. Then in L176-177, it is stated that the "delay of 4-23 days between the maximum in total HNO3 and the start of the depletion is also visible" – but how is a range of values (which arises from different years) visible in a climatological plot?]

The red dashed vertical line indeed represents the average drop temperature of 194 K calculated in the area delimited by the −10×10-6 K.m2.kg-1.s-1 PV contour; the position of the line matches the temperature of 194.2 K at 90°S. We agree that the representation of the averaged drop temperature is not clear. We now represent one isocontour for the averaged drop temperature and two vertical lines that encompass the dates on which the drop temperature is calculated. The scale for the PV contour has been corrected. We now state in the revised version that:

"The delay of some days between the maximum in total $HNO_3$ and the start of the depletion (see Fig. 2) is also visible in Fig. 3a."

[Fig. 4: Very little discussion is devoted to this figure; it is merely noted (L177-178) that it shows the reproducibility of the IASI measurements of HNO3 depletion from year to year. Since Fig. 1 already makes this point, the added value of Fig. 4 is not clear.]

The figure 4 clearly illustrates the reproducibility, from year to year, of the edge of the collar $HNO_3$ region delimited by the PV isocontour of $−5×10^{-5} K.m^2.kg^{-1}.s^{-1}$ and of the region of the strong $HNO_3$ depletion delimited by the PV isocontour of $−10×10^{-5} K.m^2.kg^{-1}.s^{-1}$ taken at 50 hPa, the pressure level considered in this study to derive the drop temperatures. This cannot be inferred from Figure 1 and this is the main reason why we think that Figure 4 has to be kept.

[Fig. 5: How relevant is the PV contour averaged over the May to October period, when the dates of the onset of HNO3 depletion are May to June (or possibly July)? Why include August, September, and October in this average?]

We fully agree with that comment. Initially, the May-October period was chosen because it encompasses the dates of drop temperatures calculated in the region considered in Fig.5 (isocontour of $−8×10^{-5} K.m^2.kg^{-1}.s^{-1}$). However, outside the polar vortex (defined by an isocontour of $−10×10^{-5} K.m^2.kg^{-1}.s^{-1}$), drop temperatures are found much above the NAT formation temperature and they do not correspond to clear minima in the second derivative of $HNO_3$ total column with respect to time. Hence, considering that period for the PV contour is indeed not appropriate here.

We now represent, in the revised version, the PV contour over the 10 May to 15 July period that encompasses the dates of the onset of $HNO_3$ depletion inside the inner vortex core. Note that, on the contrary to the submitted version, we do not only consider the average of the PV over that period, but also the minima, which we find more representative of the drop temperature given the rapid displacement of the vortex: one bin can indeed be located inside the vortex one day and outside the vortex another day. Hence, that particular bin can be characterized by a depletion in $HNO_3$ with a specific drop temperature but an averaged PV larger than the value considered here to delimit the vortex core. The contour of $−10×10^-$

$^5$K.m$^2$.kg$^{-1}$.s$^{-1}$ based on the minimum PV encountered over the 10 May to 15 July period as well as the isocontours of 195 K at 50 hPa for the averaged temperatures and the minima over the same period are also now represented in the revised Fig.5 and the distribution of the drop temperatures is much better described and explained in the revised version:

"The averaged isocontour of 195 K encircles well the area of HNO$_3$ drop temperatures lower than 195 K, which means that the bins inside that area characterize airmasses that experience the NAT threshold temperature during a long time over the 10 May – 15 July period. That area encompasses the inner vortex core (delimited by the isocontour of $-10\times10^{-5}$K.m$^2$.kg$^{-1}$.s$^{-1}$ for the averaged PV), but is larger, and show pronounced minima (lower than -0.5 x10$^{14}$ molec.cm$^{-2}$.d$^{-2}$) in the second derivative of the HNO$_3$ total column with respect to time (not shown here), which indicate a strong and rapid HNO$_3$ depletion.
The area enclosed between the two isocontours of 195 K for the temperatures, the averaged one and the one for the minimum temperatures, show higher drop temperatures and weakest minima (larger than -0.5 x10$^{14}$ molec.cm$^{-2}$.d$^{-2}$) in the second derivative of the HNO$_3$ total column (not shown). That area is also enclosed by the isocontour of $-10\times10^{-5}$K.m$^2$.kg$^{-1}$.s$^{-1}$ for the minimum PV, meaning that the bins inside correspond, at least for one day over the 10 May – 15 July period, to airmasses located at the inner edge of the vortex and characterized by temperature lower than the NAT threshold temperature. The weakest minima in the second derivative of total HNO$_3$ (not shown) observed in that area indicate a weak and slow HNO$_3$ depletion and might be explained by a short period of the NAT threshold temperature experienced at the inner edge of the vortex. It could also reflect a mixing with strong HNO$_3$-depleted and colder airmasses from the inner vortex core. The mixing with these "already" depleted airmasses could also explained the higher drop temperatures detected in those bins. Finally, note also that these high drop temperatures are generally detected later (after the HNO$_3$ depletion occurs, i.e. after the 10 May – 15 July period considered here – not shown), which supports the transport, in those bins, of earlier HNO$_3$-depleted airmasses and the likely mixing at the edge of the vortex."

[L181: "the drop 50 hPa temperatures" should be "the 50 hPa drop temperatures".]
It has been corrected.

[L183: Technically, the isocontour represents –10, not ≤ –10.]
It has been corrected.

[L184-185: First, how does the range of dates corresponding to the onset of HNO3 depletion reported here – mid-May to early July – relate to that reported (L163) in connection with Fig. 2, which was 17 May to 10 June? Does the difference in these estimates arise because the former is based on averages in 1°×1° bins, whereas the latter is based on a vortex average within the PV contour? July seems rather late for the onset of PSC formation. Similarly, the range in 50 hPa drop T is quoted as 188.2 K to 196.6 K in L164, whereas here drop Ts vary over a wider range, from 180 to 210 K. The values at both extremes of this range are unrealistic. Indeed, the date and T ranges found in connection with Fig. 5 call into question the analysis method.]
Indeed, the differences between the range in drop temperatures and corresponding dates shown in Fig.2 and in Fig.5 are simply due to the average (over the whole area delimited by a PV contour in Fig.2 vs in 1°x1° bins within the PV contour).

See our response to comment [L186, L225] above about the extreme unrealistic values of drop temperature: The total HNO$_3$ columns over eastern Antarctica which show drop temperatures much above 195K might precisely be contaminated by strong surface emissivity features above ice; We have made this clear in Section 4.2 of the revised version:

"…emissivity features that are known to yield errors in the IASI retrievals. Indeed, bright land surface such as ice might in some cases lead to poor $HNO_3$ retrievals. Although wavenumber-dependent surface emissivity atlases are used in FORLI (Hurtmans et al., 2012), it is clear that this parameter remains critical and causes poorer retrievals that, in some instances, pass pass through the series of quality filters and affect the drop temperature calculation."

[L189-196: The questionable results derived from this analysis cannot be pinned on biases in the ERA-Interim data. The statement is made that "Reanalysis data sets are, indeed, known to feature large uncertainties", but the uncertainty in modern reanalysis temperatures (typically less than ~1 K) is by no means large enough to account for drop Ts as extreme as 180 and 210 K. The reliability of reanalysis temperatures in the polar lower stratosphere (including those from ERA-Interim) has been conclusively demonstrated in several recent papers, notably by Lawrence et al. [2018] and Lambert and Santee [2018]. Although both papers are cited here, their implications have apparently been overlooked.]

We fully agree with that remark that was also made by the referee #2. The discussion about the potential role of the uncertainty of the ECMWF reanalysis temperature on the drop temperature has been removed from the section, hence, this paragraph has been strongly revised accordingly:

"Biases in ECMWF reanalysis are too small for explaining the spatial variation in drop temperatures. Thanks to the assimilation of an advanced Tiros Operational Vertical Sounder (ATOVS) around 1998–2000 in reanalyses, to the better coverage of satellite instruments and to the use of global navigation satellite system (GNSS) radio occultation (RO) (Schreiner et al., 2007; Wang et al., 2007; Lambert and Santee, 2018; Lawrence et al., 2018), the uncertainties are reduced for several years. Comparisons of the ECMWF ERA Interim dataset used in this work with the COSMIC data (Lambert and Santee, 2018) found a small warm bias, with median differences around 0.5 K, reaching 0–0.25 K in the southernmost regions of the globe at ~68–21 hPa where PSCs form."

[L197-199: This sentence is confusing and its intended meaning is unclear. It appears to be comparing apples (the spatial variability in drop T seen in the maps in Fig.5) to oranges ("natural" variations in PSC nucleation T, TTE, and PSC formation mechanism). Perhaps the authors meant the spatial variability in those parameters (and not the values themselves), but that is not how the sentence is constructed. In any case, further discussion of comparisons of Fig. 5 with previously published results is warranted.]

We here simply link the range in drop temperatures with that in PSCs nucleation temperatures (explained by a series of parameters – atmospheric conditions, TTE, type of formation mechanisms), not the spatial variability. The sentence has been rewritten for clarity:

"Except above some parts of Antarctica which are prone to larger errors, the overall range in the drop 50 hPa temperature for total $HNO_3$, inside the isocontour for the 195 K temperature, typically extends from ~187 K to 195 K, which fall within the range of PSCs nucleation temperature at 50 hPa …".

Furthermore, nota also that comparing the distributions of drop temperatures from IASI with PSC information from CALIPSO or MIPAS remains difficult given the difference in spatial coverage and, most importantly, the highly variable distribution of PSC types temporally (daily) and spatially (e.g. Höpfner et al., 2006; Lambert et al., 2012).

[L199-200: A number of other satellite data sets have captured gas-phase HNO3 depletion (from both sequestration and denitrification) on similarly large scales.]

Indeed and numerous references about HNO$_3$ measurements in the polar regions during winter are mentioned in the manuscript where appropriate.

[Conclusions: L225-226: It is stated that "the range of drop temperatures is interestingly found in line with the PSCs nucleation temperature that is known, from previous studies, to strongly depend on a series a factors". In fact, the derived range (180–210 K) is so large that it is arguably not in line with previous work, and it is therefore difficult to see how the IASI total column HNO3 measurements provide added value (as stated in L203) to studies of Antarctic PSC formation and the interannual variability therein beyond that obtained from other satellite HNO3 datasets.]

Please refer to our response to comment (L186, L225) above about the impact of the misrepresentation of the wavenumber-dependent surface emissivity above ice surface on the drop temperature calculation with some extreme values. Except for these extrema, the range of drop temperature in indeed in line with the PSCs nucleation temperature. This is now clearly mentioned in this section of the revised manuscript:

"Except for extreme drop temperatures that were found from year to year and suspected to result from unfiltered poor quality retrievals in case of emissivity issues above ice, the range of drop temperatures is interestingly found in line with the PSCs nucleation temperature"

[L230-231: The statement that this paper represents "the first time that such a large satellite observational data set of stratospheric HNO3 concentrations is exploited to monitor the evolution HNO3 versus temperatures" is wholly unsupportable. In fact, there is a substantial body of literature on the relationship between HNO3 and temperature, including studies of long-term vertically resolved datasets. In particular, Lambert et al. [2016] (which is cited in a number of places in this manuscript, but only inpassing) examined 10 years of Aura MLS HNO3 in the Antarctic winter vortex and its relationship to T – including temperature history (a factor that has been largely ignored here) and T with respect to TICE – as well as PSC composition as determined by CALIOP. In general, discussion of how the current results fit into the context of the findings from Lambert et al. [2016] and other relevant prior studies is inadequate.]

We wanted to highlight here the unprecedented exceptional spatial and temporal sampling of IASI for HNO$_3$ and certainly did not want to oversell the novelty of HNO$_3$-temperature correlations. The sentence has been rewritten:

"We show in this study that the IASI dataset allows capturing the variability of stratospheric HNO$_3$ throughout the year (including the polar night) in the Antarctic. In that respect, it offers a new observational means to monitor the relation of HNO$_3$ to temperature and the related formation of PSCs."

[L233-234: More explanation of how HNO3 total column amounts could be used to inform PSC classification schemes is needed to justify this statement, especially given how spatially heterogeneous and layered PSCs have been shown to be.]

This sentence has been removed.

[Finally, in addition to the serious substantive issues enumerated above and in the formal reviews of the official referees, the manuscript suffers from the poor quality of the writing. If this paper were to be eventually accepted for publication, it would require extensive copy-editing to improve the English.]

We hope that with the changes made in the revised manuscript, which now also includes a comparison with MLS, G. Manney and M. Santee will not go against publication. A detailed reading of the paper has been done to correct the English linguistic/grammar mistakes.

[Figure]

Figure 1. Same as Figure 2 of the manuscript but for the temperature at 30 hPa (top panel) and 70 hPa (bottom panel).

[Figure]

Figure 2. Spatial distribution of PV (×10⁻⁵ K.m². kg⁻¹.s⁻¹) taken at three potential temperatures (475 K, 530 K and 600 K) over the range of the maximum IASI sensitivity, averaged over the period 15 May – 15 July for the year 2008. The blue lines represented the isocontours PV of −5.25×10⁻⁵ K.m². kg⁻¹.s⁻¹ (at 475 K), −10×10⁻⁵ K.m². kg⁻¹.s⁻¹ (at 530 K) and −19.4×10⁻⁵ K.m². kg⁻¹.s⁻¹ (at 600 K) averaged over the considered period.

---

## Referee Report (RR1)

**acpd-2020-347**

Referee comments on the revised version or on the author responses are in green type.

Referee comments on the original version are in black type.

Author responses are in blue type.

The revised manuscript includes a qualitative comparison with contemporaneous HNO3 measurements by MLS. This analysis should have been taken further to provide a more useful quantitative comparison with the IASI HNO3 column data. Additionally, incorporating CALIOP data would have enabled much better insight into the interannual variations of the drop temperatures and their relation to the distribution of PSC types. Overall, the revised manuscript does not put forward a compelling case for the scientific utility of the "drop temperatures". I do not find that the authors have addressed previously raised concerns in the on-line referee comments sufficently well to recommend publication. Detailed comments are given below.

Line numbers from here correspond to the revised version ...

=================================================================================

good vertical sensitivity in the mid-stratosphere (around 50 hPa),

The IASI HNO3 nadir measurements can not be considered as having "good vertical sensitivity" . Prospective data users need to know the vertical resolution of the measurements and that is conveyed by the standard practice of quoting the full width at half maximum (FWHM) of the averaging kernel. There is no reason not to do this for a nadir sounder e.g. Maddy and Barnett, Vertical Resolution Estimates in Version 5 of AIRS Operational Retrievals, IEEE Trans. Geosci. Remote Sens., 46, 8, 2375-2384, 2008.

=================================================================================

nitric acid trihydrate (NAT) formation temperature

For reasons explained elsewhere the "formation temperature" can be better expressed as an "existence thresold".

=================================================================================

Although the measured
HNO3 total column does not allow differentiating the uptake of HNO3 by different types of PSC particles
along the vertical profile, an average drop temperature of ˜194.2 +/- 3.8 K, consistent with the nitric acid
trihydrate (NAT) formation temperature (close to 195 K at 50 hPa)

The averaged "drop temperature" disregards the considerable interannual variability in the early stage formation of different types of Antarctic PSCs and the role played by the exposure of liquid PSCs to low temperatures in the formation of NAT i.e. many studies have shown that NAT is not uniquely constrained to nucleate at TNAT and some supersaturation is generally needed leading to a lower temperature for NAT formation (as in fact you discuss in the text L55-L75). Therefore, stating that the drop temperature is "consistent with TNAT", which implies that PSCs are mainly NAT forming at TNAT, is invalid. On line 28 the " " sign should be deleted since a specific value and its uncertainty is quoted.

=================================================================================

potential vorticity lower than -10x10-5 K.m2.kg-1.s-1

Some corresponding indication of the equivalent latitude range would be useful here.

=======================================================================

vorticity lower than -10x10-5 K.m2.kg-1.s-1. The spatial distribution and inter-annual variability of the
drop temperature are investigated and discussed in the context of previous PSCs studies.

However, the study presented here does not include any observed data on PSCs and is therefore not a "PSC study".

=======================================================================

profile of HNO3 similar to the limb sounders, IASI provides reliable total column measurements of
HNO3 characterized by a maximum sensitivity in the low-middle stratosphere around 50 hPa (20 km)
during the dark Antarctic winter (Ronsmans et al., 2016, 2018) where the PSCs cloud form (Voigt et al.,

Please give the fullwidth half-max (FWHM) of the vertical response in km and not just the height of maximum sensitivity.

=======================================================================

In order to expand on the comparisons against FTIR measurements which
is impossible during the polar night, Figure 1 (top panel) presents the time series of daily IASI total
HNO3 columns co-located with MLS VMR measurements within 2.5x.5 grid boxes at three pressure
levels (at 30, 50 and 70 hPa), averaged in the 70 - 90  S equivalent latitude band. Similar variations in
HNO3 are captured by the two instruments with an excellent agreement for the timing of the strong
HNO3 depletion within the inner vortex core. IASI HNO3 variations generally match well those of MLS
HNO3 in each latitude band (see Figure 1 bottom panel for the 50 - 70 S equivalent latitude band; the
other bands are not shown here).

=======================================================================

The 50 hPa drop temperatures are detected between 189.2 K and 202.8 K, with an average of
194.2 +/- 3.8 K (1 standard deviation) over the ten years. Knowing that TNAT can be higher or lower
depending on the atmospheric conditions and that NAT starts to nucleate from 2–4 K below TNAT (Pitts
et al., 2011; Hoyle et al., 2013; Lambert et al., 2016), the results here demonstrate the consistency
between the 50 hPa drop temperature,

The software bug that was fixed in the revised version has changed the drop temperatures such that the year with 202.8K (previously 190.6K) is a significant outlier since it lies 8K higher than the 10-year mean drop temperature and is almost as much above the assumed 50hPa TNAT (195K). Therefore it does not support the statement on L365-366 that the 10-year range "demonstrated the good consistency between the 50 hPa drop temperature and the PSCs formation temperatures in that altitude region".

=======================================================================

Note that the high extremes in the drop temperature, which are found in some case above
eastern Antarctica, should be considered with caution:. they correspond to specific regions above ice
surface with emissivity features that are known to yield errors in the IASI retrievals (Hurtmans et al.,
2012; Ronsmans et al., 2016). Indeed, bright land surface such as ice might in some cases lead to poor
HNO3 retrievals. Although wavenumber-dependent surface emissivity atlases are used in FORLI
(Hurtmans et al., 2012), this parameter remains critical and causes poorer retrievals that, in some
instances, pass through the series of quality filters and could affect the drop temperature calculation.
...
Except for extreme drop
temperatures (~210 K) that were found from year to year above eastern Antarctica and suspected to
result from unfiltered poor quality retrievals in case of emissivity issues above ice

It is not clear why this data quality problem has not been addressed in the revised submission. Measurements that are known to be bad must be screened out.

=====================================================================================
Except above some parts of Antarctica which are prone to larger retrieval errors, the overall range in the
drop 50 hPa temperature for total HNO3 inside the isocontour for the averaged temperature of 195 K,
typically extends from ˜187 K to ˜195 K, which falls within the range of PSCs nucleation temperature
at 50 hPa: from slightly below TNAT to around 3-4 K below the ice frost point - Tice - depending on
atmospheric conditions, on TTE and on the type of formation mechanisms (Pitts et al., 2011; Peter and
Grooß, 2012; Hoyle et al., 2013).
...
the range of drop
temperatures is interestingly found in line with the PSCs nucleation temperature

Why is the discussion in L302-338 and L367-376 limited to nucleation of NAT and ice PSCs with no mention of STS? There is no nucleation barrier to STS formation and it generally forms in advance of ice nucleation except possibly under very fast cooling e.g. in mountain waves. STS is not even mentioned in the paper after the introduction in L55-72.

Line numbers from here correspond to the original version ...

=====================================================================================
Major comments
[The description of the polar HNO3 variation presented in the paper is already well known from
numerous other studies.]

The purpose of this paper is to demonstrate the interest of IASI for HNO3 stratospheric studies
(Ronsmans et al., 2018) after having undergone a rigorous validation exercise (Ronsmans et al., 2016).
If limb measurements allows resolving the HNO3 profile in the stratosphere, the potential of IASI lies in
its exceptional spatial and temporal sampling. We demonstrate here that despite its limited vertical
resolution forcing us to consider one total column, the information content that actually lies in the low
and middle stratosphere offers potential to expand on previous polar stratospheric denitrification studies,
usually performed using limb sounder measurements, and to continue the long-term records of HNO3
started with the latter. We have tried in this paper not to repeat too much of our earlier work but some
duplication was unavoidable; in particular, with respect to vertical sensitivity and errors (these are two
aspects that referee1 finds in fact insufficiently described here).

=====================================================================================
[The lack of vertical resolution in the IASI HNO3 measurements severely limits the interpretation of the
results and precludes differentiation between denitrification and renitrification e.g. consider the effect of
the vertical integration through depleted higher layers overlaying lower enhanced layers.]

We understand that the referee sees this as a limitation. However, despite the lack of vertical resolution,
which is recognized in the paper and which forces us to consider total HNO3 columns, IASI is
characterized by a good sensitivity to HNO3 at specific levels, in particular, in the range between ˜70
hPa to ˜30 hPa in the southernmost latitude in winter and as such it provides an adequate means to investigate the stratospheric processes in the polar nights.
In order to justify this further, we would like to refer to the figure 3 (top and bottom panels) of Ronsmans
et al. (2016) that presents global distributions of the degrees of freedom for signal (DOFS, top panels)
and of the altitude of maximum sensitivity of IASI to the HNO3 profile (bottom panel), separately for
January (left) and July (right) 2011, when the strong HNO3 depletion occurs within the cold Antarctic
winter. It shows clearly that the altitude of maximum sensitivity of the total columns is invariant at
equatorial and tropical latitudes, whereas it varies with seasons at middle and polar latitudes. Above the
Antarctic, the altitude of maximum sensitivity varies between ˜9 km in summer and ˜22 km in winter.
The variations of the altitude of maximum sensitivity follow the altitude variations of maximum HNO3
concentrations.
We agree that the IASI sensitivity was insufficiently put forward in the text. We made it more explicit
at several places in the revised manuscript; e.g. in Section 1: "IASI provides reliable total column
measurements of HNO3 characterized by a maximum sensitivity in the low-middle stratosphere around
50 hPa (20 km) during the dark Antarctic winter (Ronsmans et al., 2016; 2018) . . . " and in Section 2:
". . . the largest sensitivity of IASI in the region of interest, i.e. in the low and mid-stratosphere (from 70
to 30 hPa), where the HNO3 abundance is the highest (Ronsmans et al., 2016)."

The response does not address the specific case example of where IASI views HNO3 depleted higher layers
that overlay lower enhanced layers. How does the IASI column HNO3 measurement change if the HNO3
is redistributed in the vertical coordinate by denitrification and renitrification? A further question would be
how does downwelling of higher values of HNO3 affect the HNO3 column?

===============================================================================
[Although the IASI HNO3 data has much better 2D horizontal resolution than any other measurement
this has not been developed as a tool to provide information beyond that of satellite instruments that
measure only along the orbit track.]

We do not fully agree. The determination of the drop temperature using the second derivative exploits
the large dataset of daily IASI measurements. Furthermore, the spatial distributions of the drop
temperature calculated at 50hPa, which are presented in the figure 5 of the manuscript, do actually take
advantage of the excellent spatial/temporal resolution of IASI to provide information throughout the
entire vortex and outside. This would probably not be feasible with other types of measurements.

As a further example of the 2D potential, could IASI be used to image the HNO3 field to show depletion
in the cold phases of mountains waves e.g. near the Palmer peninsula (similar to the wave structures seen
in AIRS brightness temperatures) or is that defeated by the vertical integration caused by the poor vertical
resolution?

===============================================================================
[CALIOP PSC information is available for the same time frame, why was this not used? Certainly, PSC
volumes vs time would be helpful in providing the underlying interannual variability of PSC types (NAT,
STS, ice) to compare with the resulting drop temperatures derived from IASI. Similarly, at least some
comparisons of the IASI HNO3 column with integrated column calculated from Aura MLS are
necesssary to establish the validity of the measurements in the most severely depleted inner vortex core.]

Thank you for this comment. It is certainly a good idea to use the CALIOP measurements in support but
this goes beyond the goal of this paper, which is to demonstrate the capability of IASI to measure HNO3
columns that are relevant for stratospheric studies. Using CALIOP PSC information and, in particular,
comparing the spatial distributions of IASI derived drop temperatures (Figure 5 of the revised paper)
with maps of CALIOP PSC would be very interesting in order to go a step further in the analyses of the
underlying HNO3 condensation processes, but it will be challenging and add significant complexity
given the high variability in the distribution of PSC types.
Regarding the second point on a comparison with MLS, we fully agree that this is highly relevant; it was
also a request of referee"#1. We provide here below a comparison with observations by MLS in three
equivalent latitude bands (see Figure 1). We would like to point out that we here compare total columns
measured by IASI with VMR measured by MLS at several pressure levels that cover the highest
sensitivity of IASI (at ˜50 hPa, ˜70 hPa and ˜30 hPa for the sake of the comparison). Hence, the
comparison of IASI columns with MLS measurements is mostly qualitative at this stage and differences
are expected for this reason. Note also that we have preferred comparing IASI HNO3 columns with VMR
measured by MLS at specific levels instead of integrated columns calculated from MLS, given the
difference in the sensitivity profile between IASI and MLS, the non-negligible IASI sensitivity to HNO3
in the troposphere where MLS does not measure HNO3 etc, which makes the integrated columns from
IASI vs MLS not directly comparable. It should be pointed out finally that part of the differences between
IASI and MLS are likely due to the different number of co-located data within the 2.5"deg x 2.5"deg grid cells
considered here for the comparison, with a much larger number of observations for IASI (through the
quality filtering) than for MLS.
Despite this, the comparison shows similar spatial and seasonal variations between IASI total HNO3
columns and MLS VMR between ˜70 and 30 hPa in the different latitude bands, in particular, in the
southernmost equivalent latitudes (see top panel). The strong HNO3 depletion is well captured by both
IASI and MLS measurements with a perfect match for the onset of the depletion. It further supports the
good sensitivity of IASI to HNO3 in the range of these pressure levels, justifying the methodology used
in this study.
The cross-comparison with MLS is indeed insightful and gives further credit on the IASI observations
during the polar night. That comparison figure between IASI and MLS has therefore been included in
Section 2 of the revised manuscript and the text was changed to:

"In order to expand on the comparisons against FTIR measurements which is impossible during the polar
night, Figure 1 (top panel) presents the time series of daily IASI total HNO3 columns co-located with
MLS VMR measurements within 2.5x2.5 grid boxes at three pressure levels (at 30, 50 and 70 hPa),
averaged in the 70 "deg S–90 "deg S equivalent latitude band. Similar variations in HNO3 are captured by the
instruments with an excellent agreement for the timing of the strong HNO3 depletion within the inner
vortex core. IASI HNO3 variations generally match well those of MLS HNO3 in each latitude band (see
Figure 1 bottom panel for the 50 "deg S–70 "deg S equivalent latitude band; the other bands are not shown h

"CALIOP measurements ... this goes beyond the goal of this paper, which is to demonstrate the capability of
IASI to measure HNO3 columns that are relevant for stratospheric studies". That goal was largely achieved
already by Ronsmans et al (2016) and published in Atmos. Meas. Tech. This paper is under review for
Atmos. Chem. Phys. and should relate more to a science investigation rather than a technical description.
The comparisons with MLS are a welcome improvement, but unfortunately fall short of the analysis I was

===================================================================================

expecting. Surely the tropospheric contribution of HNO3 to the IASI column is not all that much (you could estimate the effect to confirm). I expected the MLS profile to be integrated with the IASI response function for a more direct comparison. That would facilitate a quantitative interpretation of the differences in the variation of the column data from the two instruments.

[Regarding the sensitivity of the IASI column HNO3 measurements, I suggest presenting a few examples
of vertical HNO3 profiles (from a model or data), ranging from non-depleted to extreme depletion with
calculations of the corresponding calculated integrated IASI column. This would help to indicate the
sensitivity of the column measurement to changes in the vertical distribution of HNO3 ... i.e. generate
profiles of the change in the IASI column HNO3 wrt the actual change in HNO3 at a level, j,
d(column)/d(HNO3)j.]

This is an example of information reported in earlier work and that we have tried not to repeat extensively
here. To summarize, the validation study of Ronsmans et al. (2016) provides a complete characterization
of the IASI HNO3 retrievals: it shows example of vertical HNO3 profiles along with the total retrieval
error, the a apriori profiles and associated averaging kernels profiles (d(HNO3ret)i/d(HNO3true)j), along
with the total column averaging kernel (d(columnret)/d(HNO3true)j) and the sensitivity profile
(d(HNO3ret)i/d(columntrue)), were already given in Figures 1 and 2 of that study. Note that the averaging
kernel profile describes how the true state changes the estimate at a specific altitude, i.e. how the retrieval
smooths the true profile. The sum of the elements of an averaging kernel characterizing the retrieval at
a specific altitude returns the sensitivity of the retrieval at that altitude, i.e. to which extent the retrieval
at that specific altitude comes from the spectral measurement rather than the apriori, while the sum of
the averaging kernels indicates how the true state at a specific altitude changes the retrieved total column,
i.e. the altitude to which the retrieved total column is mainly sensitive/representative.
Figure 3 (top and bottom panels) of Ronsmans et al. (2016) further presents the global distributions of
the degrees of freedom for signal (DOFS, top panels) and of the altitude of maximum sensitivity of the
retrieval to the HNO3 profile (bottom panel), separately for January (left) and July (right) 2011, when
the strong HNO3 depletion occurs within the cold Antarctic winter. It clearly shows that above the
Antarctic, the altitude of maximum sensitivity varies between ˜9 km in summer and ˜22 km in winter
(˜ 50 hPa) on average.
To address the comment of the referee without repeating too much of the earlier results, we have
carefully verified the manuscript with regard to unclear or incomplete statements about vertical
sensitivity. The following has been added in Section 1: "IASI provides reliable total column
measurements of HNO3 with a maximum sensitivity in the low-middle stratosphere around 50 hPa (20
km) during the dark Antarctic winter (Ronsmans et al., 2016; 2018) . . . " and in Section 2: ". . . the largest
sensitivity of IASI in the region of interest, i.e. in the low and mid-stratosphere (from 70 to 30 hPa),
where the HNO3 abundance is the highest (Ronsmans et al., 2016).
In order to convince the referee that IASI measurements capture the expected variations of HNO3 within
the polar night, we provide in Figure 1 below examples of vertical HNO3 profiles retrieved within the
dark Antarctic vortex (above Arrival Height) and outside the vortex (above Lauder). The retrieved
profiles are shown along with their associated total retrieval error and averaging kernels (the total column
AvK and the so-called "sensitivity profile" are also represented). Above Arrival Height during the dark
Antarctic winter, we clearly see depleted HNO3 levels in the low and mid-stratosphere and the altitude of maximum sensitivity at around 30 hPa. At Lauder on the contrary, HNO3 levels larger than the a priori
are observed in the stratosphere with a larger range of maximum sensitivity.

I also wanted to see specific depleted vs non-depleted cases (one with a re-nitrification layer would be good also) generated along with the simulated IASI columns and the calculated columns. I suggest that the figure provided on the averaging kernels etc could be added to a supplemental material section with a description tailored to the cases studied here in addition to just referring readers to a prior publication.

=====================================================================================
[L10: 191K is also consistent with STS temperatures (192 K) and is actually closer than TNAT (195 K)]

Indeed but as stated in the manuscript: "... recent observational and modelling studies have shown that
HNO3 starts to condense in early PSC season in liquid NAT mixtures well above Tice (~4 K below TNAT,
close to TSTS)...". The NAT nucleation temperature at 50 hPa range from slightly below TNAT to around
3-4 K below Tice, depending on atmospheric conditions, on TTE and on the type of formation
mechanisms (Pitts et al., 2011; Peter and GrooS, 2012; Hoyle et al., 2013).
Note that in replying to referee#1 we have identified a bug for the automatically detection of the drop
temperature, as well as for the detection of the corresponding dates in the figure 2 of the manuscript. It
has been corrected. The position of the drop temperatures does now perfectly match the yearly minima
of the total HNO3 second derivative. An average drop temperature over the ten years of IASI of 194.2
+/- 3.8 K is now calculated, which is even closer to TNAT.
Finally, as requested by referee #1, we also now clearly mention in Section 4.1 of the manuscript the
range of drop temperatures when calculated at two other pressure levels to better judge on the uncertainty
of the drop temperature at 50 hPa (see Figure 3 here below):
... Nevertheless, given the range of maximum IASI sensitivity to HNO3 around 50 hPa, typically
between 70 and 30 hPa (Ronsmans et al., 2016), the drop temperatures are also calculated at these two
other pressure levels (not shown here) to estimate the uncertainty of the calculated drop temperature
defined in this study at 50 hPa. The 30 hPa and 70 hPa drop temperatures range respectively over 185.7
K – 194.9 K and over 194.8 K – 203.7 K, with an average of 192.0 +/- 2.9 K and 198.0 +/- 3.2 K (1
standard deviation) over the ten years of IASI. The average values at 30 hPa and 70 hPa fall within the
1 standard deviation associated with the average drop temperature at 50 hPa. It is also worth noting the
agreement between the drop temperatures and the NAT formation threshold
(TNAT ~193 K at 30 hPa and ~197 K at 70 hPa) (Lambert et al., 2016)."

CALIOP PSC data (Pitts et al 2013, doi:10.5194/acp-13-2975-2013) have been used to show that different PSC types exist in different temperature regimes, with ice PSCs detected close to the frost point, STS follows the expected equilibrium curve and NAT exhibits two preferred mode below the NAT existence temperature. The analysis presented here is not constrained by the simultaneous presence of known PSC types and in fact there may not even be any PSCs in the atmospheric path sampled. Therefore, it is too simplistic to compare the drop temperatures to TNAT. The proximity of the 10-year mean drop temperatures to TNAT does not constitute a validation as is claimed here. Individual years could be expected to show a variation in drop temperature because of interannual atmospheric differences. For instance, the years domimated by STS should necessarily show lower drop temperature than years dominated by NAT. The highest drop temperatures are far above PSC temperatures (e.g. 202.8K at 50hPa in one particular year) and deserve more scrutiny and should be investigated thoroughly. Interannual comparisons of the drop temperature may benefit from using (T-Tice) as the temperature coordinate (rather than absolute temperature) as this removes variations due to changes in H2O partial pressure (see Fig 2 of Pitts et (2013)). There is a fundamental problem with making an assessment of the potential future scientific utility of the drop temperatures when they have only been evaluated in the absence of knowledge of the different types of PSCs present.

===============================================================================
[L18: add more recent references e.g. Peter and Gross (2012). L28: Much more has been done in the
past decade with MIPAS and CALIOP that should be referenced]

Thank you for this suggestion. Peter and GrooS (2012) was cited elsewhere in the manuscript but has
been added here as well. Note that the goal of the introduction is not to provide an exhaustive list of all
studies related to the PSC thermodynamics. Several general reference papers are cited and we have
decided to put more focus here on HNO3.

===============================================================================
[L59: This section should explain what is meant by "maximum sensitivity" etc.]

See our responses to the second major comment and specific comments above.

===============================================================================
[L79: Information on the data quality for IASI HNO3 is poor. Is the value of bias and uncertainty the
same for depleted and non-depleted conditions?]

The reader is here invited to refer to the figure 4 of Ronsmans et al. (2016) which illustrates the global
distribution of the total retrieval error for HNO3 (integrated over 5 to 35 km) separately for January (left)
and July (right) over the period of the IASI measurements. The mid- and polar latitudes are characterized
by low total retrieval errors of around ~3-5% - which corresponds to a reduction by a factor of 18-30
compared to the prior uncertainty (90%) and indicates a real gain of information – except above
Antarctica during wintertime where the errors reach 25%. They are explained by (1) a weaker sensitivity
(i.e. a larger smoothing error which represents in all cases the largest source of the retrieval error) above
such cold surface (DOFS of ~0.95 within the dark Antarctic vortex – see figure 3 of Ronsmans et al.,
2016) and by (2) a poor knowledge of the wavenumber-dependent surface emissivity above ice surface,
which also varies in time (Hurtmans et al., 2012). ). This is made more explicit in Section 2 of the revised
manuscript:
"The total columns are associated with a total retrieval error ranging from around 3% at mid- and polar
latitudes to 25% above cold Antarctic surface during winter (due to a weaker sensitivity above very cold
surface with a DOFS of ~0.95 and to an poor knowledge of the seasonally and wavenumber-dependent
emissivity above ice surfaces which induces larger forward model errors), and a low bias (lower than
12%) in polar regions over the altitude range where the IASI sensitivity is the largest, when compared
to ground-based FTIR measurements (see Hurtmans et al., 2012; Ronsmans et al., 2016)."

The response does not address the specific case of whether there are differences in bias and uncertainty for depleted and non-depleted conditions.

===============================================================================
[L82: Yet, problems with the retrievals because of cloud contamination seem to remain even after the
25% cloud fraction filter is applied.]

We do not understand the referee comment here. In this section of the manuscript, we only describe the
quality flags used in our analysis.

Even after all the quality controls are applied there are apparently still cases with poor retrievals that could
be removed.

========================================================================================
[L83: Cloud contamination? Tropospheric cloud only or also thick ice PSCs?]

The clouds that have most impact are clearly tropospheric water clouds. Cirrus clouds or PSCs are mostly
transparent in the IR; thick cirrus however show up in the longwave part of the IASI spectrum, below
900 cm-1. We have added "tropospheric cloud contamination" in the text.
Note that the threshold of 25 % cloud cover was carefully chosen after a series of tests, which have
shown that these scenes could be treated as cloud-free without significant impact on the retrievals
(Hurtmans et al., 2012).

Thick ice PSCs have been detected by AIRS, TOVS HIRS2 and AVHRR (see
Stajner et al. and refs therein,
https://doi.org/10.1029/2007GL029415). Do these have an effect on the
HNO3 retrieved by IASI?

========================================================================================
[L102: Why was 2011 chosen?]

As expected from figure 1c, any other year could have been chosen instead of the year 2011 to illustrate
the HNO3 total columns versus temperatures (at 50 hPa) histogram in figure 1b. It is now clearly
mentioned in the revised manuscript:
"Similar histograms are observed for the ten years of IASI measurements (not shown)."

========================================================================================
[L106: Heterogeneous hydrolysis of N2O5 requires aerosol particles. So this process starts with cold
binary aerosols (i.e. sulfates) before the formation of STS?]

Indeed, previous studies have shown enhanced HNO3 columns during autumn in Antarctica and have
attributed them to decreasing sunlight and conversion of N2O5 to HNO3 by the reaction of N2O5 with
background aerosols, before the formation of polar stratospheric clouds (e.g. Keys et al., Nature, 1993).
At these temperatures, the conversion may occur on binary sulfuric aerosols.
The sentence has been rewritten as follows:
"These high HNO3 levels result from low sunlight, preventing photodissociation, along with the
heterogeneous hydrolysis of N2O5 to HNO3 during autumn before the formation of polar stratospheric
clouds (Keys et al., 1993; Santee et al., 1999; Urban et al., 2009; DeZafra et al., 2001). This period also
corresponds to the onset of the deployment of the southern polar vortex which is characterized by strong diabatic descent with weak latitudinal mixing across its boundary, isolating polar HNO3-rich air from
lower latitudinal airmasses."
[L129: The onset of depletion seems to start when the temperatures fall substantially below 190K from
inspection of Fig 1(c) and quite far below the red line marked at 195K.]
The onset of HNO3 depletion starts in June at around 190K, which is in agreement with figure 1a.

====================================================================================
[L136-137: Why are two temperatures (180 and 185 K) quoted for 30hPa? Why is the actual value from
Fig1(c) (I estimate this as about 188K) for the 50hPa temperature not given in L129?]

The sentence has been rewritten for clarity:
"The results (not shown here) exhibit a similar HNO3-temperature behaviour at the different levels with,
as expected, lower and larger temperatures in R2, respectively, at 30 hPa (down to 180 K) and at 70 hPa
145 (down to 185K), but still below the NAT formation threshold at these pressure levels (TNAT =193 K
at 30 hPa and 197 K at 70 hPa) (Lambert et al., 2016)."

====================================================================================
[L138: "characterized by" seems the wrong description for the chance occurrence that the maximum
sensitivity of IASI HNO3 falls in the same altitude range as the PSCs.]

Changed to: "… the altitude range of maximum IASI sensitivity to HNO3 (see Section 2) is characterized
by temperatures that are below the NAT formation threshold at these pressure levels, enabling the PSCs
formation and the denitrification process."

====================================================================================
[L139-146: This section rather seems to belong in 711 the conclusions.]

L150-154 of the revised manuscript has been moved to the conclusions.

====================================================================================
[L148: Clearly this does not "go beyond the vertically integrated view" since the column HNO3 is all
that is available. It could be reworded as "To identify the spatial and temporal variability of the column
HNO3 …"]

Corrected as suggested.
[L165-169: Denitrification is the term used to describe the permanent removal of some HNO3 from the
gas phase by sedimentation of PSCs. Sequestration is the term used to describe the uptake of HNO3
from the gas phase into PSCs. Denitrification by STS is a lengthy process compared to NAT since the
smaller STS particles sediment slowly. STS can (and frequently does) form without the prior nucleation
of NAT. IASI alone cannot discriminate between these processes and it should not be assumed that what
is observed is the "onset of HNO3 denitrification".]
We thank the referee for this remark. We are of course aware of the definition of the so-called
"denitrification". We agree that, from IASI, we can only measure a "removal from the gas phase", caused by sequestration into particles with or without sedimentation. Careful attention has now been made in
the manuscript to avoid abusive use of the term "denitrification". Hence, "onset of HNO3 denitrification"
has been changed to "the onset of HNO3 depletion" in L.169 and where appropriated in the revised
manuscript and he title has also been changed accordingly to:
"Polar stratospheric HNO3 depletion surveyed from a decadal dataset of IASI total columns".

===========================================================================================
[L185-187: 210K is much too high for PSC formation, but could possibly be NAT that is in process of
melting? If these are observed over ocean then they warrant further investigation. However, why are
specific regions with emissivity features not flagged as such? They should be discarded rather than
"used with caution".]

Bright land surface such as desert or ice might in some cases lead to poor HNO3 retrievals due to a poor
knowledge of the wavenumber-dependent emissivity above such surfaces, which can alter the retrieval
by compensation effects (Wespes et al., 2009). FORLI relies on the monthly climatology of surface
emissivity built by Zhou et al. (2011) from several years of IASI measurements on a 0.5x0.5 grid and
for each 8461 IASI spectral channels when available, or on the MODIS climatology that is unfortunately
restricted to only 12 channels in the IASI spectral range; see Hurtmans et al. (2012) for more details.
Although wavenumber-dependent surface emissivity atlases are used in FORLI, it is clear that this
parameter remains critical and causes poorer retrievals that, in some instances, pass the posterior
filtering. The total HNO3 columns over eastern Antarctica which show drop temperatures much above
195K might precisely be related to this. We have made this clear in Section 4.2 of the revised version:
"...emissivity features that are known to yield errors in the IASI retrievals. Indeed, bright land surface
such as ice might in some cases lead to poor HNO3 retrievals. Although wavenumber-dependent surface
emissivity atlases are used in FORLI (Hurtmans et al., 2012), this parameter remains critical and causes
poorer retrievals that, in some instances, pass through the series of quality filters and could affect the
drop temperature calculation."
We refer on the good agreement with MLS (suggested by the referee) to underline the potentiality of
IASI to detect the HNO3 variations as well within the Antarctic winter (see general comment and Figure
1 here below).

What is the fraction of data that is affected by surface emissivity?

===========================================================================================
[L189: Modern reanalysis temperatures (e.g. ERA-I) do not "feature large uncertainties" large enough
to account for a 195K to 210K shift. L195-L201: The limitations of the reanalysis temperatures seems
to be an accuracy of better than 1K and clearly this in no way limits the derivation 760 of the "50hPa drop
temperature" which simply necessitates finding the 50hPa reanalysis temperature that corresponds to
the second derivative wrt time minimum in column HNO3.]

===========================================================================================
We agree with the referee's comment; the discussion about the potential role of the uncertainty of the
ECMWF reanalysis temperature on the drop temperature has been removed from the section, hence, this
paragraph has been strongly revised accordingly:

"... while biases in ECMWF reanalysis are too small for explaining the spatial variation in drop
temperatures. Thanks to the assimilation of an advanced Tiros Operational Vertical Sounder (ATOVS)
around 1998–2000 in reanalyses, to the better coverage of satellite instruments and to the use of global
navigation satellite system (GNSS) radio occultation (RO) (Schreiner et al., 2007; Wang et al., 2007;
Lambert and Santee, 2018; Lawrence et al., 2018), the uncertainties have been vastly reduced.
Comparisons of the ECMWF ERA Interim dataset used in this work with the COSMIC data (Lambert
and Santee, 2018) found a small warm bias, with median differences around 0.5 K, reaching 0–0.25 K
in the southernmost regions of the globe at ˜68–21 hPa where PSCs form."

====================================================================================================
[What is meant by "spatial variability"? The plots in Fig 5 show the spatial distribution of the drop
temperature over a number of years but what variability is being considered? Interannual? Why have
these spatial maps of drop temperatures not been compared with published maps of PSC types made by
CALIOP or MIPAS. Wouldn't some correlation be expected according to the arguments made here? i.e.
NAT PSCs at the higher temperature e.g. the highest temperatures (orange) appear downstream of the
Palmar Peninsula in the "NAT ring" structure described by Hopfner et al (2006).]

Corrected: "Figure 5 shows the spatial variability" .. "Figure 5 shows the spatial distribution".
We do not understand the referee's comment here. Figure 5 of the manuscript shows the spatial
distribution of the drop temperature calculated inside a region enclosed by an isocontour PV of -8x10-5
K.m2.kg-1.s-1, which, hence, encircles a region larger than the inner vortex core (see Figures 3 and 4 of
the manuscript). The drop temperatures much above the NAT formation temperature, which are mostly
found outside the averaged isocontour PV of -10x10-5 K.m2.kg-1.s-1, do not correspond to high minima
(¿-0.5 x1014 molec.cm-2.d-2) in the second derivative of HNO3 total column with respect to time. We
cannot argue that it corresponds to the NAT belt of Höpfner et al. (2006) downstream of the Antarctic
Peninsula, which was enclosed inside the region of the NAT threshold temperature; the highest drop
temperatures from IASI are found on the contrary outside the isocontour of the NAT threshold
temperature (see figure 5 of the revised manuscript). In addition, comparing the distributions of drop
temperatures from IASI with PSC information from CALIPSO/MIPAS remain difficult given the
difference in spatial coverage and, most importantly, the highly variable distribution of PSC types and
of the NAT belt, temporally (daily) and spatially (Höpfner et al., 2006; Lambert et al., 2012).
Finally, in response to G. Manney and M. Santee, the contour of -10 x 10-5K.m2.kg-1.s-1 based on the
minimum PV encountered at 50 hPa over the 10 May to 15 July period as well as the isocontours of 195
K at 50 hPa for the averaged temperatures and the minima over the same period are also now represented
in the revised Fig.5 and the distribution of the drop temperatures is much better described and explained
in the revised version:
"The averaged isocontour of 195 K encircles well the area of HNO3 drop temperatures lower than 195
K, which means that the bins inside that area characterize airmasses that experience the NAT threshold
temperature during a long time over the 10 May – 15 July period. That area encompasses the inner vortex
core (delimited by the isocontour of -10 x 10-5K.m2.kg-1.s-1 for the averaged PV) and show pronounced
minima (lower than -0.5 x1014 molec.cm-2.d-2) in the second derivative of the HNO3 total column with
respect to time (not shown here), which indicate a strong and rapid HNO3 depletion.
The area enclosed between the two isocontours of 195 K for the temperatures, the averaged one and the
one for the minimum temperatures, show higher drop temperatures and weakest minima (larger than -
0.5 x1014 molec.cm-2.d-2) in the second derivative of the HNO3 total column (not shown). That area is also enclosed by the isocontour of -10 x 10-5K.m2.kg-1.s-1 for the minimum PV, meaning that the bins
inside correspond, at least for one day over the 10 May - 15 July period, to airmasses located at the inner
edge of the vortex and characterized by temperature lower than the NAT threshold temperature. The
weakest minima in the second derivative of total HNO3 (not shown) observed in that area indicate a
weak and slow HNO3 depletion and might be explained by a short period of the NAT threshold
temperature experienced at the inner edge of the vortex. It could also reflect a mixing with strong HNO3-
depleted and colder airmasses from the inner vortex core. The mixing with these "already" depleted
airmasses could also explained the higher drop temperatures detected in those bins. Finally, note also
that these high drop temperatures are generally detected later (after the HNO3 depletion occurs, i.e. after
the 10 May – 15 July period considered here – not shown), which supports the transport, in those bins,
of earlier HNO3-depleted airmasses and the likely mixing at the edge of the vortex."

===============================================================================
[L205: Nothing has been presented that demonstrates PSC occurrence. For that you would need to
compare to actual data on PSCs from CALIOP and/or MIPAS.]

Corrected: "PSCs occurrence" .. "NAT formation temperature"

===============================================================================
[L224: Again, the suspect data should be discarded because of the detrimental impact on the scientific
analysis. Also, if you cannot manage to work out and apply adequate quality control to your own data
then you have no reason to expect anyone else to do so.]

See our response to comment [L185-187] above.

===============================================================================
[L230: "To the best of our knowledge, it is the first time that such a large satellite observational data set
of stratospheric HNO3 concentrations is exploited to monitor the evolution HNO3 versus temperatures"
In fact you cite several papers that have done exactly this, but let's take the one published over two
decades ago by Santee et al (1999) titled "Six years of UARS Microwave Limb Sounder HNO3
observations : Seasonal, interhemispheric, and interannual variations in the lower stratosphere".
https://doi.org/10.1029/1998JD100089. Not only does this paper compare HNO3 with UKMO
temperatures we are referred to a more complete paper on this topic on p8241 ... "The correlation of the
HNO3 behavior with temperature during this time period, and its implications for PSC phase and
composition, is explored in detail by Santee et al (1998). I noticed that the outside edge of the "HNO3
collar region" at 465K was defined by these authors as inside the 0.25 x E-4 K m2 kg-1 s-1 PV contour.
This seems at odds with the 1E-4 value that is used for the second derivative minimum calculation in
this paper and seemingly places the boundary quite far equatorward. Santee et al (1998) also includes a
description of the heterogeneous hydration of N2O5 that would be helpful in response to the question
above on L106.]

===============================================================================
We here simply refer to the unprecedented potential of IASI in terms of its exceptional spatial and
temporal sampling. Ronsmans et al. (2018) also referred to the IASI dataset and correlations with
temperature were done but in a lesser extent. In order to avoid overselling, the sentence has been rewritten:
"We show in this study that the IASI dataset allows capturing the variability of stratospheric HNO3
throughout the year (including the polar night) in the Antarctic. In that respect, it offers a new
observational means to monitor the relation of HNO3 to temperature and the related formation of PSCs."
In this study, we use the PV fields taken from the ECMWF ERA Interim Reanalysis dataset at the
potential temperature of 530 K (corresponding to ˜20 km where the IASI sensitivity to HNO3 is the
highest), while Santee et al. (1998) considered 465K. We clearly see from Figures 3a and 4 of the
manuscript that PV contours at -0.5e-4 K m2 kg-1 s-1 and at -0.8e-4 K m2 kg-1 s-858 1 encompass the so
called HNO3 collar region. The PV value of -1e-4 K m2 kg-1 s-1 is used in this study to calculate the
drop temperature based on the second derivative minimum as it clearly encompass the regions inside the
inner polar vortex (see Figure 3a and 4 of the manuscript).

===================================================================================
[L231: "It could constitute a new accurate climatological parameter that could be inserted in the PSCs
classification schemes." The analysis presented does not support this statement. Specifically, how could
the HNO3 column amount be used in a classification scheme?]

This sentence has been removed.

The conclusion of the paper is that ability to monitor the polar atmosphere over several decades with current
and planed IASI instruments "will provide an unprecedented long-term dataset of HNO3 total columns".
The drop temperature is defined as the 50hPa temperature corresponding to the greatest rate of decline of
the column HNO3 with respect to time. However, even with a record now extending over a decade the
scientific utility of this dataset has not been demonstrated.

---

## Referee Report (RR2)

**Review of "Polar stratospheric nitric acid depletion surveyed from a decadal dataset of IASI total columns" by Ronsmans et al.**

This manuscript analyzes 10 years (2008–2017) of IASI $HNO_3$ total column measurements in conjunction with ERA-Interim temperatures to characterize the onset of PSC formation in the Antarctic lower stratospheric vortex.  The high-density horizontal sampling afforded by IASI is very valuable, and the novel approach of using the minimum in the second derivative of the $HNO_3$ total column with respect to time to identify the onset of $HNO_3$ uptake into PSCs is useful.  Although I was a co-author of a Short Comment posted on the originally submitted manuscript, I was not one of the referees for it, nor have I read any of the previous revised manuscripts.  However, I have now been asked to serve as an official reviewer of the current draft, so I am reading the paper for the first time in that formal capacity.  Hence, despite the fact that the manuscript has already been revised twice in response to previous comments, some issues are being raised now for the first time.  Moreover, some of the concerns expressed in the original reviews have not yet been adequately addressed, and new issues have been introduced through the revision process.  Thus, although the manuscript has been much improved, I feel that further corrections and clarifications, as detailed below, are necessary before the paper can be accepted for publication.

Below both major substantive issues and minor points of clarification, wording suggestions, and grammar / typo corrections are listed together for each Section in sequential order.

Respectfully,
Michelle Santee

Abstract:
- L19: in the mid-stratosphere --> in the lower-to-mid stratosphere; also, delete "causal"
- L23: This wording is very awkward; I suggest: evolution of the pair $HNO_3$-temperature --> evolution of $HNO_3$ together with temperature
- L24: The meaning of "in the cycle of IASI" is not clear – do the authors mean "annual cycle"?
- L27-28: delete "differentiating"; add "to be differentiated" after "profile"
- L28: Two different values for the average drop temperature are given in the text on p. 6 (L265 and L290).  These values need to be reconciled and the correct one quoted here.
- L32-33: It is a bit of an exaggeration to say that "this paper highlights the capability of IASI to monitor the long-term evolution of polar stratospheric composition and processes involved in the depletion of stratospheric $O_3$".  It would be more accurate to state that "this paper highlights … evolution of polar stratospheric $HNO_3$, a key player in the processes involved …".

Introduction:
- In a number of places in the presentation of background material (e.g., L39, L42, L45, L51), a few citations are given for very well-established concepts, but many other equally suitable papers could have been cited instead of or in addition to the ones listed.  Obviously not all relevant papers can be referenced for these points, but "e.g." should be added in these lines to avoid giving readers the impression that the selected references are the only appropriate ones.

- In addition, the recently published PSC review paper by Tritscher et al. (Rev. Geophys., 2021) covers all of this background material and should also be cited in several places in this section.
- L44-45: Technically, denitrification delays only the reformation of $ClONO_2$ – $HNO_3$ is not required for HCl production, thus it is not quite correct to say "chlorine reservoirs" here.
- L46-47: PSCs surface --> PSC surfaces; PSCs particle --> PSC particles
- L47-48: three 3different --> three different
- L56: delete comma after "($T_{ice}$)"
- L60: A reference should be provided for the point that NAT nucleation was thought to require temperatures below $T_{ice}$.
- L73: the PSCs formation --> PSC formation
- L74: add a comma after "2019)"
- L74-75: This sentence ("The influence of $HNO_3$ in modulating $O_3$ abundances in the stratosphere is furthermore underrepresented in CCMs") is out of place and potentially confusing for many readers, who will naturally assume that it has something to do with PSC processes in the lower stratosphere as it comes at the end of a long paragraph discussing nothing but PSCs. Kvissel et al. (2012), however, describe $HNO_3$ enhancements between 10 and 1 hPa induced by energetic particle precipitation. It's not clear why this sentence has been included here; perhaps it is meant to provide motivation for this study, but I doubt that IASI total column measurements with maximum sensitivity around 50 hPa could shed much light on modest (< 6 ppbv) $HNO_3$ enhancements in the middle and upper stratosphere. In fact, this issue is not explored in the manuscript. Thus this sentence should be deleted.
- L77: measure --> have measured
- L79-82: This poorly composed sentence is grammatically incorrect and hard to interpret. I'm not sure what is meant by "follow their formation mechanisms". I suggest alternative wording, but I may have misunderstood the intent: Spaceborne instruments such as the CALIOP/CALIPSO lidar and MIPAS/Envisat measuring in the infrared are capable of detecting and classifying PSC types, allowing their formation mechanisms to be investigated (Lambert et al., 2016; Pitts et al., 2018; Spang et al., 2018, and references therein); these satellite data complement in situ measurements (Voigt et al., 2005) and ground-based lidar (Snels et al., 2019).
- L84: the PSCs formation --> PSC formation; Urban 55 et al. --> Urban et al.
- L86: depends --> depend
- L92: similar to the limb --> similar to that from the limb
- L94: where the PSCs cloud form --> where PSCs form
- L95: 10-years --> 10-year
- L96: for providing --> to provide

Section 2:
- L104: embarked on --> onboard ("embarked" is not the right word)
- L117: FWHM --> full width at half maximum (FWHM)
- Figure 1 caption:
  - L446: molec.cm$^{-2}$/molec.cm$^{-2}$ --> molec.cm$^{-2}$/molec.cm$^{-2}$
  - L446: after "molec.cm$^{-2}$/molec.cm$^{-2}$" add "; colored lines, with the altitude of each kernel represented by the colored dots)"
  - L447: Height --> Heights; 169,40 --> 169.40

- L122: Height --> Heights
- L125: Height --> Heights
- L126-127: Above (in L119) it was stated that the largest sensitivity of IASI is from ~70 to ~30 hPa, and here it says "the altitude of maximum sensitivity (at around 30 hPa for this case)". This is confusing because the grey line in the Arrival Heights panel of Fig. 1 depicting the "sensitivity profile" peaks at ~75 hPa, not 30 hPa. I believe that it is the total column averaging kernel, not the "sensitivity profile", that determines the altitude of maximum sensitivity? The authors should take into consideration that not all readers of this paper will be experts on IASI data or will have read Ronsmans et al. (2016). Please clarify the meaning of the "sensitivity profile" (grey curve) and the total column averaging kernel (black curve) and how they relate to the altitude where IASI provides the most information on $HNO_3$.
- L127: At Lauder on the contrary, --> In contrast, at Lauder
- L128: "larger range of maximum sensitivity" – again, is this conveyed by the "sensitivity profile" (grey curve) or the total column averaging kernel (black curve)?
- L129-130: The statement "from around 3% at … polar latitudes to 25% above the cold Antarctic surface during winter" is confusing, because of course the Antarctic is at polar latitudes. Please be more precise in the wording here.
- L130: DOFS --> degrees of freedom for signal (DOFS)
- L132: "lower than 12%" is ambiguous here; assuming that the values are biased low by more (not less) than 12%, this should be rewritten as "low bias (exceeding 12%)"
- L135: FTIR measurements which is not possible during --> FTIR measurements, which cannot be made during
- Figure 2 caption:
  - L454: It should be explicitly stated here (not only in the text) that the MLS total column estimates were obtained by extending the MLS partial stratospheric column values using the FORLI-HNO3 a priori information.
  - L454: 2.5 x 2.5 --> 2.5° × 2.5°
  - L454: middle --> bottom
  - L455: Figure 2 does not include a panel showing 30°–50°S EqL
- L137: 2.5 x 2.5 --> 2.5° × 2.5°
- L139: The meaning of "the averaging kernels … were considered" will not necessarily be clear to all readers. It would be better to state explicitly that the IASI averaging kernels were applied to the co-located MLS profiles.
- L140-141: "column profiles" is an oxymoron. Please specify the MLS retrieval pressures over which the partial stratospheric column was calculated. Thus, rather than "… grids, then converted into column profiles. They were also extended down to the surface by considering the …", it would be better to rephrase along the lines of "… grids, and partial stratospheric columns above xxx hPa were calculated. MLS total columns were then estimated using the …".
- L155-165: In their Short Comment, Manney & Santee pointed out: "It is highly problematic to use a single theta level to distinguish inside from outside vortex regions for column measurements. This approach implicitly (and erroneously) assumes that the vortex does not tilt, shrink, or expand with height over the altitude range considered. A better approach would have been to check PV over a range of levels and discard measurements classified as outside the vortex at any one of those levels." In my opinion, the authors have not adequately

addressed this point.  I understand that 530 K falls in the region of the atmosphere where IASI has maximum sensitivity to $HNO_3$.  Nevertheless, it is simply not credible that changes in the size, shape, or location of the vortex over the 30–70 hPa domain primarily covered by the measurements had no effect on the results on any day in the 10-year IASI record.  At this point, I am not suggesting that the authors redo their analyses, but I would like to see added to the manuscript an explicit acknowledgment of the fact that relying on the determination of the vortex boundary on a single potential temperature surface for interpretation of total column measurements inevitably introduces some uncertainty into the results because it fails to account for the possibility of changes in the size, shape, or location of the vortex with altitude.

- L158: closing ")" missing after "2016)" – it should be "2016))"
- L158: "starts a few degrees or slightly below" is awkward and confusing – "slightly" could mean "a few degrees".  It would be clearer to say: "starts within a few degrees below"
- L161: It is not quite correct to say "to identify the PSCs-containing regions" – regions with temperatures below the threshold do not necessarily contain PSCs.  It would be better to say "to identify regions of potential PSC existence".

Section 3:
- L170: delete the comma after "S)"
- L170: This wording – "over the whole period of measurements (2008–2017)" – seems to imply that IASI has taken no data since 2017, which I do not believe is the authors' intention.
- L172: 2018) where the contribution of the PSCs into --> 2018), where the contribution of PSCs to
- L173: delete "here"
- L175-176: The wording "along the $HNO_3$/temperature cycle" is not clear.  I think the authors mean "within the $HNO_3$/temperature annual cycle".  Or maybe "during" rather than "within"
- L184: "R1 in Figures 3a and b" – since the regimes are clearly labeled in the panels of Figure 3, I'm not sure that this statement is needed.  Moreover, the label "R1" is also used in Figure 3c.
- L189: deployment --> development ("deployment" is not the right word); also, add a comma after "vortex"
- L190-191: lower latitudinal airmasses --> lower-latitude airmasses
- L193: A problem with the definition of R2 is that, as shown later in the paper, the onset of strong $HNO_3$ depletion actually begins in mid-May in most years.
- L193: add commas after "R2" and "October"
- L195: the $HNO_3$ total columns average below --> average $HNO_3$ total columns are below
- L198-199: To avoid the potential for confusion with SSWs: Despite the stratosphere warming with 50 hPa temperatures up to 240 K --> Despite 50 hPa temperatures increasing up to 240 K
- L202: by the PSCs sedimentation --> by sedimentation of PSCs
- L203: add "e.g." to the list of references
- L203-204: add commas after "2012)" and "2016)"
- L204-205: The meaning of "can hardly be inferred" is ambiguous.  I expect that the lack of sensitivity of IASI total column $HNO_3$ to the LMS precludes detection of renitrification from those measurements.  Thus: can hardly --> cannot
- L206: where --> when
- L210: occurs --> occurred

- L212: HNO$_3$ total columns in 2010 were higher in September as well as in July and August.
- L219: add "and" after "(R2),"
- L220: Based on Figure 3a, the plateau of low HNO$_3$ abundances begins in July, not August.
- Figure 3c:
  - Figure 3a clearly shows that the "strong and rapid HNO$_3$ depletion" (as stated also in L219) occurs mainly during June, so why is the steep drop in HNO$_3$ in Figure 3c labeled "Jun-Aug"?
  - I do not find the regime markers (R1, R2, R3) on this panel particularly helpful – these labels are essentially floating in semi-arbitrary positions on the plot and convey no real meaning.
  - It is stated that this panel "highlights the interannual variability in total HNO$_3$" (L221). But interannual variability is much easier to interpret in Figure 3a. For example, the anomalous behavior in July–September 2010 so evident in panel (a) does not stand out in panel (c). In fact, I would argue that it is not necessary to color-code the lines by year in panel (c), as the details of individual years are better seen in panel (a) anyway. Just having 10 separate lines would still illustrate the interannual spread even without distinguishing the specific years.
  - If the lines are not color-coded by year, that would allow them to be color-coded in a different manner. For example, 12 different colors could be used to indicate the portions of the curves corresponding to each month. This would allow the interannual variability in different months to be compared at a glance. In addition, different line styles (e.g., solid, dashed, dotted) could be used to differentiate the three regimes. Reformulating the plot along these lines would make this panel much more useful than it currently is.
  - Please add minor tick marks on the axes (x and y) of all of the panels in Figure 3.
- L221-222: The two parts of this sentence ("highlights the interannual variability in total HNO$_3$" and "shows a strong consistency in the onset of the depletion between each year") seem contradictory. If the behavior is consistent from year to year, then interannual variability is small. The sentence should be re-written using more careful language. The wording is also awkward: shows a strong consistency --> is very consistent; between each year --> every year.
- L223-224: Given the span of PSCs formation over a large range of altitudes --> Given that PSC formation spans a large range of altitudes
- L224: et al., 150 2006 --> et al., 2006
- L225: that of maximum IASI sensitivity to HNO$_3$ --> that IASI has maximum sensitivity to HNO$_3$
- L229: larger temperatures --> higher temperatures
- L230: The corresponding R2 temperature at 50 hPa to which the ~180 K value at 30 hPa and the ~185 K value at 70 hPa are being compared is not clear in this discussion.
- L234: enabling the PSCs formation --> enabling PSC formation
- L234-235: The "onset of the strong total HNO$_3$ depletion" in Figure 3c clearly occurs more than 5 K below 195 K (i.e., well to the left of the vertical red line). So I do not see how this statement about the consistency between 195 K and the onset of HNO$_3$ depletion is supportable. Perhaps the authors are accounting for the fact that HNO$_3$ starts to condense at temperatures 2–4 K below $T_{NAT}$, but (a) if so they need to state that explicitly, and (b) the difference in Figure 3c is larger than 4 K.

Section 4:
- L251: areas --> area; also, the fact that the PV value specified is for 530 K should be stated
- L252: regions --> region

- L253: total HNO$_3$ depletion occurs --> the largest depletion of total HNO$_3$ occurs
- L258: I have no idea what "as the daily second-difference HNO$_3$ total column" means. The rest of the sentence makes sense, so perhaps this part could simply be deleted. Otherwise, if it is supposed to convey important information, then it has to be rephrased for clarity.
- Figure 4 and its caption:
  - Why is the red line at 195 K dashed? Similar lines in Figure 3 were solid, which would be a better choice here too as the temperature time series is also shown as a red dashed curve.
  - L482: temperature --> 50 hPa temperature
- L262: around the 195 K threshold --> around the time that temperatures drop below the 195 K threshold
- L262: "some days" suggests to the reader "a few days", whereas 23 days is more than 3 weeks. Thus: within some days --> within a few days to a few weeks
- L265-266: The drop temperature in 2014 does stand out to some degree, but nevertheless I do not think that it can simply be excluded from the IASI-mission average just because it is a bit of an outlier. Strong justification is needed to exclude any individual year from the climatological mean, otherwise the authors risk being perceived as "cherry-picking" their results.
- L269-270: It is very good to remind readers of the meaning of the term "drop temperature", but this definition should come earlier in this section since the term has already been used above (L264, L266).
- L270: PSCs formation temperature --> PSC existence temperature
- L271: could reflect the preponderance by one --> could reflect variations in the preponderance of one
- L276-278: The average drop temperatures for 30 and 70 hPa appear to have been calculated over the full 10-year record, which supports my contention that 2014 should not be excluded from the 50-hPa average drop temperature calculation.
- L284: zonal distribution --> climatological zonal distribution
- L285: Does Figure 5 use geographic or equivalent latitude? If the former, the difference from other figures (which use EqL) needs to be explained, but in any case it should be made clear.
- L287: It is stated that one isocontour of 50 hPa temperature is overlaid, but actually two are shown in Figure 5.
- L289: lines indicates --> line indicates
- L290: The average drop temperature is given here as 194.2 ± 3.8 K, but above in L265 it was stated to be 194.1 ± 2.8 K. These values need to be reconciled.
- L290: rate in --> rate of
- L291: delete "of some days"
- Figure 5 and its caption:
  - Figure 5b is referred to in L284, but there is no further discussion of this panel in the text. The relationship between IASI total column HNO$_3$ and 50 hPa temperature has already been investigated in connection with Figures 3 and 4; moreover, Figure 5a includes two contours of 50 hPa temperature. Thus, in the absence of specific discussion about it in the text, Figure 5b seems superfluous and should be deleted.
  - L493: As noted above for the main text, whether this is geographic or equivalent latitude needs to be made clear.

- o L496: The meaning of "The vertical grey dashed lines encompass the period of the second derivative minima" is unclear – the steepest rate of decrease occurs on a particular date each year, not over a prolonged period. What I believe the authors are trying to say is that the grey lines mark the earliest and latest dates for the drop temperature in the 10-year record, but this statement needs to be written more clearly.
  - o Panel 5b: the date labels (which are nearly illegible) for the overlaid lines indicate that the earliest drop temperature date is 12 May, whereas in L263 it is stated to be 11 May.
- L295: The PV contours are specified at 530 K, not 50 hPa.
- Figure 6 and its caption:
  - o In the case of Figure 6, panel (b) is not even mentioned in the text. It should be deleted.
  - o The x-axis date labels are illegible.
  - o L501: Again, whether this is geographic or equivalent latitude needs to be made clear.
- L302-305: I'm a little confused about how the −8 × $10^{-5}$ $Km^2kg^{-1}s^{-1}$ contour of PV comes into the analysis in this section. It is a bit jarring to state in one sentence that the figure examines the region delimited by the −8 PV contour and then in the next sentence characterize the −10 PV contour as delimiting "the region of interest". I presume that the authors want to investigate a region larger than that of the strong depletion in total $HNO_3$ (encircled by the −10 PV contour) while excluding the collar region (−5 PV contour), but that rationale should be stated explicitly.
- Figure 7:
  - o Similar maps of the corresponding dates are not shown, but they would be a useful addition and would help to clarify some points in the discussion, as noted below.
  - o The dark green color for the −10 × $10^{-5}$ $Km^2kg^{-1}s^{-1}$ PV contour does not show up well (at least not on my monitor). A brighter green would work better.
  - o Why are some of the temperature contours not closed?
  - o The fonts for the years in the map titles in the top row look odd.
- L303: delete the comma after "$s^{-1}$"
- L303-307: Switching back and forth between temperature and PV makes these lines harder to follow. It would be better to discuss both the averaged and the minimum PV contours together and then move on to the two temperature contours.
- L306: The PV contour is specified at 530 K, not 50 hPa.
- L307: Add a comma after "period".
- L308-309: I have several comments on this part of the analysis:
  - o The range of 50 hPa drop temperatures found here is considerably broader than that found in connection with Figure 4 in Section 4.1 (~180–210 K vs ~189–203 K), as is the range of corresponding dates (mid-May to mid-July vs mid-May to early June). These differences between the results based on vortex averages and those based on 1°×1° bins should be commented on in the text. Do the two approaches agree in terms of which years show generally lower/higher drop temperatures or earlier/later dates? The discrepancies are of concern because the only advantage that the IASI $HNO_3$ total column measurements bring over vertically resolved $HNO_3$ data sets (for which vortex averages can be calculated) is their dense horizontal sampling.
  - o The high extremes in the drop temperature are attributed to issues with the retrievals over eastern Antarctica that are not fully screened out by quality control measures. But Figure 7 shows that unrealistically high drop temperatures are not confined to eastern Antarctica.

- As mentioned above, maps of the corresponding dates would be illuminating. What is the spatial distribution of late dates vs early dates? Do those patterns match the variations in drop temperatures, or do they bear no relation to the temperatures?
- The authors need to directly confront in the paper the fact that dates as late as mid-July for the onset of $HNO_3$ depletion in the Antarctic are even more implausible than drop temperatures as high as 210 K. These findings will likely cause many readers to dismiss their analysis methodology and/or the use of IASI data for studying PSCs.
- L311: surface --> surfaces
- L313: surface --> surfaces
- L318: Some years have sizeable regions with drop temperatures below 195 K that are well outside the averaged 195 K contour. Hence, encircles well --> encircles fairly well.
- L319: bins inside that area characterize air masses --> bins inside that area include air masses
- L322: show --> shows
- L325: show --> shows
- L327: is also enclosed --> is also typically enclosed
- L330-331: The weakest minima in the second derivative of total $HNO_3$ (not shown) observed in that area indicate a weak and slow --> The fact that the weakest minima in the second derivative of total $HNO_3$ (not shown) are observed in that area indicates a weak and slow
- L333-334: Although I think it is good to point out the possible impact of mixing on these results, it should also be acknowledged that previous studies have shown that in the Antarctic mixing between the edge region and the vortex core is generally weak (e.g., Roscoe et al., JGR 2012).
- L333: reflect a mixing with strong --> reflect mixing with strongly
- L334: The mixing --> Mixing; explained --> explain
- L335-336: I'm confused by this sentence. The drop temperature is defined by the onset of $HNO_3$ depletion, so how can it be that high drop temperatures are detected "after the $HNO_3$ depletion occurs"?
- L338-339: What is meant by "the range of maximum sensitivity of IASI to $HNO_3$"? Elsewhere in the manuscript, it is stated that IASI has the largest sensitivity to $HNO_3$ in the 30–70 hPa range, but how is that altitude information relevant to the spatial variations in the maps of Figure 7?
- L339-346: The second half of the sentence in L339 ("while biases …") starts a new discussion on reanalysis temperature and is followed by several related sentences, so it should be a separate sentence (not starting with "while"). In addition, it is not clear to the reader why all of the detail presented in the following sentences is really necessary. It would be better to either end the paragraph by stating explicitly that ERA-I temperature biases of the magnitude noted in these lines could not possibly account for the large range of calculated drop temperatures, or simply delete some of the details.
- L339: for explaining --> to explain
- L344: just to be clear, add "in modern reanalyses" after "reduced".
- L348: drop 50 hPa temperature --> 50 hPa drop temperature; delete the comma after "195 K"
- L349: PSCs nucleation --> PSC nucleation
- L351: on the type of formation mechanisms --> on the specific formation mechanism (i.e., the type of PSC developing)
- L353: coverage of IASI that allows capturing the rapid and critical depletion phase --> coverage of IASI, which allows the rapid and critical depletion phase to be captured in detail

Section 5:

- L357: columns dataset --> column dataset
- L358: since other IASI instruments are mentioned later: Metop --> Metop-A
- L363-364: I find this wording unclear.  I suggest: level over a range where --> level, which lies in the range where; process occur --> processes occur
- L367: delete "various"
- L368: delete "and described along the cycle" (this wording is confusing and unnecessary)
- L369: delete "at play"
- L370: Only Antarctica is considered here, thus: in the poles --> over Antarctica
- L370: As mentioned earlier when this regime was defined, R2 starts in June but the uptake of HNO$_3$ into PSCs starts in mid-May, as shown in this paper and in previous studies.
- L371-372: PSCs nucleation --> PSC nucleation
- L372: between each year --> from year to year
- L372: R3 is actually defined (L198, Figure 3a) to begin in October, not November.
- L373: until March --> through March
- L374: PSCs sedimentation at --> PSC sedimentation to
- L376: found particularly --> found to be particularly
- L377: condensation to --> condensation into
- L379: 2.8 --> 2.8 K; also, as noted previously, the inconsistency in the average drop temperature values given in the text (L265 and L290) needs to be fixed and the correct one quoted here.
- L380: As noted above in Section 4.1, I do not think that the omission of 2014 from the climatological average is justified.
- L381: demonstrated --> demonstrate; PSCs formation --> PSC formation
- L384: PV at 50 hPa --> PV at 530 K
- L386: highest minima --> lowest minima
- L388: "from year to year" is not the right phrase; perhaps the authors mean "in some years"
- L388: As mentioned earlier, not all of the unrealistically high drop temperatures were calculated over eastern Antarctica
- L390: found in line --> found to be in line; PSCs nucleation --> PSC nucleation
- L395-396: It likely results --> These likely result; a mixing --> mixing
- L399: over the whole polar regions --> over the whole Antarctic region
- L401-403: I do not see how the authors could make the statement that the IASI dataset offers a new observational means to monitor the relation of HNO$_3$ to temperature and PSC formation because it can make measurements in darkness.  It is certainly understandable that they want to tout IASI's excellent spatial coverage and its potential for a long record.  Those are indeed very valuable contributions.  But it is simply not acceptable to ignore decades of HNO$_3$ measurements made "throughout the year (including the polar night)" by numerous satellite instruments (e.g., LIMS, UARS MLS, Aura MLS, CLAES, MIPAS, SMR, ILAS, SMILES).
- L401-402: delete "capturing"; add "to be captured" at the end of the sentence
- L404: allow to investigate --> allow investigation of

---

## Referee Report (RR3)

**Re-review of "Polar stratospheric nitric acid depletion surveyed from a decadal dataset of IASI total columns" by Ronsmans et al.**

The manuscript has been substantially revised in response to referee comments. In general, the authors have done a good job in responding to the points raised by the reviewers, and the manuscript has been improved. However, a few new issues have been introduced through the revision process. In addition, while some of the concerns noted below have just arisen, others were present in earlier drafts but escaped my attention; they have become more obvious now that most of the significant issues pointed out previously have been remedied. Section 4 in particular requires further clarification in several places.

Respectfully,
Michelle Santee

Substantive issues and minor points of clarification, wording suggestions, and grammar / typo corrections are listed together for each section in sequential order through the manuscript.

Abstract & Introduction:
- L31: It would be better to spell out "equivalent latitude" in the abstract. By the way, this abbreviation is capitalized inconsistently throughout the manuscript ("Eqlat", "eqlat").
- L48: particles type --> particle type
- L78: This sentence should be rearranged to avoid giving the impression that each of these instruments made measurements for decades: "Over the last few decades, several satellite instruments have measured stratospheric $HNO_3$ (e.g., …".
- L85: the HNO3 --> HNO3
- L88: a series of --> several
- L98: dependence to --> dependence on

Section 2:
- L122-123, 131: The discussion in these lines about "the expected variations of $HNO_3$ within the polar night", "$HNO_3$ profiles retrieved within the dark Antarctic vortex", and "during the dark Antarctic winter, we clearly see depleted $HNO_3$" could be misinterpreted to suggest that $HNO_3$ has strong diurnal variations, when in fact the largest changes in $HNO_3$ inside the Antarctic winter polar vortex are driven by PSC formation and thus temperature (not directly by sunlight per se). This should be clarified.
- L129: to which extent --> the extent to which
- L132 and Fig. 1 caption: It is slightly confusing that the text refers to values of the total column averaging kernel of ~1 but the plot shows values of 0.1. The statement "(divided by 10)" in the caption (L496) could be interpreted to apply only to the grey sensitivity profiles; it should be made clear that it refers to the black curves as well.
- L134-141: This sentence, which spans 8 lines, is difficult to read. In particular, having such a long and complicated parenthetical statement makes the sentence hard to parse. I suggest rearranging and breaking it up into two or three sentences. In addition, some additional

punctuation would make the statement currently in parentheses easier to read, so I suggest adding commas after "surface" (L136), "0.95" (L137), and "surfaces" (L138).

- L145: profiles –> profile
- L146: The term "VMR" has not been defined; since it is not used again in the manuscript, it would be better to simply say "mixing ratio" here.

Section 3:

- L185: (1) The statement that similar histograms are observed for "the 10 years of IASI measurements" is ambiguous and could be interpreted to imply that IASI made only 10 years of measurements. I suggest instead: "Similar histograms are observed for the other years in the 10-year study period." (2) The horizontal line in Fig. 3a is not red (it appears to be grey).
- L187: The three identified regimes correspond to --> The three regimes we identified are:
- L189-196: It should be acknowledged that $HNO_3$ starts to decrease by the end of this regime. That point is now stated in the description of R2, but it needs to be mentioned for R1 as well.
- L198-200: As noted above, I appreciate that the description of R2 now makes it clear that $HNO_3$ depletion typically begins by mid-May, but the current wording focuses solely on the low values and ignores the steep decrease at the start of the regime. I suggest rewriting along the lines of: "R2, which extends from June to October, follows the onset of the strong decrease in $HNO_3$ total columns that starts around mid-May in most years when the temperatures fall below 195 K. After a steep initial decline in $HNO_3$, R2 is characterized by a plateau of total $HNO_3$ minima. For much of this regime, …".
- L222: over the 10 years of IASI --> over the 10-year study period
- L223-225: In fact, Fig. 3c does not "clearly illustrate" the "strong and rapid $HNO_3$ depletion occurring in June" – it is not possible to differentiate the evolution in June from that in July and August in Fig. 3c. Only in Fig. 3a can we see that the steep drop in $HNO_3$ occurs in June. A similar comment applies to the start of the plateau in July, also not discernible in Fig. 3c alone.
- L226: $HNO_3$in --> $HNO_3$ in
- L227-228: As discussed earlier in this section, temperatures actually dip below 195 K in mid-May in most years, not June. The parenthetical part of this sentence could simply be deleted.

Section 4:

- L256-257: A slight rearrangement would make this easier to read: "potential vorticity smaller than $-10 \times 10^{-5}$ K.m$^2$.kg$^{-1}$.s$^{-1}$ at the potential temperature of 530 K".
- L268: The labels for 2013 on Fig. 4 and Fig. 6 say "11 May", while this sentence and the label on Fig. 5 say "12 May". Please be consistent.
- L270: the strongest $HNO_3$ depletion --> the strongest rate of $HNO_3$ depletion; are detected between --> are between; add a comma after "198.6 K"
- L271: with an exception for the year 2014 which --> with the exception of the year 2014, which
- L292: hence, could only have limited influence on the delimitation --> hence, have only limited influence on the identification
- L293: detection --> determination
- Fig. 5 caption: add "and" after "530 K" (L549); by a --> by the (L553)
- L297-298, 304-305, 308: Again, "over the whole IASI period", "over the IASI period", and "over the ten years of IASI" makes it sound as though IASI only operated for 10 years.

- L302-303: The $-10 \times 10^{-5}$ $K.m^2.kg^{-1}.s^{-1}$ contour of PV demarks the region of strong $HNO_3$ depletion only until October.
- L303-306: It is stated that "the red dashed vertical line indicates the average date for the 50 hPa average drop temperatures calculated in the area of PV $\leq -10 \times 10^{-5}$ $K.m^2.kg^{-1}.s^{-1}$ … It shows that the strongest rate of $HNO_3$ depletion occurs on average [at the] end of May". The date is not specified in the text, but the red dashed line is labeled "24 May" on Fig. 5. This is several days *before* the blue curve representing the $-10 \times 10^{-5}$ $K.m^2.kg^{-1}.s^{-1}$ contour of PV appears on the plot. I assume that the climatological (2008–2017) PV contour is shown, although that is not explicitly stated in the figure caption. Please clarify in the text how the "average date" is being calculated. Please explain how it is that the "average date" for the 50 hPa average drop temperatures, which are calculated in the area of PV $\leq -10 \times 10^{-5}$ $K.m^2.kg^{-1}.s^{-1}$, precedes by a week or so the existence of a significant area within that PV contour in an average sense.
- L305: average end of May --> average at the end of May
- L306-307: It is then stated that "The delay between the maximum in total $HNO_3$ and the start of the depletion (see Fig. 4) is also visible in Fig. 5." Is "maximum in total $HNO_3$" really what is meant here? Total $HNO_3$ is not shown in Fig. 4 (only the second derivative is shown in that figure) – should this reference be to Fig. 3? What is the point being made here – is this saying that the lag of a few days to a few weeks (depending on the year) between the total $HNO_3$ maximum and the strongest rate of $HNO_3$ depletion discussed in L266-268 is also evident in the climatological plot in Fig. 5? If so, then it is not "the start of the depletion" but rather the strongest rate of depletion. In any case, this needs to be written more clearly.
- L307: Delete "For the purpose of the illustrations" (this phrase is confusing and unnecessary).
- L308-312: First, this sentence is long and complicated and could be read as saying that IASI measures PV. Second, the repeatability from year to year of the morphology of the distribution of both $HNO_3$ and temperature in the southern polar region has been known for decades, shown in numerous prior publications. Therefore I suggest deleting the part about the NAT region and rearranging/rewriting the rest as: "… Fig. 6, which shows that IASI measures similar $HNO_3$ total column zonal distributions every year, in particular with respect to the edge of the collar region and of the region of strong depletion (respectively delimited by the PV isocontours of $-5 \times 10^{-5}$ $K.m^2.kg^{-1}.s^{-1}$ and $-10 \times 10^{-5}$ $K.m^2.kg^{-1}.s^{-1}$ at 530 K)."
- L312-313: The statement "Except for the year 2009, the dates for the strongest rate of $HNO_3$ depletion [match] those for the onset of decreasing temperatures below 195 K." seems out of place here – it should have been made in connection with Fig. 4, where it can be seen much more clearly. Moreover, this point seems in conflict with the statement made in L305-306 that "the strongest rate of $HNO_3$ depletion occurs on average … a few days after the temperatures decrease below 195 K". It doesn't seem to me that the difference in the timing in 2009 is really large enough to produce a mismatch of a few days in the 10-year average – please clarify.
- L313: matches --> match
- L315: It might be good to add "Spatial" in front of "Distribution" in the subsection title.
- Fig. 7: Presumably the temperatures in this figure are taken from ERA-Interim fields, thus there should be no missing values, especially averaged over a 2-month period. Therefore, I still do not understand why some of the temperature contours are not closed (PV contours are closed).
- L317-322: This 6-line sentence is hard to get through. Moreover, a crucial piece of information is relegated to a parenthetical. I suggest rewriting as: "To explore the capability of IASI to

monitor the onset of $HNO_3$ depletion at a large scale, Fig. 7 shows for each year of the study period the spatial distribution of the 50 hPa drop temperatures based on the second derivative minima of total $HNO_3$ averaged in 1°×1° grid cells.  The region of interest here is delimited by a PV value of $-8\times10^{-5}$ $K.m^2.kg^{-1}.s^{-1}$ in order to investigate an area a bit larger than the inner vortex core that was the focus of the preceding discussion (delineated in green in Fig. 7 by the PV isocontour of $-10\times10^{-5}$ $K.m^2.kg^{-1}.s^{-1}$ averaged over the interval 10 May to 15 July).”

- L327: It might be good to note that this picture is not much different from that seen in the vortex averages, perhaps something like: “… (not shown).  Although the range of drop temperatures and dates for 1°×1° bins is broader than that found for the inner vortex averages discussed above, the results are qualitatively consistent.  For example, the year 2014 …”.
- L328: highest average --> highest inner vortex average
- L329-336: Is there an explanation for why the retrievals are apparently affected by emissivity issues to a greater degree in 2014 than in other years?
- L350-353: This sentence is awkward and hard to read.  Something like this would be better: “The fact that the weakest minima in the second derivative of total $HNO_3$ are observed in that area (not shown) indicates a weak and slow $HNO_3$ depletion that might be explained by air masses at the inner edge of the vortex experiencing only a short period with temperatures below the NAT threshold.”
- L358: earlier $HNO_3$-depleted --> previously $HNO_3$-depleted
- L360: The words "JGR 2012" should be deleted here, and the paper by Roscoe et al. (2012) needs to be added to the reference list.
- L361: uncertainty into --> uncertainty in
- L364: note --> noted
- L365: delete the comma after "work"
- L366: found --> found only
- L368: The "COSMIC" acronym should probably be spelled out.

Section 5:
- L393: in the poles --> over the pole (plural "poles" implies that the Arctic is also examined)
- L394: R2 runs until October, not September.
- L399: IASI period --> 10-year study period
- L403: 3.8 --> 3.8 K
- L410: Many people may not read the main part of the paper in detail and may only concentrate on the Conclusions, so it would be good to replace "extreme" here with "unrealistic".
- L412: in case of --> arising from
- L414: a series of --> several
- L419: results --> result
- L422: regions --> region

---

## Referee Report (RR4)

**Third review of "Polar stratospheric nitric acid depletion surveyed from a decadal dataset of IASI total columns" by Ronsmans et al.**

The manuscript has again been substantially revised in response to referee comments. Most issues raised previously have been addressed, but a few new ones have been introduced in the revised text. After these minor points are taken care of, the paper will be ready for publication.

Respectfully,
Michelle Santee

- L139-141: This sentence is written in an awkward and confusing manner (plus there is a stray ")" at the end). If I understand it correctly, it would be clearer to say something like: "Together, weaker sensitivity above very cold surfaces with a degrees of freedom for signal (DOFS) of 0.95 and poor knowledge of the seasonally and wavenumber-dependent emissivity above ice surfaces induce larger forward model errors, and consequently the largest measurement errors occur over the Antarctic."
- L157: number of iteration --> number of iterations
- L225: This sentence refers to the "red vertical line" in Fig. 3c, whereas the Fig. 3 caption mentions "The orange horizontal or vertical lines". It would be better to draw all of the lines marking 195 K in Fig. 3 in the same color and refer to them consistently.
- L226: a large interannual variability --> the large interannual variability
- L266-267: "at exactly or a few days after the detection of the 195 K threshold temperature" is awkward and unclear wording. Moreover, I do not think that "detection" is the right word here – it is not that the 195 K threshold is being "detected", but rather that it is being crossed. Finally, the only additional information that this phrase conveys beyond "around the time that temperatures drop below the 195 K threshold", as already stated in L266, is that sometimes the strongest rate of $HNO_3$ depletion is seen a few days after the 195 K threshold is crossed, particularly in 2009. However, emphasizing the delay obscures the fact that occasionally the strongest rate of $HNO_3$ depletion appears to *precede* (not follow) the date on which temperatures drop below 195 K, as in 2013 (and to a lesser extent 2014 as well), according to Fig. 4. Thus I feel that it would be better to simply delete that entire parenthetical comment.
- L304: I find the insertion of the word "annual" in front of "average" in this line confusing. My understanding is that the red vertical dashed lines mark the 10-yr (2008–2017) average of the dates corresponding to the 50-hPa drop temperatures found for each year. As such, this value does not represent an "annual average".
- L311: A closing ")" is missing after "530 K".
- L311-312: My previous comments about L266-267 also apply to the phrase "An exact timing or a delay of a few days between the detection of the averaged 195 K threshold temperature …".
- L313-315: Again, "detection" is not really the right word; also, the sentence is grammatically awkward. I suggest instead "The mismatch between the 10-year averages of the dates on which the 195 K temperature threshold is crossed and the dates for the drop temperatures (see Fig. 5 a and b) is driven by the year 2013, which …" (i.e., add the comma after "2013")
- L325 and 327: 10−5 K.m2.kg−1.s−1 --> $10^{-5}$ K.m$^2$.kg$^{-1}$.s$^{-1}$ (missing superscripts)

---

## Author Response (AR2)

**Response to reviewer #1**

We thank the reviewer for having appreciated the improvements made in the previous version of the manuscript and for his/her new suggestions in order to help improve the manuscript. Kindly find below our responses to the comments (quoted between []). We hope that our responses will address the last concerns and that the changes made will convince the reviewer about the potential of IASI to measure HNO3 in the dark Antarctic polar vortex.

**Major comments**

[In response to both reviews and the detailed comments by Gloria Manney and Michelle Santee, the authors have improved their manuscript considerably. However, I still have concerns about the correctness of their estimation on errors to be applicable within the dark Antarctic polar vortex. In this respect I do appreciate the addition of a comparison with MLS as new Figure 1. However, I don't understand why the authors have stopped half way. It would have only been a small step to show a picture comparing total column amounts integrated from MLS to those of IASI and I would strongly recommend to do this. It would just be easier to put forward and discuss these arguments when the column amounts derived from MLS are presented and quantitative differences shown. E.g. one could easily analyse how much the sensitivity of IASI applied to MLS vertical profiles would change the results. Since it is claimed that the IASI dataset is valuable for polar stratospheric studies as it is, one should do this, even when any tropospheric part would not be considered (in case of MLS). Such a comparison could also help in understanding the problems of the IASI retrievals e.g. over Eastern Antarctica. Further, the argument about "the non-negligible IASI sensitivity to HNO3 in the troposphere" is quite confusing, since all over the paper it is argued that the IASI HNO3 data well represent the stratospheric situation and that, e.g. a renitrification at lower levels cannot be captured well.]

We thank the referee for this suggestion, which was also made by referee #3. We now show in the revised version a picture comparing the total columns integrated from MLS vs IASI in three equivalent latitude bands (see Figure 1 here below). For the comparison, the vertical sensitivity of IASI has been taken into account by applying the FORLI-HNO$_3$ averaging kernels on the co-located MLS profile, which was first interpolated to the FORLI-HNO3 pressure grids and then converted into column profile. The MLS profile was also extended down to the surface by considering the FORLI a priori profile.

The cross-comparison with integrated columns from MLS is very favourable and gives further credit on the IASI total column observations during the polar night. The strong HNO$_3$ depletion is well captured by both IASI and MLS measurements with a perfect match for the onset of the depletion. Note that part of the differences between IASI and MLS are likely due to the different number of co-located data within the 2.5°x2.5° grid cells considered here for the comparison, with a much larger number of observations for IASI (through the quality filtering) than for MLS.

That new comparison figure between integrated column amounts from IASI vs MLS is now included in the revised Section 2 of the manuscript, and the text was adapted to:

"In order to expand on the comparisons against FTIR measurements which is not possible during the polar night, Fig. 2 (top panel) presents the time series of daily IASI total HNO$_3$ columns co-located with MLS measurements within 2.5x2.5 grid boxes, averaged in the 70°S–90°S equivalent latitude band. In order to account for the vertical sensitivity of IASI, the averaging kernels associated with each co-located IASI retrieved profiles were considered for this cross-comparison. The MLS profiles were first interpolated to the FORLI pressure grids, then converted into column profiles. They were also extended

down to the surface by considering the FORLI-HNO3 a priori profile. Similar variations in the $HNO_3$ column are captured by the two instruments, with an excellent agreement in particular for the timing of the strong $HNO_3$ depletion within the inner vortex core. Note that a similar good agreement between the two satellite datasets is obtained in other latitude bands (see Fig. 2 bottom panel for the 50°S–70°S equivalent latitude band; the other bands are not shown)."

[L120-122: "…(Ronsmans et al., 2016). The total columns are associated with a total retrieval error ranging from around 3% at mid- and polar latitudes to 25% above cold Antarctic surface during winter (due to a weaker sensitivity above very cold surface with a DOFS of 0.95 and to an poor knowledge of the seasonally and wavenumber-dependent emissivity above ice surfaces which induces larger forward model errors)."
However, I think the error due to wavenumber-dependent emissivity has not been considered in the error estimation of 'Ronsmans et al., 2016'. So I wonder why only the 25% from that paper is reported here. Further, might there be also an error contribution due to cloud-clearing over very cold Antarctic surfaces?].
The referee is right, as specifically mentioned in Hurtmans et al., 2012 and in Ronsmans et al., 2016, the error due to wavenumber-dependent emissivity (a fixed parameter) is not directly taken into account in the total retrieval error estimation which only includes the smoothing error and the measurement error. The wavenumber-dependent emissivity introduces a bias that is especially used to filter out the contaminated spectra (based on the RMS and on the absolute bias of the residuals).

Nevertheless, if the surface emissivity is not taken into account in the total retrieval error calculation, it indirectly affects this error through compensation effects with the $HNO_3$ profile and the surface temperature that are part of the state vector (adjusted parameters). Hence, the mis-representation of the wavenumber-dependent emissivity is at least to some extent included in the $HNO_3$ total retrieval error calculation.

**Response to reviewer #3**

We thank the referee for his review and for reading the manuscript and our replies to the previous reviews with attention. We acknowledge in particular his request to go further in the comparison with MLS to "cross-validate" the IASI HNO3 product at polar latitudes. As detailed below the comparison is very favorable and this certainly strengthen the paper.

We address all other comments below on a point-by-point basis. We hope that with the changes made in the manuscript and the clarifications given below, the referee will consider that the paper can be published in ACP.

**Comments**

[**L19:** The IASI HNO3 nadir measurements can not be considered as having "good vertical sensitivity". Prospective data users need to know the vertical resolution of the measurements and that is conveyed by the standard practice of quoting the full width at half maximum (FWHM) of the averaging kernel. There is no reasonnot to do this for a nadir sounder e.g. Maddy and Barnett, Vertical Resolution Estimates in Version 5 ofAIRS Operational Retrievals, IEEE Trans. Geosci. Remote Sens., 46, 8, 2375-2384, 2008.]
We think that there may be a semantic misunderstanding related to this point. As explained in our previous responses to the referee and as stated in the manuscript, we here refer to "a good vertical sensitivity in the low and middle stratosphere", not to a good vertical resolution of the measurement.

The averaging kernels give the sensitivity of the retrieved value to the true profile with:
1/ the position of its peak indicating the altitude of maximum sensitivity
2/the FWHM giving an estimation of the vertical resolution (~30 km for $HNO_3$ from IASI; it is specifically why we consider a total column as mentioned at several places in the submitted version and in previous FORLI-$HNO_3$ studies). The FWHM is now indicated in Section 2 of the revised manuscript.

Looking at typical examples of averaging kernels (e.g. in Ronsmans et al., 2016 or provided in Figure 2 in our previous responses), it is clear that the maximum vertical sensitivity of IASI to the $HNO_3$ profile ranges in the mid-low stratosphere with values of 1 along the total columns averaging kernel in that region, indicating a good sensitivity of IASI to $HNO_3$ in that altitude range, even if the information coming from a specific level cannot be attributed by IASI/FORLI at that exact specific level due to the coarse resolution (FWHM of the averaging kernels of ~30 km) which forces us to consider a total column.

[**L29:** For reasons explained elsewhere the "formation temperature" can be better expressed as an "existence thresold"]
This is a good suggestion. It has been corrected, as suggested, in the introduction.

[**L26-29**: The averaged "drop temperature" disregards the considerable interannual variability in the early stage formation of different types of Antarctic PSCs and the role played by the exposure of liquid PSCs to low temperatures in the formation of NAT i.e. many studies have shown that NAT is not uniquely constrained to nucleate at TNAT and some supersaturation is generally needed leading to a lower temperature for NAT formation (as in fact you discuss in the text L55-L75). Therefore, stating that the drop temperature is "consistent with TNAT", which implies that PSCs are mainly NAT forming at TNAT, is invalid.]
Here, we just mention that, interestingly, the HNO3 drop temperature matches TNAT (for typical 50 hPa atmospheric conditions) for the purpose of the description of our results. We don't think that the

discussion on the different mechanisms of NAT formation, which are described later in the introduction, would be useful in the abstract.

However, in order to not give to the reader the feeling that "PSCs are mainly NAT forming at TNAT", "consistent with" has been replaced by "close to".

[**L28**: the" " sign should be deleted since a specific value and its uncertainty is quoted.]
Done

[L30: Some corresponding indication of the equivalent latitude range would be useful here]
'…in the region of potential vorticity lower than $-10 \times 10^{-5}$ K.m$^2$.kg$^{-1}$.s$^{-1}$ **(similar to the 70° – 90° S Eqlat region during winter)**" has been added here.

[L30-31: The spatial distribution and inter-annual variability of the drop temperature are investigated and discussed in the context of previous PSCs studies. However, the study presented here does not include any observed data on PSCs and is therefore not a "PSC study"]
In fact it is not our intention to provide with this paper a PSC study and in the manuscript we acknowledge that: "the measured HNO$_3$ total column does not allow differentiating the uptake of HNO$_3$ by different types of PSC particles along the vertical profile". "In the context of previous PSCs studies" has been deleted to avoid misunderstanding early on the manuscript.

[L92-94: Please give the fullwidth half-max (FWHM) of the vertical response in km and not just the height of maxi-mum sensitivity.]
See our response to the first comment above. The FWHM is now mentioned in Section 2 of the revised manuscript.

[L251-255: The software bug that was fixed in the revised version has changed the drop temperatures such that the year with 202.8K (previously 190.6K) is a significant outlier since it lies 8K higher than the 10-year mean drop temperature and is almost as much above the assumed 50hPa TNAT (195K). Therefore it does not support the statement on L365-366 that the 10-year range "demonstrated the good consistency between the 50 hPa drop temperature and the PSCs formation temperatures in that altitude region".]
The year 2014, which is an outlier with the drop temperature of 202.8K, is now excluded from the analysis. The sentence has been rewritten as follows:
"Except for the year 2014, the 50 hPa drop temperatures are detected between 189.2 K and 198.6 K (194.1 K ± 2.8 K - 1σ standard deviation - on average over the 10 years, excluding 2014 that stands out with a drop temperature of 202.8 K)."

[L295-300: It is not clear why this data quality problem has not been addressed in the revised submission. Measurements that are known to be bad must be screened out.]
As specifically mentioned in Sections 2 and 4.2 of the manuscript, a series of quality flags were applied to filter out the poor retrievals. Nevertheless, there remains indeed a few poor quality retrievals above icy surface due to a misrepresentation of the seasonally and wavenumber-dependent emissivity above such surface. This parameter still remains critical and causes poorer retrievals that, in some instances, pass through the series of quality filters and could affect the drop temperature calculation. Developing a perfect filtering for these areas has not been possible at the moment. Such contaminated spectra should for now be treated with caution while a more appropriate flag could be developed.

[L332-337: Why is the discussion in L302-338 and L367-376 limited to nucleation of NAT and ice PSCs

with no mention of STS? There is no nucleation barrier to STS formation and it generally forms in advance of ice nucleation except possibly under very fast cooling e.g. in mountain waves. STS is not even mentioned in the paper after the introduction in L55-72.]

In Section 4.2, we have simply considered the formation temperature of PSCs that first nucleate (typically NAT). This is why the averaged isocontour of 195 K is represented on figure 6. No distinction between the $HNO_3$ uptake by the different PSCs, nor specific mention of ice PSCs are made in this section.

In the paragraph L333-339, we concluded that: "… the overall **range in the drop 50 hPa temperature** for total $HNO_3$ inside the isocontour for the averaged temperature of 195 K, typically extends from ~187 K to ~195 K, which **falls within the range of PSCs nucleation temperature at 50 hPa**: **from** slightly below $T_{NAT}$ **to** around 3-4 K below the ice frost point - $T_{ice}$…", knowing that the formation temperature of STS is in between (~192 K) for typical 50 hPa atmospheric conditions.

The $HNO_3$ total column measured from IASI does not allow differentiating the uptake of $HNO_3$ by different types of PSCs along the vertical profile, as the referee points out and this is why such a discussion cannot be performed.

[The response does not address the specific case example of where IASI views HNO3 depleted higher layers that overlay lower enhanced layers. How does the IASI column HNO3 measurement change if the HNO3 is redistributed in the vertical coordinate by denitrification and renitrification? A further question would be how does downwelling of higher values of HNO3 affect the HNO3 column?]

See also above: we hope that it is clear from the replies that the coarse vertical resolution does not allow to capture such altitude-dependent processes. In the manuscript, we explicitly refer to this, e.g.
"… despite the lack of vertical resolution, which is recognized in the paper and which forces us to consider total $HNO_3$ columns, IASI is characterized by a good sensitivity to $HNO_3$ at specific levels, in particular, in the range between ~70 hPa to ~30 hPa in the southernmost latitude in winter" … "where the strong $HNO_3$ depletion occurs…". "Above the Antarctic, the altitude of maximum sensitivity … ~22 km in winter".
"The likely renitrification of the lowermost stratosphere (Braun et al., 2019; Lambert et al., 2012) where the $HNO_3$ concentrations and the IASI sensitivity to $HNO_3$ are lower (Ronsmans et al., 2016) cannot be inferred from the IASI measurements."

In an effort to specifically address this question and to quantify the potential impact of the likely renitrification of the lower stratosphere on the IASI total column, we should compare IASI measurements collocated with cases of renitrification at lower levels, identified from independent measurements of $HNO_3$ vertical profiles, to IASI observations that do not experience renitrification. It has not been investigated at this stage. Note that Braun et al. (2019) identified renitrification at the lowermost stratosphere, below ~12 km where the IASI sensitivity is the lowest.

[As a further example of the 2D potential, could IASI be used to image the HNO3 field to show depletion in the cold phases of mountains waves e.g. near the Palmer peninsula (similar to the wave structures seen in AIRS brightness temperatures) or is that defeated by the vertical integration caused by the poor vertical resolution?]

As explained in our previous responses to referee #3, Figure 7 of the revised manuscript shows the spatial distribution of the drop temperature inside a region enclosed by an isocontour PV of $-8 \times 10^{-5}$ K.m$^2$.kg$^{-1}$.s$^{-1}$, which, hence, encircles a region larger than the inner vortex core (see Figures 5 and 6 of the revised manuscript). The drop temperatures much above the NAT formation temperature, which are mostly found outside the averaged isocontour PV of $-10 \times 10^{-5}$ K.m$^2$.kg$^{-1}$.s$^{-1}$, do not correspond to high minima

($<-0.5 \times 10^{14}$ molec.cm$^{-2}$.d$^{-2}$) in the second derivative of HNO$_3$ total column with respect to time.

We cannot argue that it corresponds to the NAT belt of Höpfner et al. (2006) downstream of the Antarctic Peninsula, which was enclosed inside the region of the NAT threshold temperature; the highest drop temperatures from IASI are found on the contrary outside the isocontour of the NAT threshold temperature (see figure 7 of the revised manuscript). Comparing the distributions of drop temperatures from IASI with PSC information from CALIPSO/MIPAS remains difficult given the difference in spatial coverage and, most importantly, the highly variable distribution of PSC types and of the NAT belt, temporally (daily) and spatially (Höpfner et al., 2006; Lambert et al., 2012).

Note that Hoffmann et al. (2014; doi:10.5194/amt-7-4517-2014) has reported an intercomparison of stratospheric gravity wave observations of both AIRS and IASI instruments and "showed that AIRS and IASI provide a clear and consistent picture of the temporal development of individual gravity wave events" … "While AIRS has been used successfully in many previous gravity wave studies, IASI data are applied here for the first time for that purpose. Our study shows that gravity wave observations from different hyperspectral infrared sounders such as AIRS and IASI can be directly related to each other, if instrument-specific characteristics such as different noise levels and spatial resolution and sampling are carefully considered ».

["CALIOP measurements ... this goes beyond the goal of this paper, which is to demonstrate the capability of IASI to measure HNO3 columns that are relevant for stratospheric studies". That goal was largely achieved already by Ronsmans et al (2016) and published in Atmos. Meas. Tech. This paper is under review for Atmos. Chem. Phys. and should relate more to a science investigation rather than a technical description. The comparisons with MLS are a welcome improvement, but unfortunately fall short of the analysis I was expecting. Surely the tropospheric contribution of HNO3 to the IASI column is not all that much (you could estimate the effect to confirm). I expected the MLS profile to be integrated with the IASI response function for a more direct comparison. That would facilitate a quantitative interpretation of the differences in the variation of the column data from the two instruments.]

We thank the referee for this suggestion, which was also made by referee #1. We now show in the revised version a picture comparing the total columns integrated from MLS vs IASI in three equivalent latitude bands (see Figure 1 here below). For the comparison, the vertical sensitivity of IASI has been taken into account by applying the FORLI-HNO$_3$ averaging kernels on the co-located MLS profile, which was first interpolated to the FORLI-HNO3 pressure grids and then converted into column profile. The MLS profile was also extended down to the surface by considering the FORLI a priori profile.

The cross-comparison with integrated columns from MLS is very favourable and gives further credit on the IASI total column observations during the polar night. The strong HNO$_3$ depletion is well captured by both IASI and MLS measurements with a perfect match for the onset of the depletion. Note that part of the differences between IASI and MLS are likely due to the different number of co-located data within the 2.5°x2.5° grid cells considered here for the comparison, with a much larger number of observations for IASI (through the quality filtering) than for MLS.

That new comparison figure between integrated column amounts from IASI vs MLS is now included in the revised Section 2 of the manuscript, and the text was adapted to:

"In order to expand on the comparisons against FTIR measurements which is not possible during the polar night, Fig. 2 (top panel) presents the time series of daily IASI total HNO3 columns co-located with MLS measurements within 2.5x2.5 grid boxes, averaged in the 70°S–90°S equivalent latitude band. In order to account for the vertical sensitivity of IASI, the averaging kernels associated with each co-located

IASI retrieved profiles were considered for this cross-comparison. The MLS profiles were first interpolated to the FORLI pressure grids, then converted into column profiles. They were also extended down to the surface by considering the FORLI-HNO3 a priori profile. Similar variations in the $HNO_3$ column are captured by the two instruments, with an excellent agreement in particular for the timing of the strong $HNO_3$ depletion within the inner vortex core. Note that a similar good agreement between the two satellite datasets is obtained in other latitude bands (see Fig. 2 bottom panel for the 50°S–70°S equivalent latitude band; the other bands are not shown)."

[I also wanted to see specific depleted vs non-depleted cases (one with a re-nitrification layer would be good also) generated along with the simulated IASI columns and the calculated columns. I suggest that the figure provided on the averaging kernels etc could be added to a supplemental material section with a description tailored to the cases studied here in addition to just referring readers to a prior publication.] This is specifically what we have shown in the Figure 1 that we provided in our responses to the previous comments and that showed examples of vertical $HNO_3$ profiles retrieved within the dark Antarctic vortex (depleted case above Arrival Height) and outside the vortex (non-depleted case above Lauder). The retrieved profiles were provided along with their associated total retrieval error and averaging kernels.

As suggested, we now provide that figure in the manuscript. Note that we have found it better to include it in Section 2 than in a supplementary material.

A re-nitrification case can hardly been identified from IASI (see comment above).

[CALIOP PSC data (Pitts et al 2013, doi:10.5194/acp-13-2975-2013) have been used to show that different PSC types exist in different temperature regimes, with ice PSCs detected close to the frost point, STS follows the expected equilibrium curve and NAT exhibits two preferred mode below the NAT existence temperature. The analysis presented here is not constrained by the simultaneous presence of known PSC types and in fact there may not even be any PSCs in the atmospheric path sampled. Therefore, it is too simplistic to compare the drop temperatures to TNAT. The proximity of the 10-year mean drop temperatures to TNAT does not constitute a validation as is claimed here. Individual years could be expected to show a variation in drop temperature because of interannual atmospheric differences. For instance, the years domimated by STS should necessarily show lower drop temperature than years dominated by NAT. The highest drop temperatures are far above PSC temperatures (e.g. 202.8K at 50hPa in one particular year) and deserve more scrutiny and should be investigated thoroughly. Interannual comparisons of the drop temperature may benefit from using (T-Tice) as the temperature coordinate (rather than absolute temperature) as this removes variations due to changes in H2O partial pressure (see Fig 2 of Pitts et (2013)). There is a fundamental problem with making an assessment of the potential future scientific utility of the drop temperatures when they have only been evaluated in the absence of knowledge of the different types of PSCs present.]
We are not sure to follow the referee well:

1/ As mentioned above, we do not aim at a PSC study, given that: "the measured $HNO_3$ total column does not allow differentiating the uptake of $HNO_3$ by different types of PSC particles along the vertical profile", as mentioned in the manuscript. So indeed, we have simply considered the formation temperature of PSCs that first nucleate, i.e. NAT. This is why the threshold temperature of 195 K is considered in the study to identify likely PSCs-containing regions. We don't have the possibility to perform a more thorough investigation of the NAT existence. However it is true that years dominated by STS should induce lower drop temperatures than years dominated by NAT and this has been now specifically mentioned in the revised manuscript when discussing the variability observed in the 50 hPa drop temperatures: "The range observed in the 50 hPa drop temperature could reflect the preponderance

by one type of PSCs over another." But here again, the influence of the different types of PSCs on the drop temperature that is calculated from the HNO₃ total column can hardly be investigated.

2/ As discussed in a comment above, the year 2014 with the drop temperature of 202.8K is standing out and is now excluded from the analysis.

3/ We agree that working in the T – Tice coordinate system, as in Fig.2 of Pitts et al. (2013), may be interesting for illustrating theoretical equilibrium uptake of HNO3 by STS and NAT as it removes variations due to differences in atmospheric pressure level (illustrated at 30 hPa et 50hPa in Pitts et al. (2013)). In our case, it has no influence at all on the temperature timeseries (see Figure 2 below). Tice considered here is determined as 188 K by Murphy and Koop (2005) for typical 50 hPa atmospheric conditions only. Hence we suggest to keep the absolute temperature in Fig.4 of the revised manuscript.

4/ The concept of "drop temperature" is exploited for the first time in this study; it allows relating the strong decrease in HNO₃ to the likely existence of PSCs. We believe that the concept could be used in future studies by nadir or limb sounders.

[The response does not address the specific case of whether there are differences in bias and uncertainty for depleted and non-depleted conditions.]
We think that the referee may have missed that in our responses and in the submitted manuscript. The total retrieval error values outside vs inside the polar regions during winter, i.e. during depleted-conditions, were in fact provided in Section 2 (line 129-135). We report a larger error (25%) due to a weaker sensitivity (lower DOFS) above very cold surface and to an poor knowledge of the seasonally and wavenumber-dependent emissivity above ice surfaces. When compared to ground-based FTIR measurements (whatever the latitudes and seasons; see Ronsmans et al., 2016), IASI is always positively biased. At Arrival Heights, the bias is not larger than at the other stations and it is not larger for depleted than for non-depleted conditions. For the polar stations, a bias lower than 12% is calculated over the altitude range where the IASI sensitivity is the largest.

[Thick ice PSCs have been detected by AIRS, TOVS HIRS2 and AVHRR (see Stajner et al. and refs therein,https://doi.org/10.1029/2007GL029415). Do these have an effect on the HNO3 retrieved by IASI?]
No. From that paper, thick ice PSCs are detected on the AIRS moisture channel at 6.79-$\mu$m (i.e. 1473 cm-1), while in the atmospheric window region used for the HNO₃ retrieval from IASI: "Comparisons of AIRS spectra with a radiative transfer model in the window region 10-12.5 μm show signatures of near-micron sized cirrus ice particles [Kahn et al., 2003]."

[Even after all the quality controls are applied there are apparently still cases with poor retrievals that could be removed.]
Please see our response to the same comment related to L295-300 above.

[What is the fraction of data that is affected by surface emissivity?]
This is difficult to estimate since such data pass through the filtering criteria that include bias and RMS of the residuals specifically used to flag the remaining cloudy scenes or the surfaces with sharp emissivity variations at 11.5 µm. These emissivity features are suspected to explain the few hotspots in HNO₃ observed above Antarctica (likely due to compensation effect in order to reduce as much as possible the residual during the iterative phases of the retrieval), and similarly, the high extremes in the drop temperatures found above eastern Antarctica over some years.

[The conclusion of the paper is that ability to monitor the polar atmosphere over several decades with current and planed IASI instruments "will provide an unprecedented long-term dataset of HNO3 total columns". The drop temperature is defined as the 50hPa temperature corresponding to the greatest rate of decline of the column HNO3 with respect to time. However, even with a record now extending over a decade the scientific utility of this dataset has not been demonstrated].

See our previous responses. The concept of "drop temperature" is exploited for the first time in this study and while it does obviously not inform on the detailed formation mechanisms of PSCs from HNO3, it provides a robust indication on the occurrence of some polar processes at play in the stratosphere. There is an increasing interest and use of the IASI data products for "climate" studies (i.e. through so-called "thematic climatic data records"). We are confident that the HNO3 dataset will contribute to these data records in the near-future and benefit several modelling studies.

[Figure]

Figure 1. Time series of daily IASI total HNO$_3$ column (blue) co-located with MLS and of MLS total HNO$_3$ columns (orange) within 2.5x2.5 grid boxes, averaged in the 70°S–90°S (top panel), the 50°S–70°S (middle panel) and the 30°S–50°S (bottom panel) equivalent latitude bands. The error bars (blue) represents 3σ, where σ is the standard deviation around the IASI total HNO$_3$ daily average.

[Figure]

Figure 2. Time series of total $HNO_3$ second derivative (blue, left y-axis) and of the temperature at 50 hPa (red, right y-axis) in the $T - T_{ice}$ coordinate system (where $T_{ice}$ is the frost point temperature determined by Murphy and Koop (2005)), in the region of potential vorticity lower than $-10 \times 10^{-5}$ $K.m^2.kg^{-1}.s^{-1}$. The red horizontal line corresponds to the 195 K temperature. The vertical dashed lines indicate the second derivative minimum in $HNO_3$ for each year. The corresponding dates (in bold, on the x-axis) and temperatures are also indicated. The time series of total $HNO_3$ second derivative (dashed blue) and of temperature at 50 hPa (grey) in the 70–90°S Eqlat band are also represented.

---

## Author Response (AR3)

**Response to reviewer #1**

We thank the reviewer for reading the last version of the manuscript and for his/her new technical corrections. We hope that the corrections suggested by the two referees, which are implemented in that new version, will satisfy the reviewers and convince them about the potential of the IASI HNO3 dataset.

**Technical corrections**

[l289: 'indicates' -> 'indicate']
[l381: 'demonstrated' -> 'demonstrate']
[l391: 'a factors' -> 'of factors']
[l455: 'represents' -> 'represent']
All the technical corrections are implemented in the new version of the manuscript.

**Response to reviewer #4 – M. Santee**

We deeply thank Michel Santee for her review, for reading the manuscript with attention and for her help in rewriting some unclear sentences; her corrections are clearly improving the paper. We address all her comments below on a point-by-point basis. We hope that with the corrections made in the manuscript and the clarifications given below, Michel Santee will consider that the paper can be published in ACP.

**Comments**

[L19: in the mid-stratosphere --> in the lower-to-mid stratosphere; also, delete "causal"]
Done

[L23: This wording is very awkward; I suggest: evolution of the pair HNO3-temperature --> evolution of HNO3 together with temperature]
Done

[L24: The meaning of "in the cycle of IASI" is not clear – do the authors mean "annual cycle"?]
Indeed, "annual" has been added.

[L27-28: delete "differentiating"; add "to be differentiated" after "profile"]
Done

[L28: Two different values for the average drop temperature are given in the text on p. 6 (L265 and L290). These values need to be reconciled and the correct one quoted here.]
$194.2 \pm 3.8$ K is the drop temperature including the year 2014, while $194.1 \pm 2.8$ K is the drop temperature when 2014 is excluded. The year 2014 is now taken into account through the revised manuscript when mentioning the average drop temperature.

[L32-33: It is a bit of an exaggeration to say that "this paper highlights the capability of IASI to monitor the long-term evolution of polar stratospheric composition and processes involved in the depletion of stratospheric O3". It would be more accurate to state that "this paper highlights … evolution of polar stratospheric HNO3, a key player in the processes involved …"]

Done as suggested.

[In a number of places in the presentation of background material (e.g., L39, L42, L45, L51), a few citations are given for very well-established concepts, but many other equally suitable papers could have been cited instead of or in addition to the ones listed. Obviously not all relevant papers can be referenced for these points, but "e.g." should be added in these lines to avoid giving readers the impression that the selected references are the only appropriate ones.]
Done

[In addition, the recently published PSC review paper by Tritscher et al. (Rev. Geophys., 2021) covers all of this background material and should also be cited in several places in this section.]
We thank Michel Santee for suggesting that impressive review paper. As required, it is now cited in several places in this section.

[L44-45: Technically, denitrification delays only the reformation of $ClONO_2$ – $HNO_3$ is not required for HCl production, thus it is not quite correct to say "chlorine reservoirs" here.]
Changed to: "The denitrification of the polar stratosphere during winter delays the reformation of $ClONO_2$, a chlorine reservoir, and…".

[L46-47: PSCs surface --> PSC surfaces; PSCs particle --> PSC particles]
Done

[L47-48: three 3different --> three different]
Done

[L56: delete comma after "($T_{ice}$)"]
Done

[L60: A reference should be provided for the point that NAT nucleation was thought to require temperatures below $T_{ice}$.]
Zondlo et al. (2000) and Voigt et al. (2003) have been added here.

[L73: the PSCs formation --> PSC formation]
Done

[L74: add a comma after "2019)"]
Done

[L74-75: This sentence ("The influence of $HNO_3$ in modulating $O_3$ abundances in the stratosphere is furthermore underrepresented in CCMs") is out of place and potentially confusing for many readers, who will naturally assume that it has something to do with PSC processes in the lower stratosphere as it comes at the end of a long paragraph discussing nothing but PSCs. Kvissel et al. (2012), however, describe $HNO_3$ enhancements between 10 and 1 hPa induced by energetic particle precipitation. It's not clear why this sentence has been included here; perhaps it is meant to provide motivation for this study, but I doubt that IASI total column measurements with maximum sensitivity around 50 hPa could shed much light on modest (< 6 ppbv) $HNO_3$ enhancements in the middle and upper stratosphere. In fact, this issue is not explored in the manuscript. Thus this sentence should be deleted.]
The sentence has been deleted as suggested.

[L77: measure --> have measured]
Done

[L79-82: This poorly composed sentence is grammatically incorrect and hard to interpret. I'm not sure what is meant by "follow their formation mechanisms". I suggest alternative wording, but I may have misunderstood the intent: Spaceborne instruments such as the CALIOP/CALIPSO lidar and MIPAS/Envisat measuring in the infrared are capable of detecting and classifying PSC types, allowing their formation mechanisms to be investigated (Lambert et al., 2016; Pitts et al., 2018; Spang et al., 2018, and references therein); these satellite data complement in situ measurements (Voigt et al., 2005) and ground-based lidar (Snels et al., 2019).]
Corrected as suggested.

[L84: the PSCs formation --> PSC formation; Urban 55 et al. --> Urban et al.]
Done

[L86: depends --> depend]
Done

[L92: similar to the limb --> similar to that from the limb]
Done

[L94: where the PSCs cloud form --> where PSCs form]
Done

[L95: 10-years --> 10-year]
Done

[L96: for providing --> to provide]
Done

[L104: embarked on --> onboard ("embarked" is not the right word)]
Done

[L117: FWHM --> full width at half maximum (FWHM)]
Done

[Figure 1 caption:
o L446: molec.cm-2/molec.cm-2 --> molec.cm-2/molec.cm-2
o L446: after "molec.cm-2/molec.cm-2" add "; colored lines, with the altitude of each kernel represented by the colored dots)"
o L447: Height --> Heights; 169,40 --> 169.40]
Done

[L122 & 125: Height --> Heights]
Done

[L126-127: Above (in L119) it was stated that the largest sensitivity of IASI is from ~70 to ~30 hPa, and here it says "the altitude of maximum sensitivity (at around 30 hPa for this case)". This is confusing because the grey line in the Arrival Heights panel of Fig. 1 depicting the "sensitivity profile" peaks at

~75 hPa, not 30 hPa. I believe that it is the total column averaging kernel, not the "sensitivity profile", that determines the altitude of maximum sensitivity? The authors should take into consideration that not all readers of this paper will be experts on IASI data or will have read Ronsmans et al. (2016). Please clarify the meaning of the "sensitivity profile" (grey curve) and the total column averaging kernel (black curve) and how they relate to the altitude where IASI provides the most information on $HNO_3$.]

- The averaging kernel profile describes how the true state changes the estimate at a specific altitude, i.e. how the retrieval smooths the true profile ($d(HNO3_{ret})i/d(HNO3_{true})j$).
- The total column averaging kernel ($d(column_{ret})/d(HNO3_{true})j$), i.e. the sum of the averaging kernels, indicates how the true state at a specific altitude changes the retrieved total column, i.e. the sensitivity of the total column measurement to changes in the vertical distribution of HNO3, hence, the altitude to which the retrieved total column is mainly sensitive/representative.
- The sensitivity profile ($d(HNO3_{ret})i/d(column_{true})$), i.e. the sum of the elements of an averaging kernel characterizing the retrieval at a specific altitude, returns the sensitivity of the retrieval at that altitude, i.e. to which extent the retrieval at that specific altitude comes from the spectral measurement rather than the apriori.

The following sentence has been added in the revised version: "The total column averaging kernel (in black) indicates the sensitivity of the total column measurement to changes in the vertical distribution of $HNO_3$, hence, the altitude to which the retrieved total column is mainly sensitive/representative, while the sensitivity profile indicates to which extent the retrieval at one specific altitude comes from the spectral measurement rather than the a priori."

Indeed, the altitude of maximum sensitivity of the total column measurement refers to the total column averaging kernel (black curve). This is now clarified in the revised text in order to avoid confusion.

[L127: At Lauder on the contrary, --> In contrast, at Lauder]
Done

[L128: "larger range of maximum sensitivity" – again, is this conveyed by the "sensitivity profile" (grey curve) or the total column averaging kernel (black curve)?]
See our response to the comment [L126-127] here above.

[L129-130: The statement "from around 3% at … polar latitudes to 25% above the cold Antarctic surface during winter" is confusing, because of course the Antarctic is at polar latitudes. Please be more precise in the wording here.]
Corrected: "… from around 3% at mid- and polar latitudes (except above Antarctica) to 25% above cold Antarctic surface …".

[L130: DOFS --> degrees of freedom for signal (DOFS)]
Done

[L132: "lower than 12%" is ambiguous here; assuming that the values are biased low by more (not less) than 12%, this should be rewritten as "low bias (exceeding 12%)"]
Corrected to: "… and a low absolute bias (smaller than 12%)".

[L135: FTIR measurements which is not possible during --> FTIR measurements, which cannot be made during]
Done

[Figure 2 caption:
o L454: It should be explicitly stated here (not only in the text) that the MLS total column estimates were obtained by extending the MLS partial stratospheric column values using the FORLI-HNO3 a priori information.
o L454: 2.5 x 2.5 --> 2.5° × 2.5°
o L454: middle --> bottom
o L455: Figure 2 does not include a panel showing 30°–50°S EqL]
Corrected

[L137: 2.5 x 2.5 --> 2.5° × 2.5°]
Done

[L139: The meaning of "the averaging kernels … were considered" will not necessarily be clear to all readers. It would be better to state explicitly that the IASI averaging kernels were applied to the co-located MLS profiles.]
Corrected as suggested.

[L140-141: "column profiles" is an oxymoron. Please specify the MLS retrieval pressures over which the partial stratospheric column was calculated. Thus, rather than "… grids, then converted into column profiles. They were also extended down to the surface by considering the …", it would be better to rephrase along the lines of "… grids, and partial stratospheric columns above xxx hPa were calculated. MLS total columns were then estimated using the …".]
Rephrased to: "The MLS VMR profiles over the 215-1.5 hPa pressure range were first interpolated to the FORLI pressure grids and extended down to the surface by using the FORLI-HNO3 a priori profile, and then converted into partial columns."

[L155-165: In their Short Comment, Manney & Santee pointed out: "It is highly problematic to use a single theta level to distinguish inside from outside vortex regions for column measurements. This approach implicitly (and erroneously) assumes that the vortex does not tilt, shrink, or expand with height over the altitude range considered. A better approach would have been to check PV over a range of levels and discard measurements classified as outside the vortex at any one of those levels." In my opinion, the authors have not adequately addressed this point. I understand that 530 K falls in the region of the atmosphere where IASI has maximum sensitivity to HNO3. Nevertheless, it is simply not credible that changes in the size, shape, or location of the vortex over the 30–70 hPa domain primarily covered by the measurements had no effect on the results on any day in the 10-year IASI record. At this point, I am not suggesting that the authors redo their analyses, but I would like to see added to the manuscript an explicit acknowledgment of the fact that relying on the determination of the vortex boundary on a single potential temperature surface for interpretation of total column measurements inevitably introduces some uncertainty into the results because it fails to account for the possibility of changes in the size, shape, or location of the vortex with altitude.]
In order to address this comment, we have added this paragraph in the revised section 4.1:

"Finally, it should be noted that, because the size, shape or location of the vortex vary slightly over the altitude range to which IASI is sensitive (from ~30 to ~70 hPa during the polar night), the use of a single potential temperature surface for the calculation of drop temperatures could introduce some uncertainties into the results. However, several tests suggest that these variations of the vortex are overall minor and, hence, could only have limited influence on the delimitation of the inner polar vortex (delimited by a PV value of $-10 \times 10^{-5} K.m^2.kg^{-1}.s^{-1}$ at 530 K) and on the detection of the average drop temperature inside that region."

[L158: closing ")" missing after "2016)" – it should be "2016))"]
Done

[L158: "starts a few degrees or slightly below" is awkward and confusing – "slightly" could mean "a few degrees". It would be clearer to say: "starts within a few degrees below"]
Done

[L161: It is not quite correct to say "to identify the PSCs-containing regions" – regions with temperatures below the threshold do not necessarily contain PSCs. It would be better to say "to identify regions of potential PSC existence".]
Done

[L170: delete the comma after "S)"]
Done

[L170: This wording – "over the whole period of measurements (2008–2017)" – seems to imply that IASI has taken no data since 2017, which I do not believe is the authors' intention.]
Changed to: over the whole study period

[L172: 2018) where the contribution of the PSCs into --> 2018), where the contribution of PSCs To]
Done

[L173: delete "here"]
Done

[L175-176: The wording "along the $HNO_3$/temperature cycle" is not clear. I think the authors mean "within the $HNO_3$/temperature annual cycle". Or maybe "during" rather than "within"]
Changed to: during the $HNO_3$/temperature annual cycle.

[L184: "R1 in Figures 3a and b" – since the regimes are clearly labeled in the panels of Figure 3, I'm not sure that this statement is needed. Moreover, the label "R1" is also used in Figure 3c.]
"R1 in Figures 3a and b" has been deleted.

[L189: deployment --> development ("deployment" is not the right word); also, add a comma after "vortex"]
Done

[L190-191: lower latitudinal airmasses --> lower-latitude airmasses]
Done

[L193: A problem with the definition of R2 is that, as shown later in the paper, the onset of strong $HNO_3$ depletion actually begins in mid-May in most years.]
Changed to: "R2, which extends from June to October, follows the onset of the strong decrease in $HNO_3$ total columns, which starts around mid-May in most years when the temperatures fall below 195 K, and is characterized by a plateau of total $HNO_3$ minima."

[L193: add commas after "R2" and "October"]
Done

[L195: the HNO₃ total columns average below --> average HNO₃ total columns are below]
Done

[L198-199: To avoid the potential for confusion with SSWs: Despite the stratosphere warming with 50 hPa temperatures up to 240 K --> Despite 50 hPa temperatures increasing up to 240 K]
Done

[L202: by the PSCs sedimentation --> by sedimentation of PSCs]
Done

[L203: add "e.g." to the list of references]
Done

[L203-204: add commas after "2012)" and "2016)"]
Done

[L204-205: The meaning of "can hardly be inferred" is ambiguous. I expect that the lack of sensitivity of IASI total column HNO₃ to the LMS precludes detection of renitrification from those measurements. Thus: can hardly --> cannot]
Done

[L206: where --> when]
Done

[L210: occurs --> occurred]
Done

[L212: HNO₃ total columns in 2010 were higher in September as well as in July and August.]
Changed to: "in July - September"

[L219: add "and" after "(R2),"]
Done

[L220: Based on Figure 3a, the plateau of low HNO₃ abundances begins in July, not August.]
Corrected

[Figure 3c:
o Figure 3a clearly shows that the "strong and rapid HNO₃ depletion" (as stated also in L219) occurs mainly during June, so why is the steep drop in HNO₃ in Figure 3c labeled "Jun-Aug"?
The strong HNO3 depletion starts in June and the minima in HNO3 levels are reached in July or August. Based on Fig. 3c, it is quite hard to dissociate these months.
o I do not find the regime markers (R1, R2, R3) on this panel particularly helpful – these labels are essentially floating in semi-arbitrary positions on the plot and convey no real meaning.
We agree that the regime markers are already clearly mentioned in Figures 3a and 3b, but, for more consistency, we believe it is useful that they are indicated in Figure 3c as well.
o It is stated that this panel "highlights the interannual variability in total HNO₃" (L221). But interannual variability is much easier to interpret in Figure 3a. For example, the anomalous behavior in July–September 2010 so evident in panel (a) does not stand out in panel (c). In fact, I would argue that it is

not necessary to color-code the lines by year in panel (c), as the details of individual years are better seen in panel (a) anyway. Just having 10 separate lines would still illustrate the interannual spread even without distinguishing the specific years. If the lines are not color-coded by year, that would allow them to be color-coded in adifferent manner. For example, 12 different colors could be used to indicate the portions of the curves corresponding to each month. This would allow the interannual variability in different months to be compared at a glance. In addition, different line styles (e.g., solid, dashed, dotted) could be used to differentiate the three regimes. Reformulating the plot along these lines would make this panel much more useful than it currently is.

As suggested, we have tried to color-code the lines in order to represent the portions of the curves corresponding to each month, but the dissociation of the months is not that evident with superposition of the HNO3-temperature values over the previous month and the next month, which makes the plot quite hard to visualize. We concluded for this reason that it is better to keep Figure 3c as it is, with one color per year.

o Please add minor tick marks on the axes (x and y) of all of the panels in Figure 3.]
Done

[L221-222: The two parts of this sentence ("highlights the interannual variability in total HNO₃" and "shows a strong consistency in the onset of the depletion between each year") seem contradictory. If the behavior is consistent from year to year, then interannual variability is small. The sentence should be re-written using more careful language. The wording is also awkward: shows a strong consistency --> is very consistent; between each year --> every year.]
Rephrased as follows:
"Figure 3c also highlights a large interannual variability in total HNO$_3$ in R3, while the strong depletion in HNO$_3$ in R2 is consistent every year"

[L223-224: Given the span of PSCs formation over a large range of altitudes --> Given that PSC formation spans a large range of altitudes]
Corrected as suggested

[L224: et al., 150 2006 --> et al., 2006]
Done

[L225: that of maximum IASI sensitivity to HNO3 --> that IASI has maximum sensitivity to HNO3]
Done

[L229: larger temperatures --> higher temperatures]
Done

[L230: The corresponding R2 temperature at 50 hPa to which the ~180 K value at 30 hPa and the ~185 K value at 70 hPa are being compared is not clear in this discussion.]
As stated in the manuscript, the relationship between HNO$_3$ and temperature has been tested at two other pressure levels, namely 70 and 30 hPa. The results exhibit a similar HNO$_3$-temperature behavior at these levels with lower and higher temperatures in R2, respectively, at 30 hPa and at 70 hPa, but still below the NAT formation threshold at these pressure levels. Temperatures down to ~180 K at 30 hPa and down to ~185 K at 70 hPa as compared to temperatures down to ~182 K at 50 hPa are observed. It is now better explained in the corrected manuscript.

[L234: enabling the PSCs formation --> enabling PSC formation]
Done

[L234-235: The "onset of the strong total HNO3 depletion" in Figure 3c clearly occurs more than 5 K below 195 K (i.e., well to the left of the vertical red line). So I do not see how this statement about the consistency between 195 K and the onset of HNO3 depletion is supportable. Perhaps the authors are accounting for the fact that HNO3 starts to condense at temperatures 2–4 K below T$_{NAT}$, but (a) if so they need to state that explicitly, and (b) the difference in Figure 3c is larger than 4 K.]

From Fig.3a, it is true that the onset of the HNO3 depletion occurred in June with temperatures ranging from ~188 K to ~195 K. Note, however, that (as mentioned in the manuscript and the caption), Fig. 3c represents the evolution of daily averaged HNO$_3$ total columns with the highest occurrence; it is not based on averaged values as in Fig.3a. To avoid misunderstanding, "Fig. 3c" has been deleted in the last sentence of the paragraph.

[L251: areas --> area; also, the fact that the PV value specified is for 530 K should be stated]
Done

[L252: regions --> region]
Done

[L253: total HNO3 depletion occurs --> the largest depletion of total HNO3 occurs]
Done

[L258: I have no idea what "as the daily second-difference HNO3 total column" means. The rest of the sentence makes sense, so perhaps this part could simply be deleted. Otherwise, if it is supposed to convey important information, then it has to be rephrased for clarity.]
It indicates that the second derivative of HNO$_3$ total column with respect to time is calculated as the daily second-difference "in" HNO$_3$ total columns. "in" has been added in the sentence.

[Figure 4 and its caption:
o Why is the red line at 195 K dashed? Similar lines in Figure 3 were solid, which would be a better choice here too as the temperature time series is also shown as a red dashed curve.
The red line at 195 K is now solid as suggested.
o L482: temperature --> 50 hPa temperature]
Done

[L262: around the 195 K threshold --> around the time that temperatures drop below the 195 K Threshold]
Done

[L262: "some days" suggests to the reader "a few days", whereas 23 days is more than 3 weeks. Thus: within some days --> within a few days to a few weeks]
Done

[L265-266: The drop temperature in 2014 does stand out to some degree, but nevertheless I do not think that it can simply be excluded from the IASI-mission average just because it is a bit of an outlier. Strong justification is needed to exclude any individual year from the climatological mean, otherwise the authors risk being perceived as "cherry-picking" their results.]
Cfr comment [L28] above. The year 2014 is now taken into account when calculating the 50-hPa average drop temperature. The 50-hPa average drop temperature has, hence, been corrected consequently.

[L269-270: It is very good to remind readers of the meaning of the term "drop temperature", but this definition should come earlier in this section since the term has already been used above (L264, L266).
Corrected

[L270: PSCs formation temperature --> PSC existence temperature]
Done

[L271: could reflect the preponderance by one --> could reflect variations in the preponderance of one]
Done

[L276-278: The average drop temperatures for 30 and 70 hPa appear to have been calculated over the full 10-year record, which supports my contention that 2014 should not be excluded from the 50-hPa average drop temperature calculation.]
See the response to the comment [L265-266] above. 2014 is now included in the calculation of the 50-hPa average drop temperature.

[L284: zonal distribution --> climatological zonal distribution]
Done

[L285: Does Figure 5 use geographic or equivalent latitude? If the former, the difference from other figures (which use EqL) needs to be explained, but in any case it should be made clear.]
Both Figures 5 and 6 which represent zonal distributions use geographic latitudes. This is now clearly mentioned in the revised figure captions. The use of equivalent latitude does not make sense here.

[L287: It is stated that one isocontour of 50 hPa temperature is overlaid, but actually two are shown in Figure 5.]
Corrected in the text.

[L289: lines indicates --> line indicates]
Done

[L290: The average drop temperature is given here as $194.2 \pm 3.8$ K, but above in L265 it was stated to be $194.1 \pm 2.8$ K. These values need to be reconciled.]
See responses to the comments [L265-266] & [L276-278] above: $194.2 \pm 3.8$ K is the 50-hPa average drop temperature including the year 2014, while $194.1 \pm 2.8$ K is the 50-hPa average drop temperature when 2014 is excluded. As suggested by the referee, the year 2014 is now taken into account in the revised manuscript when calculating the average drop temperature.

[L290: rate in --> rate of]
Done

[L291: delete "of some days"]
Done

[Figure 5 and its caption:
o Figure 5b is referred to in L284, but there is no further discussion of this panel in the text. The relationship between IASI total column $HNO_3$ and 50 hPa temperature has already been investigated in connection with Figures 3 and 4; moreover, Figure 5a includes two contours of 50 hPa temperature.

Thus, in the absence of specific discussion about it in the text, Figure 5b seems superfluous and should be deleted]

The two contours of 50 hPa temperature are now clearly specified in the revised text and Figure 5b is now discussed: " … and the isocontours for the 195 K temperature (pink) and for the averaged 194.2 K drop temperature (purple) at 50 hPa. They further illustrate the relationship between the IASI total $HNO_3$ columns and the 50 hPa temperatures." "… It shows that the strongest rate of $HNO_3$ depletion occurs on average end of May, a few days after the temperature decreases below 195 K.".

[o L493: As noted above for the main text, whether this is geographic or equivalent latitude needs to be made clear.]

Done

[o L496: The meaning of "The vertical grey dashed lines encompass the period of the second derivative minima" is unclear – the steepest rate of decrease occurs on a particular date each year, not over a prolonged period. What I believe the authors are trying to say is that the grey lines mark the earliest and latest dates for the drop temperature in the 10-year record, but this statement needs to be written more clearly.]

Re-written as suggested

[o Panel 5b: the date labels (which are nearly illegible) for the overlaid lines indicate that the earliest drop temperature date is 12 May, whereas in L263 it is stated to be 11 May.]

Corrected

[L295: The PV contours are specified at 530 K, not 50 hPa.]

Corrected

[Figure 6 and its caption:

o In the case of Figure 6, panel (b) is not even mentioned in the text. It should be deleted.

For the purpose of the illustration, we would prefer to keep the figure as it is. Figure 6b is now discussed in the revised version: "… as well as the reproducibility of the NAT threshold temperature region that encompasses the inner vortex core. Except for the year 2009, the dates for the strongest rate of $HNO_3$ depletion matches those for the onset of decreasing temperatures below 195 K.".

o The x-axis date labels are illegible.

The labels have been enlarged in the revised version

o L501: Again, whether this is geographic or equivalent latitude needs to be made clear.]

Done

[L302-305: I'm a little confused about how the $-8 \times 10_{-5}$ Km2kg-1s-1 contour of PV comes into the analysis in this section. It is a bit jarring to state in one sentence that the figure examines the region delimited by the –8 PV contour and then in the next sentence characterize the –10 PV contour as delimiting "the region of interest". I presume that the authors want to investigate a region larger than that of the strong depletion in total HNO3 (encircled by the –10 PV contour) while excluding the collar region (–5 PV contour), but that rationale should be stated explicitly.]

This was indeed our intention. The text has been clarified as suggested: "… inside a region delimited by a PV value of $-8 \times 10^{-5}$ $K.m^2.kg^{-1}.s^{-1}$ for each year of the IASI period in order to investigate a region a bit larger than that of the strong depletion in total $HNO_3$ encircled by the PV isocontour of $-10 \times 10^{-5}$ $K.m^2.kg^{-1}.s^{-1}$, averaged over the 10 May – 15 July period for each year, which delimits our region of interest (in green)."

[ Figure 7:

o Similar maps of the corresponding dates are not shown, but they would be a useful addition and would help to clarify some points in the discussion, as noted below.

Such a figure is provided for the referee in Figure 1 below. We find it, however, not necessary for the paper.

o The dark green color for the $-10 \times 10^{-5}$ $Km^2kg^{-1}s^{-1}$ PV contour does not show up well (at least not on my monitor). A brighter green would work better.

The color of that PV contour has been changed for clarity.

o Why are some of the temperature contours not closed?

The contours are not closed in case of missing values for the gridded temperatures.

o The fonts for the years in the map titles in the top row look odd.

We do not see odd fonts for the years in the map titles.

[L303: delete the comma after "$s^{-1}$"]
Done

[L303-307: Switching back and forth between temperature and PV makes these lines harder to follow. It would be better to discuss both the averaged and the minimum PV contours together and then move on to the two temperature contours.]
Changed as suggested.

[L306: The PV contour is specified at 530 K, not 50 hPa.]
Done

[L307: Add a comma after "period".]
Done

[L308-309: I have several comments on this part of the analysis:

o The range of 50 hPa drop temperatures found here is considerably broader than that found in connection with Figure 4 in Section 4.1 (~180–210 K vs ~189–203 K), as is the range of corresponding dates (mid-May to mid-July vs mid-May to early June). These differences between the results based on vortex averages and those based on $1° \times 1°$ bins should be commented on in the text. Do the two approaches agree in terms of which years show generally lower/higher drop temperatures or earlier/later dates? The discrepancies are of concern because the only advantage that the IASI $HNO_3$ total column measurements bring over vertically resolved $HNO_3$ data sets (for which vortex averages can be calculated) is their dense horizontal sampling.

Indeed, the 50 hPa drop temperatures represented on figure 7 and the corresponding dates are broader than the those reported on Figure 4 that represents average values inside the region delimited by a PV isocontour of $-10 \times 10^{-5}$ $K.m^2.kg^{-1}.s^{-1}$. Hence, the two figures are complementary and don't show any mismatch. For instance, the year 2014 that shows the highest drop temperature in Figure 4 is indeed characterized by the highest drop temperatures calculated above the eastern Antarctic. It is now clearly mentioned in the revised manuscript.

o The high extremes in the drop temperature are attributed to issues with the retrievals over eastern Antarctica that are not fully screened out by quality control measures. But Figure 7 shows that unrealistically high drop temperatures are not confined to eastern Antarctica.

This is true, but inside the inner vortex core delimited by the PV isocontour of $-10 \times 10^{-5}$ $K.m^2.kg^{-1}.s^{-1}$, averaged over the period 10 May – 15 July, most of the unrealistic drop temperatures are found above the eastern Antarctic.

o As mentioned above, maps of the corresponding dates would be illuminating. What is the spatial distribution of late dates vs early dates? Do those patterns match the variations in drop temperatures, or do they bear no relation to the temperatures?

As already stated in the manuscript, the area enclosed between the two isocontours of 195 K for the temperatures, the averaged one and the one for the minimum temperatures, shows generally higher drop temperatures and less pronounced minima in the second derivative of the $HNO_3$ total column with respect to time, as well as later corresponding dates (after the strong $HNO_3$ depletion occurs in the inner vortex core, i.e. after the 10 May – 15 July period considered here). This indicates strong relationship to the temperatures. As suggested by the referee, similar maps of the corresponding dates are shown in Figure 1 below for the purpose of the illustration. We can clearly see that the early strong depletion (before 15 June) occur inside the isocontours of $-10 \times 10^{-5}$ $K.m^2.kg^{-1}.s^{-1}$ at 530 K for the averaged PV.

o The authors need to directly confront in the paper the fact that dates as late as mid-July for the onset of $HNO_3$ depletion in the Antarctic are even more implausible than drop temperatures as high as 210 K. These findings will likely cause many readers to dismiss their analysis methodology and/or the use of IASI data for studying PSCs.]

We thank the referee for this recommendation. Some careful sentences were already in the manuscript with respect to these but the word unrealistic has now been introduced twice to make it fully clear. I.e. "Mixing with these already depleted airmasses could also explain the higher drop temperatures detected in those bins. These sometimes unrealistic high drop temperatures are generally detected later (after the strong $HNO_3$ depletion occurs in the inner vortex core, i.e. after the 10 May – 15 July period considered here – not shown), which supports the transport, in those bins, of earlier $HNO_3$-depleted airmasses and the likely mixing at the edge of the vortex."

In addition, it is also stated that: "Except above some parts of Antarctica which are prone to larger retrieval errors and where unrealistic high drop temperatures are found, the overall range in the 50 hPa drop temperature for total $HNO_3$ inside the isocontour for the averaged temperature of 195 K typically extends from ~187 K to ~195 K, which falls within the range of PSC nucleation temperature at 50 hPa". "… and where unrealistic high drop temperatures are found …" has been added here to underline the high drop temperatures calculated over Antarctica.

[L311 & 313: surface --> surfaces]
Done

[L318: Some years have sizeable regions with drop temperatures below 195 K that are well outside the averaged 195 K contour. Hence, encircles well --> encircles fairly well.]
Done

[L319: bins inside that area characterize air masses --> bins inside that area include air masses]
Done

[L322 & 325: show --> shows]
Done

[L327: is also enclosed --> is also typically enclosed]
Done

[L330-331: The weakest minima in the second derivative of total $HNO_3$ (not shown) observed in that area indicate a weak and slow --> The fact that the weakest minima in the second derivative of total $HNO_3$ (not shown) are observed in that area indicates a weak and slow]
Done

[L333-334: Although I think it is good to point out the possible impact of mixing on these results, it should also be acknowledged that previous studies have shown that in the Antarctic mixing between the edge region and the vortex core is generally weak (e.g., Roscoe et al., JGR 2012).]
It is now clearly mentioned in the revised version as suggested: "Note, however, that previous studies have shown a generally weak mixing in the Antarctic between the edge region and the vortex core (e.g. Roscoe et al., JGR 2012)."

[L333: reflect a mixing with strong --> reflect mixing with strongly]
Done

[L334: The mixing --> Mixing; explained --> explain]
Done

[L335-336: I'm confused by this sentence. The drop temperature is defined by the onset of $HNO_3$ depletion, so how can it be that high drop temperatures are detected "after the $HNO_3$ depletion occurs"?]
It has been clarified in the revised text: "after the strong $HNO_3$ depletion occurs in the inner vortex core, i.e. after the 10 May – 15 July period considered here".

[L338-339: What is meant by "the range of maximum sensitivity of IASI to $HNO_3$"? Elsewhere in the manuscript, it is stated that IASI has the largest sensitivity to $HNO_3$ in the 30–70 hPa range, but how is that altitude information relevant to the spatial variations in the maps of Figure 7?]
Changed to: "these spatial variations might also partly reflect some uncertainty into the drop temperature calculation, introduced by the use of temperature at a single pressure level (50 hPa) and of PV on a single potential temperature surface (530 K) while the sensitivity of IASI to changes in the $HNO_3$ profiles extends over a range from ~30 to ~70 hPa during the polar night."

[L339-346: The second half of the sentence in L339 ("while biases …") starts a new discussion on reanalysis temperature and is followed by several related sentences, so it should be a separate sentence (not starting with "while"). In addition, it is not clear to the reader why all of the detail presented in the following sentences is really necessary. It would be better to either end the paragraph by stating explicitly that ERA-I temperature biases of the magnitude noted in these lines could not possibly account for the large range of calculated drop temperatures, or simply delete some of the details.]
This sentence has been rewritten in order to address the comment of the referee. Details related to the uncertainties in ECMWF ERA Interim temperature have been deleted.

[L339: for explaining --> to explain]
Done

[L344: just to be clear, add "in modern reanalyses" after "reduced".]
This part of the sentence has been deleted (see comment above).

[L348: drop 50 hPa temperature --> 50 hPa drop temperature; delete the comma after "195 K"]
Done

[L349: PSCs nucleation --> PSC nucleation]
Done

[L351: on the type of formation mechanisms --> on the specific formation mechanism (i.e., the type of PSC developing)]

Done

[L353: coverage of IASI that allows capturing the rapid and critical depletion phase --> coverage of IASI, which allows the rapid and critical depletion phase to be captured in detail]
Done

[L357: columns dataset --> column dataset]
Done

[L358: since other IASI instruments are mentioned later: Metop --> Metop-A]
Done

[L363-364: I find this wording unclear. I suggest: level over a range where --> level, which lies in the range where; process occur --> processes occur]
Done

[L367: delete "various"]
Done

[L368: delete "and described along the cycle" (this wording is confusing and unnecessary)]
Done

[L369: delete "at play"]
Done

[L370: Only Antarctica is considered here, thus: in the poles --> over Antarctica]
The sentence "R1 is defined during April and May and characterized by a rapid decrease in 50 hPa temperatures while HNO$_3$ accumulates in the poles" remains as it is. The strong depletion occurs in the area defined by a PV of $-10 \times 10^{-5}$ K.m$^2$.kg$^{-1}$.s$^{-1}$ at 530 K (typically over Antarctica indeed) but HNO$_3$ accumulates outside that area over the poles.

[L370: As mentioned earlier when this regime was defined, R2 starts in June but the uptake of HNO$_3$ into PSCs starts in mid-May, as shown in this paper and in previous studies.]
See comment [L193] above. Changed to: "R2, from June to September, follows the onset of the depletion that starts around mid-May in most years when the 50 hPa temperatures fall".

[L371-372: PSCs nucleation --> PSC nucleation]
Done

[L372: between each year --> from year to year]
Done

[L372: R3 is actually defined (L198, Figure 3a) to begin in October, not November.]
Corrected

[L373: until March --> through March]
Done

[L374: PSCs sedimentation at --> PSC sedimentation to]

Done

[L376: found particularly --> found to be particularly]
Done

[L377: condensation to --> condensation into]
Done

[L379: 2.8 --> 2.8 K; also, as noted previously, the inconsistency in the average drop temperature values given in the text (L265 and L290) needs to be fixed and the correct one quoted here. L380: As noted above in Section 4.1, I do not think that the omission of 2014 from the climatological average is justified.]
Corrected. See our response to the comment [L290] above.

[L381: demonstrated --> demonstrate; PSCs formation --> PSC formation]
Done

[L384: PV at 50 hPa --> PV at 530 K]
Done

[L386: highest minima --> lowest minima]
Done

[L388: "from year to year" is not the right phrase; perhaps the authors mean "in some years"]
Corrected

[L388: As mentioned earlier, not all of the unrealistically high drop temperatures were calculated over eastern Antarctica]
See our response to the comment [L308-309] above.

[L390: found in line --> found to be in line; PSCs nucleation --> PSC nucleation]
Done

[L395-396: It likely results --> These likely result; a mixing --> mixing]
Done

[L399: over the whole polar regions --> over the whole Antarctic region]
Done

[L401-403: I do not see how the authors could make the statement that the IASI dataset offers a new observational means to monitor the relation of $HNO_3$ to temperature and PSC formation because it can make measurements in darkness. It is certainly understandable that they want to tout IASI's excellent spatial coverage and its potential for a long record. Those are indeed very valuable contributions. But it is simply not acceptable to ignore decades of $HNO_3$ measurements made "throughout the year (including the polar night)" by numerous satellite instruments (e.g., LIMS, UARS MLS, Aura MLS, CLAES, MIPAS, SMR, ILAS, SMILES).]
It is of course not our intention to ignore the long-time series already available from other instruments, which are referenced in the introduction, but rather to underline the interest of the IASI record for the

future. In order to avoid a feeling of overselling "…IASI dataset offers new observational means to…" has been changed to "…IASI dataset offers a valuable observational means to…".

[L401-402: delete "capturing"; add "to be captured" at the end of the sentence]
Done

[L404: allow to investigate --> allow investigation of]
Done

**Figure captions**

[Figure]

**Figure 1.** Spatial distribution (1°×1°) of the dates corresponding to the drop temperatures at 50 hPa (calculated from the total $HNO_3$ second derivative minima) for each year of IASI (2008–2017), in a region defined by a PV of $-8\times10^{-5}$ K.m$^2$.kg$^{-1}$.s$^{-1}$. The isocontours of $-10\times10^{-5}$ K.m$^2$.kg$^{-1}$.s$^{-1}$ at 530 K for the averaged PV (in red) over the same period are represented.

---

## Author Response (AR4)

**Response to reviewer #1**

We thank the reviewer for his/her new technical corrections. We hope that, with the corrections suggested by the two referees, the revised manuscript will convince the reviewers that the paper can be published in ACP.

**Response to reviewer #4 – M. Santee**

We deeply thank Michel Santee for her new in-depth review and her new suggestions to clarify unclear sentences. The corrections made in the new version improve considerably the manuscript. We only address the main comments here below, using the same line numbers as the ones quoted by Michel Santee in her review. All the minor comments, suggestions and technical corrections have been addressed/implemented in the new version of the manuscript.

**Main comments**

**Section 2**
[L122-123, 131]:
"polar night" and "dark Antarctic winter/vortex" refer to nighttime only. There is no mention of diurnal variations anywhere in the manuscript. "...variations of HNO3..." has been changed to "depletion of HNO3" for clarity in the new version.

**[L132 and Fig. 1 caption]:**
"(Both divided by 10)" has been added in the caption of Figure 1.

**Section 3**
**[L189-196]:**
The sentence: "The end of the R1 period marks the start of the strong total $HNO_3$ decrease that intensifies later in R2." has been added at the end of the paragraph.

[L223-225]:
The sentence has been deleted.

**Section 4**
[L268]:
The sentence and the label on figure 5 have been corrected.

[L303-306]:
1/ The sentence has been changed to: "The red vertical dashed line indicates the annual average of the dates on which the 50 hPa drop temperatures are calculated in the area of $PV \leq -10 \times 10^{-5}$ $K.m^2.kg^{-1}.s^{-1}$ (194.2 ± 3.8 K; see Fig. 4)."
2/ The average date is now specifically mentioned in the text.
3/ The fact that the average date just precedes the PV contour on Figure 5 is explained by the fact the we represent a climatological PV contour based on zonal averages of PV values (contrarily to Figure 4 where the date of the 50 hPa drop temperature indeed coincides or just follows the existence of an area of the $-10 \times 10-5$ K.m2.kg−1.s−1 PV value) and an average date. One can clearly see on Figure 5 that the existence of the area within the PV contour falls into the two dashed vertical lines that encompass the dates on which the drop temperature is calculated. When looking at Figure 6 that illustrates every

year, we clearly see that the dates corresponding to the 50 hPa drop temperatures better match the existence of the area of PV <= -10x10-5 k.m2.kg-1.s-1.

4/ The fact that a climatological and zonally averaged PV contour is used in now explicitly stated in the text and in the figure caption.

[L306-307] and [L312-313]:

We thank M. Santee for pointing this out.

1/ The sentence L306-307 is indeed unclear and misleading. What was underlined here is the delay between the detection of the averaged 195 K threshold temperature and the start of the $HNO_3$ depletion.

Because of the redundancy with the previous sentence, it has been moved below when discussing figure 6 and it now replaces L312-313:

"An exact timing or a delay of a few days between the detection of the averaged 195 K threshold temperature and the start of the $HNO_3$ depletion is visible every year in Fig. 6. In particular, the year 2009 shows the longest delay (see also Fig. 4)".

2/ The delay is also now specifically mentioned above when discussing Figure 4: "... (at exactly or a few days after the detection of the 195 K threshold temperature, particularly for the year 2009) ...".

3/ Actually, the mismatch in the 10-year average (Fig. 5) between the detection of the averaged 195 K threshold temperature and the average date (24 may) for the drop temperatures is not driven by the year 2009 (that in fact has the latest date for the drop temperature - the 8th of June - among all years; see text), but by the year 2013 that shows the earliest date for the drop temperature (11th of May; see text) due to the lowest temperatures in the Antarctic winter. It is now clarified at the end of section 4.1:

"Note that the mismatch observed in the 10-year average between the detection of the averaged 195 K threshold temperature and the average date for the drop temperatures (see Fig. 5 a and b) is driven by the year 2013 which is characterized by the lowest temperatures during the Antarctic winter over the 10-year study period and, hence, the earliest date for the drop temperature (11th of May; see Fig. 4 and Fig. 6)."

[Figure 7]:

The fact that some temperature contours are not closed is explained by the fact that we work in an area delimited by a PV value <= $-8\times10^{-5}$ $k.m^2.kg^{-1}.s^{-1}$. Please see, as an example, figure 1 here below that shows the spatial distribution of the temperatures (K) at 50 hPa averaged over the period 10 May –15 July for the year 2015, in a region delimited by a PV of $-8\times10^{-5}$ $K.m^2.kg^{-1}.s^{-1}$. The isocontours of 195 K at 50 hPa for the minimum (in pink) and the averaged (in red) temperatures as well as the isocontours of $-10\times10^{-5}$ $K.m^2.kg^{-1}.s^{-1}$ and of $-8\times10^{-5}$ $K.m^2.kg^{-1}.s^{-1}$ at 530 K for the minimum PV (in green and in cyan, respectively) are represented.

[L329-336]:

The fact that the Antarctic land in 2014 shows a lot of cells characterized by a high drop temperature has to be investigated in more details. Interestingly, May 2014 (and also 2016) shows a significant positive land surface temperature anomaly in that region over the 10-year study period. It probably induces measurements with a better signal-to-noise-ratio, which are less prone to rejection based on the applied quality filters, while they remain characterized by strong emissivity features. This may bias the drop temperature calculation.

[Figure]

**Figure 1.** Spatial distribution (1°×1°) of the temperatures at 50 hPa (K), averaged over the period 10 May –15 July of the year 2015, in a region defined by a PV of $-8\times10^{-5}$ K.m$^2$.kg$^{-1}$.s$^{-1}$. The isocontours of $-10\times10^{-5}$ K.m$^2$.kg$^{-1}$.s$^{-1}$ at 530 K for the averaged PV (in green) and the minimum PV (in cyan) encountered over the selected period and the isocontours of 195 K at 50 hPa for the averaged (in red) and the minimum (in pink) temperatures over the same period are represented.

---

## Author Response (AR5)

**Response to reviewer #4 – M. Santee**

We thank Michel Santee for her new corrections to clarify unclear sentences. All the minor comments and technical corrections have been addressed/implemented in the new version of the manuscript, as follows:

[L139-141: This sentence is written in an awkward and confusing manner (plus there is a stray ")" at the end). If I understand it correctly, it would be clearer to say something like: "Together, weaker sensitivity above very cold surfaces with a degrees of freedom for signal (DOFS) of 0.95 and poor knowledge of the seasonally and wavenumber-dependent emissivity above ice surfaces induce larger forward model errors, and consequently the largest measurement errors occur over the Antarctic."]
Rewritten for clarity: "The highest retrieval error measured over the Antarctic arises from a weaker sensitivity above very cold surface with a degrees of freedom for signal (DOFS) of 0.95, as well as from a poor knowledge of the seasonally and wavenumber-dependent emissivity above ice surfaces."

[L157: number of iteration --> number of iterations]
Done

[L225: This sentence refers to the "red vertical line" in Fig. 3c, whereas the Fig. 3 caption mentions "The orange horizontal or vertical lines". It would be better to draw all of the lines marking 195 K in Fig. 3 in the same color and refer to them consistently.]
Corrected: "The red …" → "The orange …"

[L226: a large interannual variability --> the large interannual variability]
Done

[L266-267: "at exactly or a few days after the detection of the 195 K threshold temperature" is awkward and unclear wording. Moreover, I do not think that "detection" is the right word here – it is not that the 195 K threshold is being "detected", but rather that it is being crossed. Finally, the only additional information that this phrase conveys beyond "around the time that temperatures drop below the 195 K threshold", as already stated in L266, is that sometimes the strongest rate of HNO 3 depletion is seen a few days after the 195 K threshold is crossed, particularly in 2009. However, emphasizing the delay obscures the fact that occasionally the strongest rate of HNO 3 depletion appears to *precede* (not follow) the date on which temperatures drop below 195 K, as in 2013 (and to a lesser extent 2014 as well), according to Fig. 4. Thus I feel that it would be better to simply delete that entire parenthetical comment.]
Rewritten for clarity: "… is found closely around the time that temperatures drop below the 195 K threshold (except for the year 2009 that shows a longest delay) …"

[L304: I find the insertion of the word "annual" in front of "average" in this line confusing. My understanding is that the red vertical dashed lines mark the 10-yr (2008–2017) average of the dates corresponding to the 50-hPa drop temperatures found for each year. As such, this value does not represent an "annual average".]

[L311: A closing ")" is missing after "530 K".]
Added

[L311-312: My previous comments about L266-267 also apply to the phrase "An exact timing or a delay of a few days between the detection of the averaged 195 K threshold temperature ...".]
Rephrased to: "Like for Fig.4, an exact timing or a few days between the time that temperatures drop below the 195 K threshold and the start of the $HNO_3$ depletion is visible every year in Fig. 6. A longest delay is also observed for the year 2009."

[L313-315: Again, "detection" is not really the right word; also, the sentence is grammatically awkward. I suggest instead "The mismatch between the 10-year averages of the dates on which the 195 K temperature threshold is crossed and the dates for the drop temperatures (see Fig. 5 a and b) is driven by the year 2013, which ..." (i.e., add the comma after "2013")]
Changed to: "Note that the mismatch between the 10-year average of the dates on which the 195 K threshold temperature is reached and that of the dates for the drop temperatures (see Fig. 5 a and b) is driven by the year 2013, which …"

[L325 and 327: $10^{-5}$ K.m2.kg−1.s−1 --> $10^{-5}$ K.m2 .kg−1 .s −1 (missing superscripts)]
Done